# Clinical-grade autonomous cytopathology through whole-slide edge tomography

Nao Nitta[1,16 ✉], Yuko Sugiyama[2,3,16], Takeaki Sugimura[1], Takahiko Ito[2], Koichi Ikebata[2], Hitoshi Abe[2], Shuhei Ishii[4], Hiroyuki Kanao[3], Nagisa Hosoya[1], Raihan Ull Islam[1], Aditya Jain[1], Meisam Hasani[1], Joseph Zonghi[1], Peter Koh[1], Yukihito Mase[1], Miki Kanematsu[1], Noureldin M. Z. Ali[1], Yoshihiko Murata[5], Ayumi Shikama[6], Yusuke Kobayashi[6], Daisuke Matsubara[5], Yukari Himeji[7], Hiroshi Nakamura[8], Akane Hashizume[8], Miyaka Umemori[9], Hiroyuki Ohsaki[9], Yingdong Luo[10], Tianben Ding[10], Fernando C. Schmitt[11], Robert Y. Osamura[1,12], Tomohiro Chiba[2,4 ✉] & Keisuke Goda[1,10,13,14,15 ✉]

Cytopathology, often abbreviated as cytology, has a central role in the early detection of cancer, such as cervical, lung and bladder cancers, owing to its speed, simplicity and minimally invasive nature[1–9]. However, its effectiveness is limited by variability in diagnostic accuracy stemming from subjective visual interpretation[10–21]. Although many artificial intelligence (AI)-powered systems have been proposed to improve consistency[22–26], none have achieved fully autonomous, clinical-grade performance. Existing approaches serve as assistive tools and still rely on human oversight for interpretation and decision-making[22–26]. Here we present a clinical-grade autonomous cytopathology pipeline that combines high-resolution, real-time optical whole-slide tomography with edge computing to deliver end-to-end automation. The system achieves practical performance in imaging speed, quality and data volume, with localized data compression enabling streamlined storage and accelerated AI-driven analysis. In addition to supporting cell-level classification, the platform enables flow cytometry-like, population-wide morphological profiling for comprehensive interpretation of cellular distributions and patterns. A vision transformer achieved area under the receiver operating characteristic (ROC) curve (AUC) values exceeding 0.99 at the single-cell level for detecting low-grade squamous intraepithelial lesions (LSILs), high-grade squamous intraepithelial lesions (HSILs) and adenocarcinoma. In a multicentre evaluation of 1,124 cervical liquid-based cytology samples across four centres, the AI model achieved slide-level AUC values of 0.86–0.91 for LSIL[+] and 0.89–0.97 for HSIL[+], with LSIL counts correlating strongly with human papillomavirus positivity and HSIL counts scaling with diagnostic severity. The system enables autonomous triage cytology, offering a foundation for routine, scalable and objective diagnostics.

Cytology lies at the core of early detection of cervical, lung and bladder cancers because it is minimally invasive, low cost and widely deployable[1–7]. In routine practice, cytologists examine glass slides containing approximately 10,000–1,000,000 cells per slide, assessing individual cells and clusters under an optical microscope. Diagnostic judgements rely on abnormalities in three-dimensional (3D) nuclear and cytoplasmic morphology and spatial relationships between neighbouring cells. Cervical screening, in particular, is dominated by liquid-based Pap testing[8,9], which has reduced cervical

cancer incidence and mortality by improving sample quality and throughput[4,5].

Despite these strengths, cytology exhibits variable sensitivity and specificity because it fundamentally depends on subjective visual interpretation[10,11]. Inter-observer and intra-observer variability arises from differences in training and experience, as well as from cognitive biases such as confirmation and anchoring[12–16]. Incomplete adherence to guidelines and quality-control protocols[17,18], together with fatigue, time pressure and high case volumes[16], further degrades accuracy

[1]CYBO, Tokyo, Japan. [2]Department of Cytology, Cancer Institute Hospital of JFCR, Tokyo, Japan. [3]Department of Gynecology, Cancer Institute Hospital of JFCR, Tokyo, Japan. [4]Department of Pathology, Cancer Institute Hospital of JFCR, Tokyo, Japan. [5]Department of Pathology, University of Tsukuba, Tsukuba, Japan. [6]Department of Obstetrics and Gynecology, University of Tsukuba, Tsukuba, Japan. [7]Kaetsu Comprehensive Health Development Center, Shibata, Japan. [8]Department of Pathology, Juntendo University Urayasu Hospital, Urayasu, Japan. [9]Department of Clinical Laboratory Technology, Juntendo University, Urayasu, Japan. [10]Department of Chemistry, University of Tokyo, Tokyo, Japan. [11]RISE-Health, Department of Pathology, Medical Faculty of Porto University, Porto, Portugal. [12]Department of Diagnostic Pathology, Nippon Koukan Hospital, Kawasaki, Japan. [13]Institute of Technological Sciences, Wuhan University, Wuhan, China. [14]International Center for Synchrotron Radiation Innovation Smart, Tohoku University, Sendai, Japan. [15]Department of Bioengineering, University of California, Los Angeles, CA, USA. [16]These authors contributed equally: Nao Nitta, Yuko Sugiyama. ✉e-mail: nitta@cybo.co.jp; tomohiro.chiba@jfcr.or.jp; goda@chem.s.u-tokyo.ac.jp

and contributes to missed or delayed diagnoses. The CervicalCheck cancer scandal[19–21] illustrates the potential clinical consequences of subjective cytology. Manual review is also challenged by the low prevalence of abnormal cells, preparation artefacts and overlapping cellular structures, which obscure findings and make early lesions difficult to detect[22,23].

Artificial intelligence (AI) has therefore been widely explored to support cytological interpretation[15,22,24–26]. However, most approaches operate on two-dimensional (2D) images and target small subsets of 'representative' cells. More fundamentally, existing frameworks rarely scale to whole-slide analysis of hundreds of thousands of spatially dispersed cells seen in routine practice. Although 3D imaging can capture richer structural information that better reflects how cytologists interpret slides, it substantially increases demands on image acquisition, processing, storage and data transfer. Compared with histopathology and radiology[27–32], cytology generates larger data volumes because morphologically diverse, non-cohesive cells must be imaged in depth. These constraints have limited the scalability of AI models, leaving current tools assistive and reliant on human interpretation and decision-making rather than functioning autonomously[15,22,24–26].

To overcome these limitations, we present a real-time, clinically validated autonomous cytology platform that integrates high-speed, high-resolution optical whole-slide tomography with edge computing—a distributed computing architecture that processes data close to its source[33–36]. The system acquires and locally compresses gigavoxel 3D whole-slide images before storage, accelerating model development and deployment without compromising image quality or AI performance. This enables routine digitization of thick cytology samples containing abnormal cell clusters, a long-standing challenge[37–39], and delivers strong AI performance without requiring very large datasets; approximately 1,000 original 3D images per class are sufficient for effective AI training.

A key innovation is the cluster of morphological differentiation (CMD), an image-derived analogue of the cluster of differentiation used in immunophenotyping[40,41]. The CMD supports a flow cytometry-like framework on the basis of morphology rather than fluorescence markers, enabling population-wide visualization and interrogation. Unlike earlier AI-based cytology tools[15,19–22,24–26], CMD analysis lets cytologists explore cell populations through scatter plots, hierarchical gating and dimensionality reduction, improving interpretability, error detection and discovery of new phenotypes. Together, these advances establish a scalable, real-time cytology pipeline with clinical-grade autonomy and lay the foundation for an objective, reproducible and discovery-driven diagnostic paradigm.

## Whole-slide edge tomography

As illustrated in Fig. 1, our edge computer-integrated optical whole-slide tomograph, referred to as the whole-slide edge tomograph, comprises a light-emitting diode as a light source, an *XY* translation stage, a *Z* translation stage integrated with imaging optics and a complementary metal–oxide–semiconductor (CMOS) image sensor, and an edge computer equipped with an image sensor field-programmable gate array (FPGA) and a system on module (SOM) for real-time digital image processing (see Extended Data Fig. 1 for details). This configuration enables the acquisition of 2D images across multiple depth layers and facilitates the construction, compression and archiving of 3D images during slide scanning in the lateral (*XY*) and longitudinal (*Z*) directions. The CMOS image sensor captures high-resolution 2D bright-field images (4,480 × 4,504 pixels per imaging section) at a rate of up to 50 frames per second, with 173 or 485 imaging sections per layer for Sure-Path (Becton, Dickinson and Company) or ThinPrep (Hologic) slides, respectively, and 40 layers per slide, yielding approximately 140 or 391 gigavoxels per slide, respectively. These 2D images are transmitted to the FPGA for initial image processing. The processed images are then

sent to the graphics processing unit (GPU) through a dual four-lane Mobile Industry Processor Interface (MIPI), in which extra tasks, such as background correction, focus adjustment, 3D image construction and 3D image compression, are performed using the hardware encoder, a dedicated module in the SOM for real-time image compression. Similar to video compression, in which redundancies within each frame (intra-frame) and between consecutive frames (inter-frame) reduce the data size, our system exploits intra-layer compression by reducing spatial redundancy within individual 2D images and inter-layer compression by exploiting similarities between adjacent optical sections along the *Z* axis. This strategy enables efficient compression of sectional 3D image stacks while preserving diagnostic content. The compressed images are stored in the high-efficiency video coding (HEVC) format, which supports both intra-prediction and inter-prediction schemes and is well suited for volumetric image sequences. These sectional 3D images are then transmitted to a back-end server, where they are decoded and stitched into a comprehensive 3D image of the entire slide, covering approximately 10,000 to 1,000,000 cells. This whole-slide 3D image can be viewed in real time by cytologists on a high-definition monitor, allowing interactive functions, such as movement, magnification and focus adjustments, using a deep zoom image (DZI) viewer. Simultaneously, the 3D image undergoes AI-based population analysis. Nuclei within individual cells are detected using the high-performance object detection model YOLOX[42], followed by cropping the best-focused objects and classification using the vision transformer model MaxViT[43]. Additionally, population analysis is conducted using the CMD, with the results provided to the cytologists. Both the 3D image and AI-generated analysis results are accessible in real time, supporting efficient diagnosis and evaluation.

## System performance

As shown in Extended Data Fig. 2, the whole-slide edge tomograph achieves high-quality imaging and efficient data compression. Extended Data Fig. 2a presents the image quality under different HEVC compression settings (high, medium and low) across three liquid-based cytology slides. For each condition, 300 tomographic images (3 slides × 10 imaging sections × 10 layers) were acquired, and peak signal-to-noise ratio (PSNR) distributions are shown as histograms. Extended Data Fig. 2b summarizes the corresponding file sizes of the 3D whole-slide images, demonstrating an inverse relationship between compression level and data size: approximately 1 GB, 500–800 MB and 170 MB for high, medium and low compression of a ten-layer SurePath slide, respectively. File size scaled linearly with the number of *Z* layers. Extended Data Fig. 2c shows representative tomograms (2D cross-sections) of glandular, low-grade squamous intraepithelial lesion (LSIL), high-grade squamous intraepithelial lesion (HSIL) and adenocarcinoma cells across varying PSNR levels. The system resolves subcellular structures, such as nucleoli and nuclear membranes, with high fidelity at PSNR ≥ 40 dB, corresponding to medium-to-high compression. Extended Data Fig. 2d–f details the imaging timeline across different *Z*-layer settings (10, 20 and 40 layers) for the first eight imaging sections. The time logs break down the process into *XY* stage motion, image acquisition, reconstruction and compression. These operations were pipelined, allowing computation and data encoding to proceed in parallel with mechanical movements, minimizing idle time (Extended Data Fig. 2g,h). Extended Data Fig. 2i quantifies task latency per imaging section, showing that while *XY* motion time remained constant, acquisition, processing and encoding times scaled proportionally with the number of *Z* layers. Extended Data Fig. 2j reports total imaging times per slide of approximately 3 min, 4.5 min and 8 min for 10, 20 and 40 layers, respectively, with minimal influence from compression setting. Furthermore, Extended Data Fig. 3 confirms real-time system responsiveness; even with on-the-fly HEVC decompression during viewing, most high-resolution tile requests were completed within 100 ms. These results collectively demonstrate the

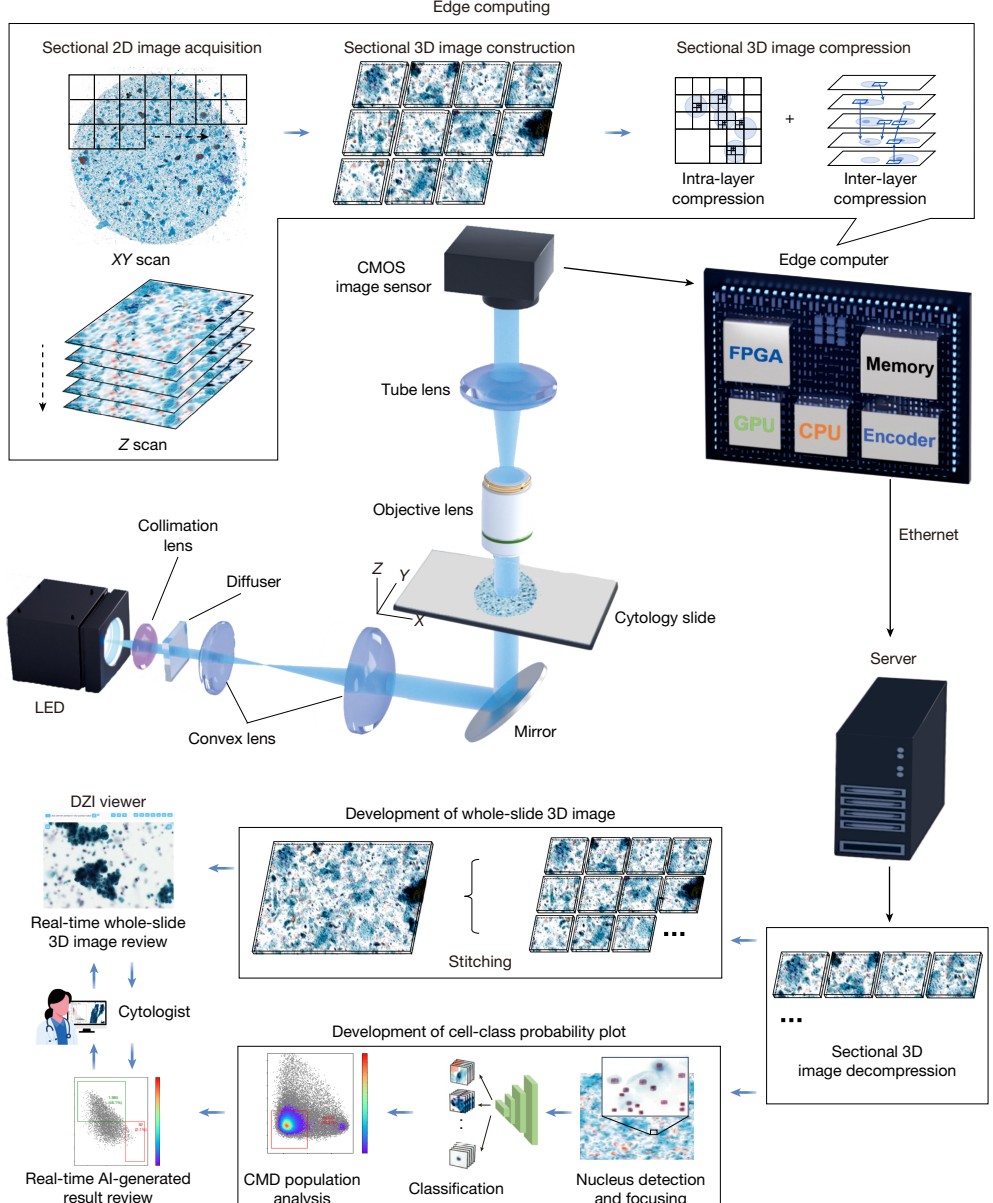

**Fig. 1 | Whole-slide edge tomography.** The edge computer facilitates the acquisition of 2D images across multiple depth layers and enables the construction, compression and archiving of 3D images during slide scanning in both the lateral (*XY*) and longitudinal (*Z*) directions. The CMOS image sensor captures high-resolution 2D bright-field images (4,480 × 4,504 pixels per imaging section) at up to 50 frames per second, with 173 or 485 imaging sections per layer and 40 depth layers per slide, yielding approximately 140 or 391 gigavoxels per SurePath or ThinPrep slide, respectively. These 2D images are first transmitted to the FPGA within the edge computer for initial processing. Subsequently, they are sent to the GPU, where further tasks such as background correction, focus adjustment, 3D image construction and compression are carried out. The processed sectional 3D images are then transmitted to a server, where the GPU stitches them together into a comprehensive 3D representation of the entire slide, encompassing approximately 10,000 to 1,000,000 cells. This whole-slide 3D image can be viewed in real time by cytologists on a high-definition monitor, enabling interactive functions such as movement, magnification and focus adjustments. Simultaneously, AI-based population analysis is performed on the 3D image. Population analysis is further refined using the CMD, and the results are provided to the cytologists. Both the 3D image and AI-generated analysis results are accessible in real time, enabling efficient diagnosis and evaluation.

predictable, high-performance operation of the system across varying imaging conditions.

## Imaging performance

The whole-slide edge tomograph enables rapid, high-resolution 3D visualization of thick cytology samples, including Pap smears, sputum smears and brush biopsy smears, as well as liquid-based cytology preparations, such as ThinPrep, SurePath and body fluid samples. The system achieves a lateral (*XY*) resolution of 220 nm and an axial (*Z*) resolution of 1 μm, generating approximately 140 gigavoxels per SurePath slide or 391 gigavoxels per ThinPrep slide. This resolution enables detailed assessment of cellular architecture, structural deformation and abnormal morphology in any imaging plane, including oblique sections. To demonstrate its capabilities, we acquired six 3D imaging datasets at 1-μm *Z* intervals, representing keratinizing and non-keratinizing squamous cell carcinoma (SCC), human papillomavirus (HPV)-associated and HPV-independent adenocarcinoma, fibroadenoma (benign breast tumour) and follicular thyroid neoplasm (follicular thyroid carcinoma). Representative tomographic slices extracted at 5-μm intervals are

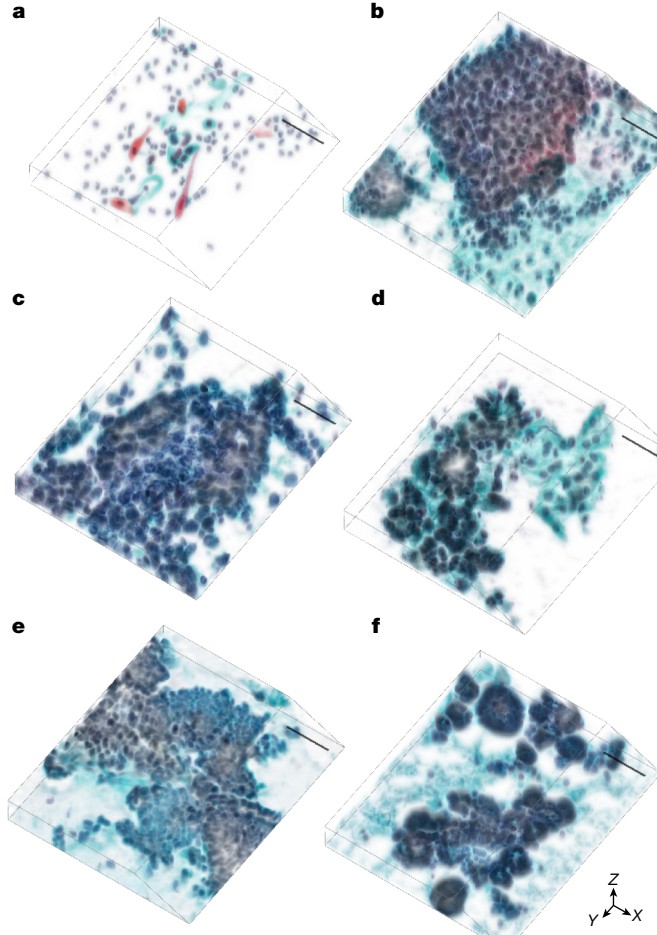

**a**, **b**, **c**, **d**, **e**, **f**

Z
Y X

**Fig. 2 | Representative reconstructed 3D images of cytology samples.**
**a**, Keratinizing SCC. **b**, Non-keratinizing SCC. **c**, HPV-associated adenocarcinoma.
**d**, HPV-independent adenocarcinoma. **e**, Fibroadenoma (benign breast tumour).
**f**, Follicular thyroid neoplasm (follicular thyroid carcinoma). These samples
correspond to those shown in Extended Data Fig. 4 and are visualized along the
*XYZ* axes to reveal volumetric cellular architectures and spatial relationships.
The images highlight depth-resolved morphological features and structural
deformations. *Z* intervals, 1 μm. Scale bars, 50 μm.

shown in Extended Data Fig. 4, showing distinct nuclear irregularity,
chromatin texture and nuclear-to-cytoplasmic ratio—features not
discernible in 2D imaging. The corresponding 3D reconstructions are
shown in Fig. 2, visualized along the *XYZ* axes to emphasize volumetric
structure. Supplementary Videos 1–12 further illustrate these data-
sets in both 2D and 3D, providing dynamic visualizations that enhance
understanding of their spatial and morphological characteristics.

## AI cell detection and classification

The availability of high-quality 3D images enables the development
of a high-performance AI model for accurate cell classification. The
complete workflow, from cell detection to classification, is outlined
in Supplementary Fig. 1. Initial 3D cell detection was performed on
whole-slide tomographic images acquired at 3-μm intervals using a
YOLOX-based model, trained and validated on 242,669 annotated
nuclei across 348 images (Supplementary Figs. 2 and 3). Detected
nuclear centroids were used to extract in-focus images of individual
cells, following the focus-selection procedure detailed in Methods.
These focus-refined images were then processed using a MaxViT-based
vision transformer for cell type classification. Extended Data Fig. 5a
presents a confusion matrix comparing expert cytologist annotations

with AI predictions. The model was trained and validated on a dataset
of 168,569 augmented images (rotation and flipping) derived from 354
donor-based whole-slide images, with 50,222 images reserved for vali-
dation (Supplementary Fig. 4). The classifier achieved high accuracy,
with specificity exceeding 98% across all classes. Extended Data Fig. 5b
shows receiver operating characteristic (ROC) curves, with area under
the ROC curve (AUC) values greater than 0.99 for all classes, except
glandular cells and squamous metaplasia. The relatively lower clas-
sification performance of glandular cells can be attributed to intrinsic
variability in nuclear polarity and frequent clustering, which complicate
single-cell morphological assessment even for expert cytologists.
Although perfect accuracy in this class is not critical for the present
objectives, it enhances the capacity of the model to capture diagnosti-
cally relevant cellular diversity. Extended Data Fig. 5c–k shows repre-
sentative tomograms and corresponding AI inference results for key
gynaecological cytology cell types. Extended Data Fig. 5c–h includes
leukocytes, superficial/intermediate squamous cells, parabasal cells,
squamous metaplasia, glandular cells and miscellaneous clusters.
Extended Data Fig. 5i–k highlights the LSIL, HSIL and adenocarcinoma
cells. These results demonstrate the high sensitivity and reliability of
the model, even in the context of dense or aggregated cell populations.

## CMD cell population analysis

We characterized cell populations using the CMD framework and visual-
ized them in a statistically robust and interpretable manner by incorpo-
rating AI-derived class probability values as axes in scatter plots. In this
framework, each cell is represented by a ten-dimensional class probabil-
ity vector (see Methods for details), which serves as the CMD markers.
Figure 3a,b shows analyses from cytological samples diagnosed as nega-
tive for intraepithelial lesion or malignancy (NILM), whereas Fig. 3c,d
corresponds to samples diagnosed as LSILs and HSILs, respectively. Each
panel includes four plots (from left to right): (1) a scatter plot for separat-
ing leukocytes and irrelevant objects; (2) a histogram of LSIL probability
scores; (3) a histogram of HSIL probability scores; and (4) a uniform
manifold approximation and projection (UMAP) plot. A gating strategy is
applied to the scatter plots to isolate the bottom-left population, consist-
ing primarily of epithelial cells, by excluding leukocytes and debris. The
histograms are generated from this gated population, with thresholds
applied to identify LSIL-positive and HSIL-positive cells; absolute counts
and proportions are annotated. The UMAP plots visualize the spatial dis-
tribution of gated cells, with colour-coded cell types enabling intuitive
exploration of the cytological landscape. In Fig. 3b, eight representative
points (p1–p4 and q1–q4) illustrate two morphological trajectories:
p1–p4 from parabasal to superficial/intermediate squamous cells and
q1–q4 from glandular to metaplastic cells. The corresponding images
are shown in Fig. 3e, capturing gradual transitions in morphology and
suggesting that the UMAP encodes a pseudo-histological distribution
akin to in vivo tissue organization. In the UMAP plots of Fig. 3c,d, LSIL
and HSIL cells are indicated by orange and red arrows, respectively. Rep-
resentative cell images from these regions, shown in Fig. 3f–h, confirm
the extraction of cells exhibiting cytological atypia consistent with the
corresponding diagnostic categories.

## Clinical-grade performance

To demonstrate the clinical-grade performance of our autonomous
cytology platform, comprising the whole-slide edge tomograph and
the CMD-based analysis, we evaluated cervical liquid-based cytology
samples from 318 donors at the Cancer Institute Hospital of JFCR. For
each case, the platform quantified the number of normal, LSIL, HSIL and
adenocarcinoma cells. Figure 4a,b presents the whole-slide cell counts
by class. Figure 4a shows the results for all cell types, whereas Fig. 4b
focuses on positive (abnormal) cell classes. In both panels, slides are
grouped by cytology result along the horizontal axis and sorted by total

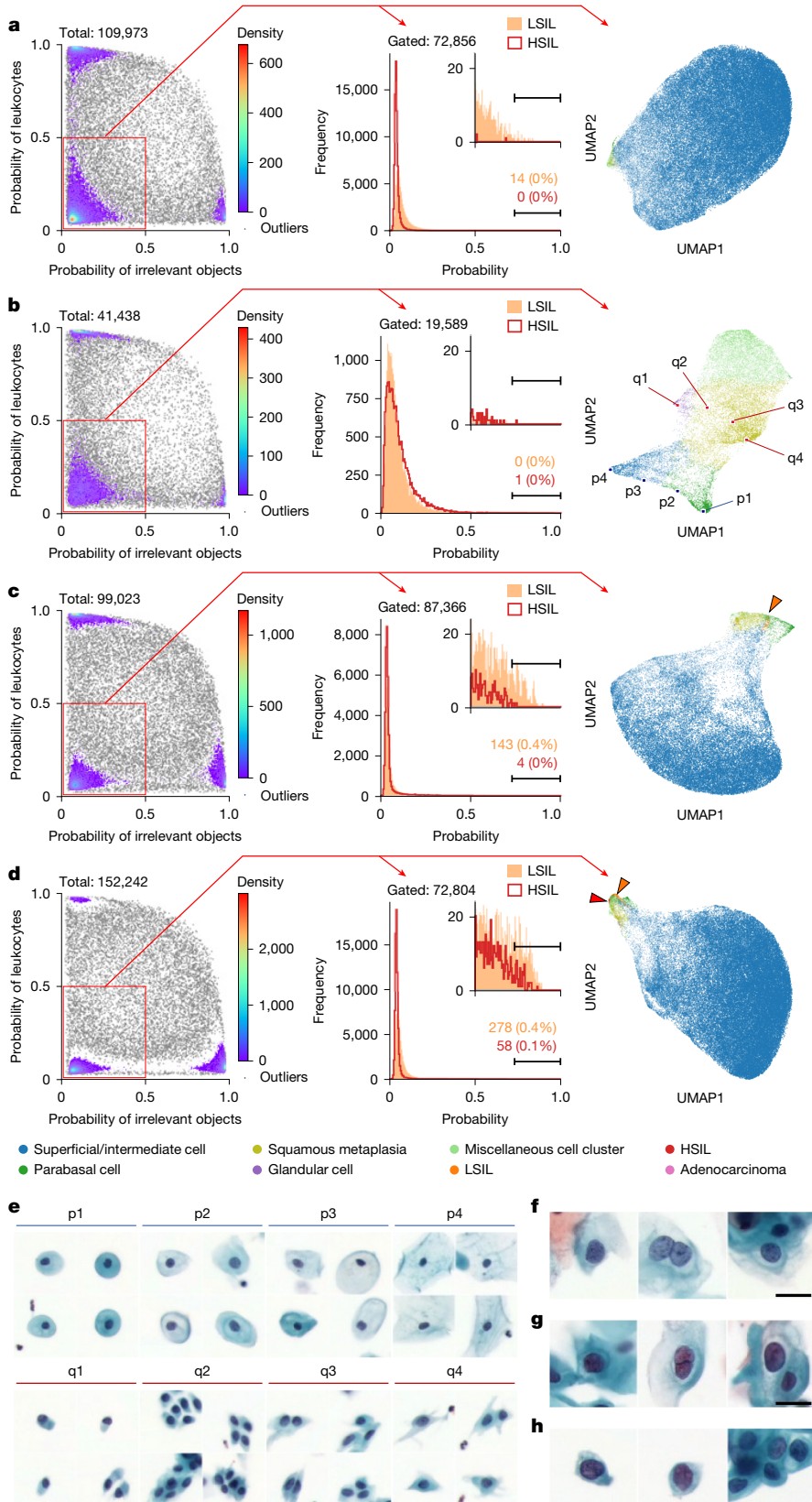

**Fig. 3 | CMD-based cell population analysis. a,b,** CMD-based analysis of cytological samples diagnosed as NILM from a 47-year-old (**a**) and a 62-year-old (**b**) individual. **c,d,** CMD-based analysis of samples diagnosed as LSIL (**c**) and HSIL (**d**). Each panel includes three plots, from left to right: scatter plot for gating out leukocytes and irrelevant objects, overlaid histograms of LSIL and HSIL probability scores and UMAP visualizing the distribution of remaining epithelial cells. **e,** Representative single-cell images illustrating gradual morphological transitions observed in **b**: from parabasal to superficial/intermediate squamous cells (p1–p4) and from glandular to metaplastic cells (q1–q4). **f–h,** Representative images of LSIL cells at the orange arrow in **c** (**f**) and of LSIL and HSIL cells at the orange and red arrows in **d** (**g,h**, respectively). Scale bars, 20 μm.

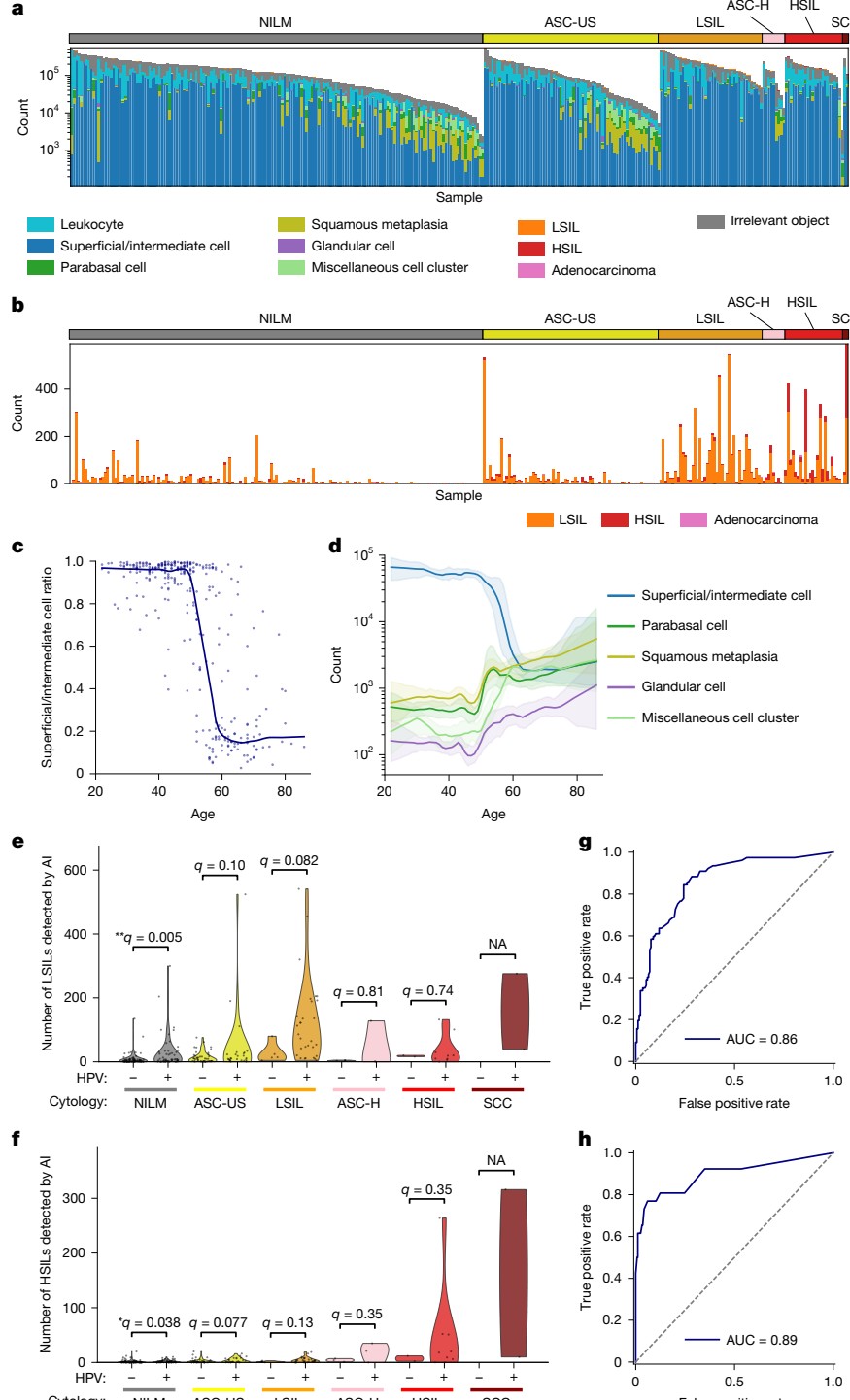

**Fig. 4 | Clinical-grade performance. a,b**, Whole-slide cell counts from 318 cervical liquid-based cytology samples, grouped by cytological diagnosis and sorted within each group by total cell count: counts across all cell types (**a**) and counts limited to abnormal (positive) cell classes (**b**). **c**, Ratio of superficial/intermediate squamous cells as a function of donor age (*n* = 318). Each dot represents one slide; solid lines indicate locally weighted scatter plot smoothing (LOWESS) trends. **d**, log-scaled absolute counts of five epithelial cell types plotted against age across the same slides (*n* = 318; for each subtype, slides with zero counts for that subtype were excluded from the corresponding curve). Solid lines show LOWESS-smoothed estimates of cell counts on the basis of log-transformed data; shaded areas indicate 95% confidence intervals on the basis of bootstrap resampling (100 resamples). **e,f**, AI-detected LSIL (**e**) and HSIL (**f**) cell counts, stratified by cytological diagnosis and HPV status; analyses restricted to slides with available HPV results (*n* = 266). One-sided Mann–Whitney tests (HPV+ > HPV−) within each diagnosis was used. Benjamini–Hochberg-adjusted *q* values are reported with significance indicated as \**q* < 0.05 and \*\**q* < 0.01. The results show small effect sizes in NILM that remain significant (LSIL counts, $\delta$ = 0.31 and *q* = 0.005; HSIL counts, $\delta$ = 0.22 and *q* = 0.038), whereas other classes are not significant (*q* ≥ 0.08). NA, not applicable. **g,h**, Slide-level ROC curves for detecting LSIL+ (**g**) and HSIL+ (**h**) cases using AI-derived cell counts as predictors; AUC values are shown in the legend. LSIL+ (ASC-US excluded; *n* = 246): negatives, NILM; positives, LSIL, ASC-H, HSIL and SCC; predictor, number of LSILs detected by the AI model. HSIL+ (ASC-H excluded; *n* = 309): negatives, NILM, ASC-US and LSIL; positives, HSIL and SCC; predictor, number of HSILs detected by the AI model.

cell count within each group along the vertical axis. Figure 4a reveals that total cell counts per slide vary by up to two orders of magnitude, ranging from several thousand to several hundred thousand. Slides with high cell counts typically have a higher proportion of superficial/intermediate squamous cells, indicative of robust exfoliation of superficial layers, whereas slides with lower cell counts tend to contain a greater abundance of metaplastic and parabasal cells, reflecting variations in sampling depth or patient-specific epithelial remodelling.

Suspecting that this variation in cell composition might relate to donor age, we further investigated the ratio of superficial/intermediate squamous cells by age group (Fig. 4c). This analysis was motivated by the well-established observation that epithelial turnover and differentiation are influenced by hormonal changes occurring during puberty, reproductive age, perimenopause and postmenopause[44], which in turn affect the overall cellular make-up of cervical cytology samples. We also examined the absolute counts of major epithelial cell types, including superficial/intermediate squamous cells, parabasal cells, squamous metaplasia, glandular cells and miscellaneous cell clusters, across different age groups (Fig. 4d). These results confirm that cell population composition changes substantially after the age of 50, with notable increases in the proportions of metaplastic and parabasal cells in older individuals. Such shifts in cellular composition may reflect age-related changes in the cervical epithelium and have important implications for the interpretation of cytological findings.

Figure 4b shows several slides diagnosed as NILM, in which the AI model nonetheless detected large numbers of LSIL cells, suggesting possible underdiagnosis by human review or the presence of subtle HPV-related changes that were not identified by conventional cytology. Conversely, we observed some slides diagnosed as LSIL with only a few positive cells detected by the AI, indicating potential overcalls or limitations in sampling. To evaluate the validity and consistency of these cytology and AI assessments, we compared per-slide abnormal-cell counts between HPV-positive and HPV-negative samples within each cytological category (Fig. 4e,f). In NILM cases, HPV-positive slides exhibited significantly higher counts of both LSIL (**$q$ = 0.005) and HSIL (*$q$ = 0.038) cells, indicating that the model specifically identifies HPV-associated cellular abnormalities rather than generic atypia. HSIL cell counts were elevated in cases categorized as atypical squamous cells, cannot exclude HSIL (ASC-H) and above, further supporting the clinical relevance and potential diagnostic value of the results of the AI model in stratifying cervical disease severity.

To evaluate whether LSIL and HSIL positivity could be determined on the basis of the number of respective abnormal cells, we analysed the proportion of positive cases within each cytological category as a function of the cutoff threshold for LSIL (Extended Data Fig. 6a) and HSIL (Extended Data Fig. 6b) cell counts. In these plots, the horizontal axis represents the cutoff value used to define positivity for LSIL or HSIL cells, whereas the vertical axis indicates the percentage of samples classified as positive (those exceeding the specified threshold) within each cytological category. This approach allows for systematic exploration of the relationship between quantitative cell counts and cytological classification. The results show that LSIL counts were consistently higher in HPV-positive LSIL cases compared with lower-grade samples, and HSIL counts were similarly elevated in HPV-positive HSIL and ASC-H cases relative to lower-grade samples, suggesting a clear separation of cytological severity on the basis of these quantitative cell-count metrics.

To further assess the diagnostic utility of these cell count-based metrics, we performed ROC curve analyses on all samples (Fig. 4g,h). For this analysis, cases with LSIL, ASC-H, HSIL or SCC were defined as positive for low-grade or higher-grade abnormalities (hereafter referred to as LSIL[+]), and cases with HSIL or SCC as positive for high-grade abnormalities (hereafter referred to as HSIL[+]). The sum of LSIL and HSIL cell counts was used as a score to detect LSIL[+] cases, whereas the HSIL count alone was used to detect HSIL[+] cases. The resulting AUC was 0.84 for LSIL[+] detection and 0.89 for HSIL[+] detection, indicating high discriminative performance for both metrics and supporting the potential of these quantitative measures to serve as reliable indicators of disease severity. These findings reinforce the value of these AI-derived cell counts as quantitative, reproducible biomarkers that complement morphology-based interpretation, particularly in borderline cytology cases, such as atypical squamous cells of undetermined significance (ASC-US) or LSIL without HPV testing.

## Multicentre evaluation

Using whole-slide edge tomography and CMD-based analysis, we analysed cervical liquid-based cytology samples collected across four centres ($n$ = 1,124 slides in total): Cancer Institute Hospital of JFCR (C), a leading private cancer-specialty hospital; University of Tsukuba Hospital (T) and Juntendo University Urayasu Hospital (J), two university hospitals; and Kaetsu Comprehensive Health Development Center (K), a community health screening centre. The distribution of cytology classes and HPV status is summarized in Supplementary Table 1. For the multicentre study, we prepared a revised 11-dimensional classification model that explicitly added a navicular cell class, because its glycogen-rich cytoplasm can mimic LSIL morphology and contribute to false positives (Methods and Extended Data Fig. 7). Following this change, false positives decreased in our error analysis. Extended Data Fig. 8 provides a side-by-side comparison of expert-annotated LSIL cells and AI-reported cells, illustrating spatial correspondence. Overall performance showed a modest improvement, most notably at centre K, which serves a comparatively younger screening population where navicular cells are more prevalent (Extended Data Fig. 9a,b).

Applying the 11-class model to all 1,124 slides, we classified every detected cell and summarized per-slide class composition as stacked bar charts (Extended Data Fig. 9c,d), showing consistent trends across centres; slides diagnosed as LSIL or higher contain more LSIL cells, and slides diagnosed as HSIL or SCC contain more HSIL cells. Per-slide counts of LSIL-class and HSIL-class cells, stratified by diagnostic category and centre, are shown as violin plots in Fig. 5a (LSIL) and Fig. 5b (HSIL), demonstrating the same pattern within each centre. Summary statistics by category and centre, together with significance testing versus NILM, are provided in Supplementary Table 2, in which almost all positive categories show significant increases. Similarly, stratification by HPV result (−/+) and centre demonstrates that HPV-positive slides have higher LSIL and HSIL cell counts than HPV-negative slides at every centre, with statistically significant differences (Fig. 5c,d).

We next evaluated the slide-level detection of LSIL[+] and HSIL[+] using ROC curve analysis for each centre (Fig. 5e,f). The resulting AUC values for centres C, T, K and J were 0.91/0.91, 0.90/0.97, 0.86/0.94 and 0.91/0.89 for LSIL[+]/HSIL[+], respectively, indicating consistent performance across centres with mean AUC values around 0.9. Furthermore, because our CMD-based analysis applies a confidence threshold to identify positive cells, we examined how varying this threshold affected AUC values. Across a wide range (0.60–0.90), AUC values were largely stable (Extended Data Fig. 9e,f), indicating robustness to threshold choice. Performance was similarly stable across liquid-based cytology preparation methods, with comparable AUC values for SurePath and ThinPrep samples (Extended Data Fig. 10). Finally, to benchmark AI performance against conventional cytology using HPV positivity as the reference, we overlaid human cytology operating points (ASC-US[+] and LSIL[+]) onto the AI-based ROC curves. Across centres, the AI model outperformed ASC-US[+] triage but fell slightly below LSIL[+] performance, whereas at centre C, in which cytology and HPV testing were performed on the same day, AI performance was comparable to LSIL[+] (Fig. 5g,h).

## Discussion

This study presents a real-time clinical-grade autonomous cytology platform that integrates high-speed and high-resolution whole-slide

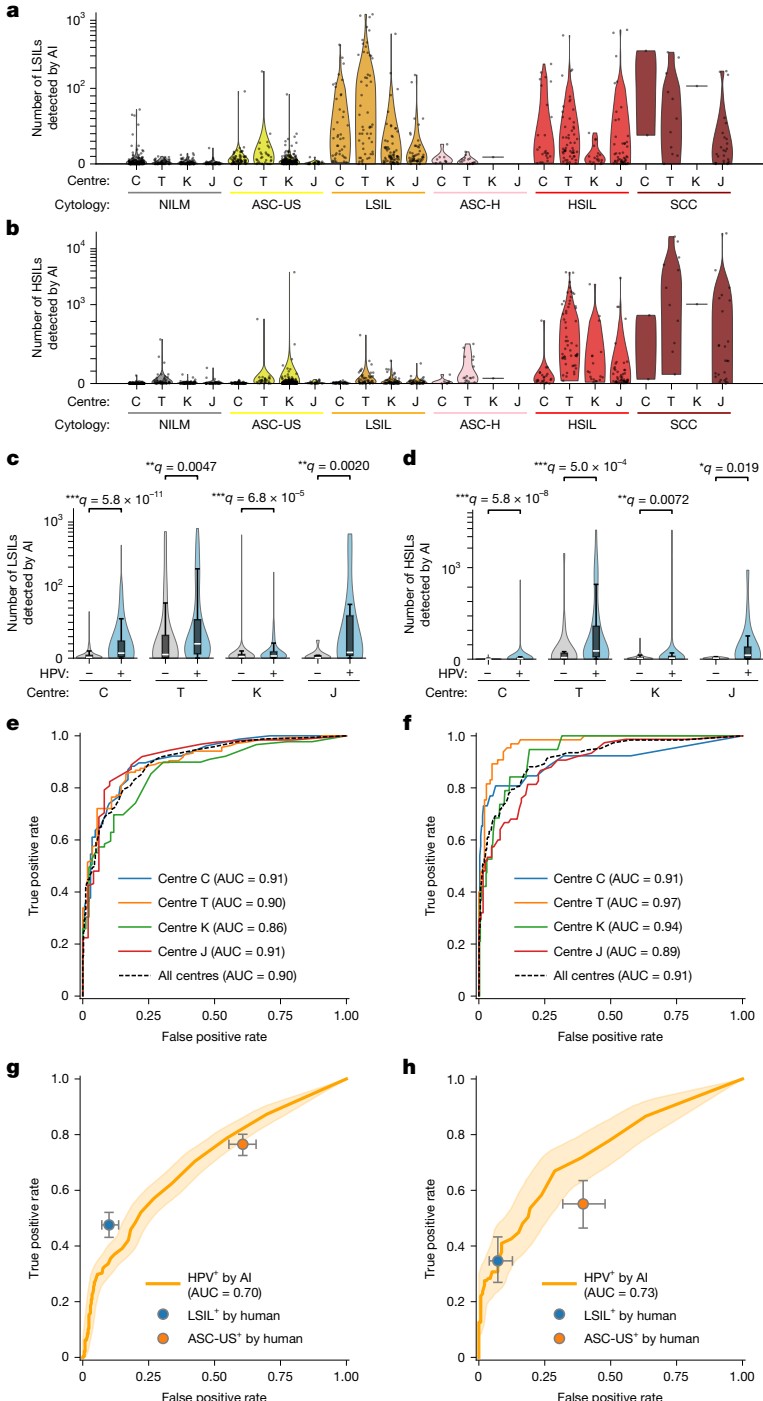

**Fig. 5 | Multicentre evaluation of clinical performance. a,b,** AI-detected LSIL (**a**) and HSIL (**b**) cell counts across four centres (C, T, K and J), stratified by cytological diagnosis. Violin plots show count distributions with individual slides overlaid. **c,d,** AI-detected LSIL (**c**) and HSIL (**d**) cell counts by HPV status (− and +) within each centre. Violin plots include overlaid box plots ($n = 814$; centre line, median; box, Q1–Q3; whiskers, 1.5 × IQR, where IQR is the interquartile range). One-sided Mann–Whitney tests (a priori HPV[+] > HPV[−]) were performed within each centre; Benjamini–Hochberg-adjusted $q$ values are indicated (*$q < 0.05$, **$q < 0.01$ and ***$q < 0.001$). Effect sizes (Cliff's $\delta$; HPV[+] > HPV[−]) were 0.47, 0.32, 0.24 and 0.67 for LSIL and 0.38, 0.43, 0.15 and 0.47 for HSIL at centres C, T, K and J, respectively. In **a–d,** the $y$ axis uses a hybrid linear-log scale to accommodate broad dynamic ranges: linear up to $10^2$ (LSIL) or $5 \times 10^2$ (HSIL); logarithmic above. **e,f,** Slide-level ROC curves for detecting LSIL[+] (**e**; LSIL, ASC-H, HSIL and SCC) and HSIL[+] (**f**; HSIL and SCC), computed separately by centre. C ($n = 318$), T ($n = 222$), K ($n = 385$) and J ($n = 199$), and for all centres

combined ($n = 1,124$), using AI-derived cell counts as predictors. AUC values are listed in the legend. **g,h,** Slide-level ROC curves for AI-based detection of HPV positivity (solid lines, computed from all available slides; shaded bands, 95% confidence bands estimated by stratified bootstrap resampling with 2,000 resamples). Human cytology operating points (ASC-US[+] and LSIL[+]) are overlaid with 95% Wilson confidence intervals for sensitivity and specificity. Pairwise comparisons between AI and the expert cytologist at matched specificity (ΔTPR) and matched sensitivity (ΔFPR) were performed using two-sided bootstrap tests, without multiple comparison adjustment (Methods). **g,** All-centre analysis ($n = 814$ slides; 473 HPV[+] and 341 HPV[−]). AI significantly outperformed ASC-US[+] at matched specificity and sensitivity (both $P = 0.034$) but was slightly inferior to LSIL[+] (both $P < 0.001$). **h,** Centre C only ($n = 266$ slides; 127 HPV[+] and 139 HPV[−]). AI significantly exceeded ASC-US[+] (both $P = 0.002$) and showed no significant difference versus LSIL[+].

optical tomography with edge computing. By combining rapid 3D imaging, on-the-fly data compression and population-scale morphological analysis, the system addresses long-standing challenges in cytological diagnostics: subjectivity, inconsistency and limited scalability. Although high-throughput mesoscale gigapixel imaging systems, including multi-camera array scanners, continue to advance[37], our platform emphasizes a seamless end-to-end clinical workflow, from edge-side compression and secure data transfer to on-site AI inference, enabling real-time operation in routine diagnostic settings. As data handling and system integration for mesoscale imaging mature, such approaches may serve as complementary wide-field front ends to our 3D morphology-aware analytical framework. We have overcome the critical hurdle of digitizing cytology samples in conventional cytological smears and thick cell clusters in liquid-based cytology, especially given their inherent thickness[37–39], by leveraging whole-slide edge tomography with high spatial resolution. In contrast to traditional approaches, such as extended depth-of-field imaging, which often fail to resolve overlapping cells and struggle to capture structural irregularities[45,46], our platform provides detailed 3D reconstructions that retain essential morphological information.

We have demonstrated this capability by successfully digitizing 3D structures of SCC and adenocarcinoma cells in cervical cytology smears. Notably, adenocarcinoma often presents as hyperchromatic crowded groups[47]—densely packed structures that have traditionally posed major challenges for digital imaging owing to limited axial resolution and optical sectioning[16,48–50]. Our system addresses this by enabling high-resolution optical slicing along the $Z$ axis with minimal computational cost, allowing clear visualization of individual nuclei and subnuclear features within tightly clustered cells. This advancement is broadly applicable to other cytological samples from the breast, lung, thyroid, endometrium and salivary glands[16,50], where similarly complex 3D architectures are common. High-fidelity digitization is essential not only for remote diagnostics and educational training but also for validating AI-generated results in clinical workflows.

A central innovation of our platform is the CMD, an interpretable embedding space analogous to cluster of differentiation markers used in flow cytometry. The CMD enables cytologists to explore large-scale morphological landscapes using intuitive tools, such as scatter plots, histograms and UMAPs. Through CMD, we identified clear morphological trajectories reflecting epithelial differentiation and neoplastic transformation, offering insight into transitional states not easily captured by conventional categorical labels. Quantitative validation using silhouette scores and phenotype clustering confirmed that CMD captures biologically meaningful structures aligned with both cytological classification and HPV status.

The CMD-based quantitative cytology framework offers considerable potential for advancing diagnostic precision and screening efficiency. As shown in Fig. 4g,h, the system successfully distinguished LSIL and HSIL cells, respectively, suggesting that CMD-based cytology could serve not only as a supplement to primary screening but also as a triage tool for stratifying HPV-positive individuals. Previous studies have highlighted that human interpretation in cytology can be biased by knowledge of HPV status, potentially affecting diagnostic outcomes[51–53]. By contrast, our AI-driven quantitative approach delivers consistent and objective assessments, thereby reducing such bias and enhancing diagnostic reliability across diverse clinical settings.

In the multicentre evaluation across four clinical sites ($n = 1,124$ slides in total), the 11-class model has demonstrated reproducible performance at the slide level. Site-wise ROC analyses yielded AUC values of 0.86–0.97 for detecting LSIL⁺ and HSIL⁺ (Fig. 5e,f), and these values remained stable over a wide range of positive-cell confidence thresholds (Extended Data Fig. 9e,f), indicating the robustness of the CMD-based pipeline. Consistent with biological expectation, HPV-positive slides showed higher AI-derived counts of LSIL-class and HSIL-class cells than HPV-negative slides at all centres (Fig. 5c,d).

Benchmarking against conventional cytology using HPV positivity as the reference further contextualized performance; across centres, the AI model outperformed the ASC-US⁺ triage, and at one centre where cytology and HPV samples were collected on the same day, AI performance was comparable to the LSIL⁺ triage (Fig. 5g,h). Notably, these results were obtained with a model that explicitly incorporated a navicular cell class; the ability to incorporate a new phenotype with only a modest number of curated examples highlights the extensibility and adaptability of the system for future expansions.

Beyond classification, CMD enables high-dimensional phenotypic analysis. As illustrated in Fig. 3c,d, UMAP trajectories capture gradual morphological transitions, offering insight into spatial and temporal cell-state dynamics, including potential disease progression. Figure 4c,d and Extended Data Fig. 9a further reveal age-associated shifts in epithelial composition, underscoring the utility of CMD for population-level cytological profiling. Notably, Fig. 4e shows that in NILM samples, the AI model identified a higher number of abnormal cells in HPV-positive cases than in HPV-negative ones—cells that may have been overlooked by human screeners. As shown in Fig. 4a,b, these cases often exhibited high overall cell counts and a predominance of superficial/intermediate squamous cells—conditions that can overwhelm manual reviewers and hinder detection of rare abnormalities. The ability of the AI model to detect such subtle patterns highlights a key advantage of CMD-based large-scale single-cell analysis. Collectively, these findings demonstrate the power of CMD-based AI cytology not only as a diagnostic aid but also as a discovery tool for uncovering latent patterns in complex cellular populations.

Although further development and validation are needed, our system presents strong potential to enable a new generation of applications in cytological diagnostics and research. First, the integration of 3D digitization with AI-based analysis could help mitigate the global shortage of cytotechnologists[54–56] by streamlining diagnostic workflows and expanding access to rapid on-site procedures, such as rapid on-site evaluation[57], which are restricted to specialized centres at present. Second, the system can extend cytological diagnostics to rural and underserved regions where trained cytologists are unavailable, thereby improving access to timely and reliable care. Third, by transforming cellular morphology into quantitative and reproducible metrics, this approach can facilitate knowledge transfer, reduce dependence on subjective expertise and accelerate the discovery of new cell types or diagnostic markers. Finally, coupling this system with technologies such as image-activated cell sorting[58,59] could enable real-time isolation of rare or abnormal cells on the basis of subtle morphological features. This would pave the way for downstream molecular analyses, linking cytological imaging directly to genomics and proteomics. Together, these advances point towards a more accessible, autonomous and data-driven future for cytology.

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

## Methods

### Whole-slide edge tomography

As shown in Extended Data Fig. 1, the whole-slide edge tomograph comprises multiple hardware modules optimized for high-speed 3D imaging and edge-side data processing. The illumination system uses a high-power light-emitting diode (XQ-E; Cree) as the light source, paired with a motorized iris (Nihon Seimitsu Sokki) to control the numerical aperture. This illumination passes through the cytology sample and is projected onto a camera board equipped with a CMOS image sensor (IMX531; Sony) and imaging optics. The camera board (e-con Systems) is mounted on a $Z$ stage (Chuo Precision Industrial), which executes precise axial scanning under the control of a real-time controller. The $XY$ stage translates the slide in the lateral plane for complete coverage during image acquisition.

These mechanical components are tightly integrated with the edge computer, which includes several modules: an image sensor FPGA (CertusPro-NX; Lattice Semiconductor), a real-time controller equipped with an extra FPGA (Artrix7; Advanced Micro Devices), an $XY$ stage controller on the basis of a microcontroller (STM32; STMicroelectronics), an illumination controller on the basis of a microcontroller (RL78; Renesas Electronics) and an SOM unit (Jetson Xavier NX; NVIDIA). The SOM features a multicore central processing unit (CPU), a GPU, a hardware encoder and main memory used as an image buffer. An application running on SOM manages internal communications over USB and SPI protocols to coordinate the $XY$ stage, $Z$ stage and illumination modules. Captured images from the CMOS sensor are first transmitted to the FPGA, where real-time high-speed signal conditioning and protocol conversion are performed.

To support high spatial and temporal resolution, the system uses a dual four-lane MIPI, which effectively doubles the data throughput compared with a single MIPI–Camera Serial Interface configuration. This allows continuous transmission of 4,480 × 4,504 resolution images at up to 50 frames per second from the FPGA to the SOM, facilitating reliable real-time handling of large volumetric datasets. Upon receipt by the SOM, image data undergo a three-step processing pipeline: (1) 3D image acquisition; (2) 3D reconstruction through axial alignment using both the GPU and CPU; and (3) real-time compression using the on-board encoder. For compression, the system leverages the NVENC library of NVIDIA to encode 3D image stacks into the HEVC format with hardware acceleration. This process ensures substantial data reduction while maintaining the critical structural features needed for downstream visualization and analysis. The resulting compressed image data are stored locally on a solid-state drive integrated within the edge computer.

From there, compressed image data are transmitted to a back-end server where they are stitched into full-slide 3D volumes and stored on a network-attached storage (NAS) system. These reconstructed volumes are subsequently used for both interactive visualization and AI-based computational analysis. The back-end server hosts a DZI viewer, which enables smooth and responsive visualization by dynamically decompressing and transmitting only the requested tile regions on the basis of user inputs, such as zooming, panning and focus adjustments. These operations are accelerated by a GPU (RTX 4000 Ada; NVIDIA), which handles stitching, image rendering and hardware-accelerated decoding. In parallel, an AI analysis server retrieves the compressed data from the NAS, decodes it using hardware acceleration and performs diagnostic or morphological inference using a high-performance GPU (RTX 6000 Ada; NVIDIA). The resulting predictions and associated metadata are stored back on the NAS for subsequent review or downstream integration.

### Sectional 3D image construction and compression

The imaging workflow involves tightly coordinated real-time interactions among multiple software and hardware components operating in parallel. The real-time controller adjusts the $Z$ stage to sequentially position the slide at specified focal depths, enabling the acquisition of sectional 2D images across various $Z$ planes. Concurrently, each captured image is transmitted to the image signal processing unit of the FPGA, which forwards the data to the GPU buffer on the edge computer. Upon completion of image acquisition at a given region, the $XY$ stage promptly moves the slide to the next imaging section while image processing and compression begin immediately, achieving a pipelined, non-blocking execution flow. A dedicated 3D image construction module processes the acquired $Z$-stack by enhancing colour uniformity and dynamic range and selecting optimal focal planes to ensure that all cells appear sharply focused. In parallel, a 3D image compression module uses the hardware encoder integrated in the SOM to compress the processed image stack into an HEVC-format video file. These modules operate simultaneously, enabling high-throughput scanning without computational bottlenecks.

To evaluate the timing characteristics of individual imaging tasks, time logs were recorded under three $Z$-layer configurations: 10, 20 and 40 layers. Representative task sequences for the first eight imaging sections at the beginning of a whole-slide scan are shown in Extended Data Fig. 2d (10 layers), Extended Data Fig. 2e (20 layers) and Extended Data Fig. 2f (40 layers). These logs delineate task execution for $XY$ stage motion, image acquisition, 3D construction and compression. The first imaging section includes an initialization step and thus takes slightly longer than subsequent areas.

To quantify system performance across the full scan, we calculated the average and standard deviation of the latency of each task per imaging section for each $Z$-layer setting. As summarized in Extended Data Fig. 2i, the $XY$ stage motion time remained constant regardless of $Z$-stack depth, whereas image acquisition, construction and compression durations increased linearly with the number of $Z$ layers. The larger error bars associated with $XY$ stage motion reflect variations in travel distance between imaging sections. These results confirm the predictable and efficient scaling behaviour of the system across varying imaging depths.

### Sectional 3D image compression

Sectional 3D image compression was implemented using the HEVC codec, with optimized parameters defining three selectable modes: high, medium and low, corresponding to target bit rates of 40.36 Mbps, 24.21 Mbps and 8.07 Mbps, respectively. To evaluate compression performance, we assessed both image quality and resulting file size. Image quality was quantified by calculating the PSNR between the original and compressed YUV images. For each compression setting, 3 cytology slides were scanned, with 10 imaging sections per slide and 10 $Z$ layers per area, yielding 300 frames per setting. PSNR values were computed for every frame and visualized as histograms in Extended Data Fig. 2a.

To analyse frame-wise variation in compression artefacts, we further plotted the PSNR values across the $Z$ layers for each slide and imaging section combination (30 lines per setting). These $Z$-layer-wise profiles, shown in Extended Data Fig. 2g, revealed that under low-quality settings, the PSNR values exhibited an alternating pattern between even and odd layers—an effect that diminished at higher bit rates. In many imaging sections under the low setting, PSNR fluctuated around 41–42 dB, whereas medium and high settings showed more stable PSNR levels centred around 42 dB and 43 dB, respectively. We also noted that some imaging sections exhibited higher-than-average PSNR values across all conditions. Upon inspection of the corresponding regions in the original slides, we found that these locations coincided with red ink markings manually applied to the coverslip. Because these markings present simpler and more uniform visual features than the surrounding cellular structures, they probably resulted in less distortion during compression, yielding higher PSNR.

To investigate the correlation between PSNR and visual acceptability for cytological interpretation, we generated compressed image sets

with finely adjusted compression parameters that produced PSNR values ranging from 38 dB to 44 dB (Extended Data Fig. 2c). Visual inspection showed that image degradation was negligible when PSNR exceeded 40 dB and virtually imperceptible above 42 dB. Because the predefined high and medium compression modes consistently yielded PSNR values around 42 dB and above 40 dB, respectively, we concluded that these settings maintain sufficient fidelity for reliable cytological assessment.

We next examined how compression quality settings influence processing time (Extended Data Fig. 2h). Under the ten-layer condition, we compared time distributions across high, medium and low compression modes for each task. As expected, *XY* stage motion, image acquisition and image construction times remained unaffected by compression settings. Image compression time increased slightly with higher-quality settings, but the increase was modest and did not materially impact overall system throughput. These results confirm that high-quality image compression can be achieved without compromising imaging efficiency.

Finally, we evaluated file size and imaging time using three more cytology slides, independently from those used for PSNR analysis. Whole-slide 3D images were acquired under three compression quality levels (high, medium and low) and *Z*-layer counts (10, 20 and 40). The resulting data sizes are shown in Extended Data Fig. 2b. As expected, the file size increased proportionally with the number of *Z* layers and decreased with stronger compression. Imaging time under these same conditions was measured separately and is presented in Extended Data Fig. 2j. These values represent the duration of the pure image acquisition process, excluding slide loading or system preparation time. Imaging time scaled linearly with the number of *Z* layers and was not substantially affected by compression settings.

## Sectional 3D image decompression and viewing

Extended Data Fig. 3a illustrates the architecture of the DZI viewer system, which enables an interactive web-based visualization of whole-slide 3D cytology images. The system is composed of three primary layers: front end, back end and data layer. Following image acquisition, sectional 3D images are transmitted from the imaging system to the back-end server over a network. The back-end software then performs stitching of individual imaging sections using positional metadata, generating the alignment information required to reconstruct the full 3D whole-slide image. Once stitching is completed, the image data are made available for viewing through the DZI interface.

In the front end, users can browse a list of available slides and interact with the whole-slide image using familiar operations, such as zooming, panning, rotation and navigation across *Z* layers. A preview image assists with rapid slide identification, and annotation tools allow users to flag or mark suspicious cells. The ability to scroll through focal planes enables inspection of diagnostically relevant features that may be missed in conventional 2D views.

On the back end, the server responds to front-end requests through two application programming interfaces (APIs): a slide API for metadata and an image API for tile access. When a specific image tile is requested, the back end retrieves the corresponding compressed frame from the data layer and decompresses it using NVDEC, a hardware video decoder integrated into the NVIDIA GPU. The Decord library interfaces with NVDEC, enabling efficient, hardware-accelerated HEVC frame decoding and supporting random access for rapid tile retrieval. The system is capable of handling more than ten concurrent image tile requests in parallel, ensuring responsive, low-latency performance even under high user load.

The performance of the sectional 3D image decompressor was evaluated to assess the ability of the back-end server to respond to image requests from the viewer in real time. A series of tests measured the response time as a function of request frequency and image tile size, quantifying the capability of the system to retrieve and render tomograms efficiently under varying conditions. The evaluation involved benchmarking the time required to retrieve and display tomographic frames. The results, presented in Extended Data Fig. 3b, show that the system maintained an average response time of under 100 ms per image tile request, even when on-the-fly HEVC decompression was required. Integration of NVDEC interfaces substantially reduced decompression latency, whereas the Decord library enabled fast, hardware-accelerated frame extraction. These findings confirm the efficiency and scalability of the system, demonstrating its suitability for high-throughput real-time applications in medical image analysis, in which low latency and responsiveness are critical for clinical utility.

## Detection of cell nuclei

Cell nuclei were detected using a YOLOX object detection model trained on a cytology-specific dataset derived from images acquired by the whole-slide edge tomograph (Supplementary Fig. 1a). A total of 348 images were used, with 278 allocated for training and 70 for validation. Initial nucleus annotations were generated using a semi-automated pipeline on the basis of traditional image processing methods and then manually reviewed and corrected using the Computer Vision Annotation Tool. This process yielded 242,669 annotated nuclei in total: 199,552 for training and 43,117 for validation (Supplementary Fig. 2).

To reduce computational overhead, both training and inference were conducted on downsampled images resized from the original $4,480 \times 4,504$ resolution to $1,024 \times 1,024$ pixels. Model training was performed on an NVIDIA RTX 6000 Ada GPU. Inference performance was evaluated using ROC curve analysis across intersection-over-union thresholds of 0.8, 0.6, 0.4 and 0, yielding AUC values of 0.79, 0.77, 0.70 and 0.63, respectively (Supplementary Fig. 3a). To maximize sensitivity and minimize missed detections during this critical initial stage, we selected an intersection-over-union threshold of 0 and a detection probability cutoff of 0.005.

To assess detection validity, we compared automated nucleus counts to manual counts across the validation dataset. As shown in Supplementary Fig. 3b, the two methods exhibited strong agreement, with a regression line of $y = 1.0098x$ and $R^2 = 0.9487$, indicating high correlation. Outlier cases, in which the model overestimated counts, were further examined (Supplementary Fig. 3c,d). These discrepancies were largely attributed to nonspecific detections near cell boundaries or in background regions. However, such false positives were deemed acceptable because downstream cell classification processes are designed to filter out irrelevant detections. By contrast, false negatives at this stage would result in the exclusion of cells from subsequent analysis, which would be more detrimental to overall performance. Moreover, atypical cells exhibit greater morphological variability than normal cells and are therefore more prone to occasional detection misses. Because these atypical cells represent the primary diagnostic targets, we intentionally biased the detector towards higher sensitivity to ensure broad coverage of abnormal cell populations.

## Extraction of in-focus single-cell images centred on nuclei

To extract morphologically informative single-cell images, we used a multistep pipeline involving nucleus detection, *Z*-layer grouping, focus evaluation and image cropping (Supplementary Fig. 1b). Initial nucleus detection was performed using the YOLOX model on downsampled versions ($1,024 \times 1,024$ pixels) of the original 3D whole-slide images. To balance axial resolution and inference speed, 2D optical sections were subsampled at 3-μm intervals from the *Z*-stack. This approach often resulted in multiple redundant detections of the same nucleus across neighbouring slices owing to partial visibility in adjacent planes.

To resolve this redundancy, we implemented a grouping algorithm that clustered spatially proximate and axially aligned detections across *Z* layers, treating them as a single nucleus instance. For each grouped nucleus, we identified its *Z* range and retrieved full-resolution ($4,480 \times 4,504$ pixels) image patches centred at the nucleus coordinates

but only within the identified $Z$ range. This selective retrieval ensured that focus evaluation was conducted on the relevant high-resolution slices, minimizing unnecessary computation.

A focus evaluation metric was then applied to the extracted $Z$-stack to identify the slice with the best optical focus. The slice with the highest focus score on the basis of criteria such as local contrast or sharpness was selected. From this slice, a 224 × 224 pixel patch centred on the nucleus was cropped, capturing both the nucleus and its surrounding cytoplasmic context. These high-quality nucleus-centred single-cell images served as inputs for downstream classification models.

### Classification of single-cell images using MaxViT

Single-cell images extracted from the focused optical section of each nucleus-centred region were classified using a vision transformer model on the basis of the MaxViT-base architecture[43]. The model was trained to differentiate among ten cytological categories: leukocytes, superficial/intermediate squamous cells, parabasal cells, squamous metaplasia cells, glandular cells, miscellaneous cell clusters, LSIL cells, HSIL cells, adenocarcinoma cells and irrelevant objects (such as debris, non-cellular material and defocused images). The training and validation datasets were constructed from expert-annotated cell images derived from 354 donor-derived whole-slide images. The numbers of annotated cells used for training and validation in each class were as follows: 18,219/5,281 leukocytes, 23,158/7,557 superficial/intermediate squamous cells, 4,296/1,243 parabasal cells, 2,056/487 squamous metaplastic cells, 836/105 glandular cells, 5,387/994 miscellaneous cell clusters, 1,846/936 LSIL cells, 1,433/262 HSIL cells, 912/420 adenocarcinoma cells and 14,752/5,115 irrelevant objects. All annotations were performed by professional cytologists. To improve generalization performance, the dataset was augmented using random rotations and horizontal/vertical flipping. This resulted in a total of 168,569 training images and 50,222 validation images. Representative examples of the training images are shown in Supplementary Fig. 4. Model training was performed using an NVIDIA RTX 6000 Ada GPU.

For the clinical study across multiple centres, we performed a second round of training using 14 more whole-slide samples from centre K. The first round used samples only from centre C, whereas the second round expanded the taxonomy by introducing a new category, navicular cells, defined as squamous cells showing yellow cytoplasmic glycogen (Extended Data Fig. 7a). Expert-annotated single-cell images were added with the following class counts (training/validation): leukocytes, 2,017/805; superficial or intermediate squamous cells, 4,804/2,130; parabasal cells, 57/214; squamous metaplasia cells, 523/504; glandular cells, 767/166; miscellaneous cell clusters, 527/411; LSIL cells, 100/13; HSIL cells, 84/16; irrelevant objects, 2,064/846; and navicular cells, 267/52 (total 11,210/5,157). The same data augmentation pipeline as in the first round (random rotations and horizontal/vertical flipping) was applied, scaling these datasets sevenfold, to produce effective 78,470 training and 36,099 validation images. Training procedures and hardware were identical to those used in the first round. Validation performance was summarized separately for the original and newly curated validation sets (Extended Data Fig. 7b,c).

### CMD-based cell population analysis

For each classified cell image, the vision transformer model outputs a 10-dimensional or 11-dimensional vector of class probabilities, referred to as the CMD values. These CMD vectors are computed by applying a sigmoid function to the final layer's logits of the MaxViT model, converting them into values between 0 and 1 for each of the ten cytological classes. Each value represents the confidence level of the model for assigning a cell to a given class. Representative CMD outputs are visualized alongside corresponding cell images in Extended Data Fig. 5c–k.

To analyse cell populations at the whole-slide level, we performed a series of visualization and gating-based analyses using the CMD vectors of all detected cells (Fig. 3a–d). The analysis began with a 2D

scatter plot using the CMD probabilities for the 'irrelevant objects' and 'leukocytes' classes. This enabled negative gating to exclude non-cellular, defocused or leukocyte-dominated regions and to isolate epithelial-lineage cells. Within this gated population, we generated histograms of CMD values for the LSIL, HSIL and adenocarcinoma classes to evaluate lesion-associated probability distributions. For each histogram, class-specific thresholds were applied to identify cells with high classification confidence, allowing sensitivity tuning for detecting abnormal populations.

Before UMAP visualization, each cell was assigned a class label using a hierarchical rule-based decision process. Cells that fell beyond the irrelevant-object threshold in the scatter plot were first labelled as 'irrelevant'. Of the remaining cells, those exceeding the leukocyte threshold were labelled as 'leukocytes'. Among the rest, any cell that crossed a predefined gate in the LSIL, HSIL or adenocarcinoma histograms was labelled accordingly. Cells that did not meet any lesion threshold were assigned to the class with the highest CMD value among the five remaining categories: superficial/intermediate squamous cells, squamous metaplasia, parabasal cells, glandular cells and miscellaneous clusters.

These class assignments were then used to colour-code cells in UMAP plots, with each point representing a single cell. Alpha transparency was modulated to reflect prediction confidence, enhancing visual interpretability. The resulting UMAP embeddings and class labels served as the basis for all subsequent whole-slide cytological analyses.

### Human participants

We analysed cervical cytology samples collected at the Cancer Institute Hospital of JFCR (C) and at three more centres: University of Tsukuba Hospital (T), Kaetsu Comprehensive Health Development Center (K) and Juntendo University Urayasu Hospital (J). At centre C, 770 samples (766 liquid-based cytology and 4 conventional smears) were obtained from patients undergoing routine cervical screening between 2011 and 2019. Of these, 452 samples contributed to model development (training/validation), and 318 samples were held out for clinical performance test (Supplementary Table 1). Separately, at centre C, we also obtained two non-cervical conventional smears from routine diagnostic cases, and these were used solely as imaging exemplars. From the other centres, 222 (T), 384 (K) and 199 (J) samples were included for evaluation (Supplementary Table 1). In addition, 14 samples from centre K were used to augment training. Each participating centre obtained institutional approval from its institutional ethics committee: the Medical Research Ethics Review Committee at the Cancer Institute Hospital of JFCR (Institutional Review Board No. 2019-GA-1190, covering centres C and K), the Clinical Research Ethics Review Committee at the University of Tsukuba Hospital (R07-175) and the Research Ethics Committee of the Faculty of Health Science at Juntendo University (2025-016). All procedures were conducted in accordance with the Declaration of Helsinki and all relevant institutional and national guidelines and regulations. Informed consent was obtained through the opt-out process of each institution, in accordance with local policy.

### Sample preparation and evaluation

At the Cancer Institute Hospital of JFCR, cervical cells were collected using a Bloom-type brush (J fit-Brush; Muto Pure Chemicals). Liquid-based cytology samples were prepared using one of the following methods: ThinPrep (Hologic) or SurePath (Becton, Dickinson and Company), in accordance with the manufacturer's instructions. The samples were subsequently stained using the Papanicolaou method and evaluated by cytotechnologists on the basis of the Bethesda System for Reporting Cervical Cytology[60]. In parallel with cytological examination, HPV testing was conducted using the Hybrid Capture 2 assay (QIAGEN). The same ThinPrep or SurePath sample was used for this test. The Hybrid Capture 2 test was carried out strictly according to the manufacturer's protocol. At the other participating centres (T, K and J), sample collection, liquid-based cytology sample preparation, staining,

cytologic evaluation and HPV testing were performed according to each centre's routine clinical protocols. The cytology categories in this study were NILM, ASC-US, LSIL, ASC-H and HSIL (Bethesda terminology); SCC appears where applicable in descriptive summaries. Supplementary Table 1 summarizes the sample counts by centre and HPV status (−, + and 'N/A' for missing HPV results) across cytology categories, with a total column for each row.

## Comparison of sample preparation methods

To assess potential influence of sample preparation methods, we compared SurePath and ThinPrep slides with comparable age distributions (Extended Data Fig. 10a). Both preparation methods yielded similarly favourable ROC curves and AUC values (Extended Data Fig. 10b,c), demonstrating that model performance is robust to slide preparation.

## Procedure for clinical-grade performance evaluation

For the clinical-grade performance analysis, the cervical liquid-based cytology samples from 318 donors were analysed. Expert cytological diagnoses and HPV test results had been obtained in advance, as described elsewhere. Whole-slide image acquisition and CMD-based classification were conducted using previously described methods, and the same classification gates defined in Fig. 3a–d were applied in this analysis. AI-based classification results were aggregated on a per-slide basis to compute the number of detected objects in each class. Figure 4a presents the total cell counts across the ten classes, including epithelial cell types and irrelevant objects, whereas Fig. 4b focuses specifically on abnormal cell classes. In both figures, samples are grouped by cytological diagnosis and sorted within each group according to the total cell count.

To investigate age-related variation in epithelial composition, the ratio of superficial/intermediate squamous cells was calculated for samples diagnosed as NILM and plotted against donor age (Fig. 4c). Additionally, absolute counts of five cytologically normal epithelial components (superficial/intermediate squamous cells, parabasal cells, squamous metaplastic cells, glandular cells and miscellaneous clusters) were calculated and plotted as a function of donor age on a logarithmic scale, with LOWESS smoothing applied. The shaded regions in Fig. 4d represent 95% confidence intervals estimated through bootstrap resampling ($n = 100$).

The number of LSIL and HSIL cells detected by the AI model for each slide was visualized using violin plots (Fig. 4e,f), with samples grouped by cytological diagnosis and HPV test results. Within each cytological category, we compared AI-detected abnormal-cell counts between HPV-negative and HPV-positive slides using a one-sided Mann–Whitney $U$-test (alternative hypothesis: HPV$^+$ > HPV$^-$). Effect sizes (Cliff's $\delta$) were also computed (see 'Statistical analysis' section). Benjamini–Hochberg-adjusted $q$ values are reported, with significance indicated as *$q < 0.05$, **$q < 0.01$ and ***$q < 0.001$. Analyses in Fig. 4e,f and Extended Data Fig. 6 were restricted to cases with available HPV results ($n = 266$ of 318 test slides). To assess the relationship between abnormal cell counts and diagnostic categories, we calculated the proportion of slides that exceeded various cutoff thresholds for LSIL$^+$ and HSIL$^+$ classification within each cytological group (Extended Data Fig. 6). LSIL$^+$ was defined as cases diagnosed as LSIL, ASC-H, HSIL or SCC, whereas HSIL$^+$ was defined as cases diagnosed as HSIL or SCC. ROC analysis was performed on all samples ($n = 318$) using the same diagnostic criteria. The total count of LSIL and HSIL cells was used for LSIL$^+$ detection, and the HSIL count alone was used for HSIL$^+$ detection. The resulting AUC values are shown in Fig. 4g,h.

## Multicentre evaluation

We extended the clinical performance study conducted at the Cancer Institute Hospital of JFCR (C) by prospectively acquiring more cervical liquid-based cytology slides from three other centres: University of Tsukuba Hospital (T), Kaetsu Comprehensive Health Development Center (K) and Juntendo University Urayasu Hospital (J) (Supplementary Table 1). Age distributions varied across centres (Extended Data Fig. 9b). The median ages (IQR) were 49 (41–59) for C, 42 (35–53) for T, 40 (30–48) for K and 42 (34–55) for J.

All slides were imaged using our whole-slide edge tomograph at high image-quality settings, acquiring 40 $Z$ layers per field, and subsequently processed using the 11-class detector–classifier that included the navicular cell class. This extra class was introduced after observing frequent navicular cell morphology among false-positive predictions relative to local diagnoses at centre K. Per-slide cell counts were computed using the previously described CMD-based population analysis, with the classification probability threshold for positive classes (LSIL and HSIL) set to 0.80 unless otherwise specified.

Across the four centres, we analysed 1,124 whole-slide images and summarized per-slide cell burdens at scale (Extended Data Fig. 9b–d). Age-dependent trends for six epithelial cell types (including navicular cells) were visualized with confidence bands. Navicular cell counts peaked among donors in their early 20s, consistent with reports linking these cells to pregnancy and progestin exposure, providing a plausible explanation for the initially lower performance at centre K, whose donor population is younger than those of the cancer-specialty and university hospital cohorts (Extended Data Fig. 9a,b). To assess cross-centre variability, we visualized the distributions of whole-slide cell counts for all classes, as well as for abnormal-only classes, using a hybrid linear–logarithmic scale to accommodate their wide dynamic range (Extended Data Fig. 9c,d). Within each centre, AI-detected LSIL and HSIL counts were summarized by cytological diagnosis (NILM, ASC-US, LSIL, ASC-H, HSIL and SCC) using violin plots with individual slides overlaid (Fig. 5a,b). Complementary summary statistics and within-centre significance testing (NILM versus comparators; Cliff's $\delta$, positive when comparator is greater than NILM; Benjamini–Hochberg-adjusted $q$ values from one-sided Mann–Whitney tests) are reported in Supplementary Table 2.

For HPV-stratified analyses, we used the subset of slides with the available HPV results ($n = 814$). Counts were compared between HPV$^-$ and HPV$^+$ slides within each centre using one-sided Mann–Whitney $U$-tests under the a priori hypothesis HPV$^+$ > HPV$^-$ for both LSIL and HSIL counts, and Benjamini–Hochberg-adjusted $q$ values were reported (Fig. 5c,d). Effect sizes (Cliff's $\delta$) were also computed and reported (see 'Statistical analysis' section). To benchmark AI performance against conventional cytology using HPV positivity as the reference, we computed AI-based ROC curves using slide-level AI scores (Fig. 5g,h). Human operating points were defined from routine cytology as LSIL$^+$ (LSIL, ASC-H, HSIL and SCC) and ASC-US$^+$ (ASC-US, LSIL, ASC-H, HSIL and SCC), and plotted as single points with 95% Wilson confidence intervals for sensitivity and specificity. ROC AUC values and true-positive-rate confidence bands were estimated through stratified bootstrap (2,000 resamples) with linear interpolation on a uniform false-positive-rate grid; 95% confidence intervals were taken as percentile intervals. Pairwise comparisons between AI and human performance were conducted under two matchings: (1) matched specificity, testing $\Delta TPR = TPR_{AI} − TPR_{human}$; and (2) matched sensitivity, testing $\Delta FPR = FPR_{AI} − FPR_{human}$, where TPR and FPR stand for true positive rate and false positive rate, respectively. Uncertainty was obtained using the same bootstrap procedure, and two-sided $P$ values were calculated as $2 \times$ the smaller tail probability of the bootstrap distribution. Centre-specific analyses (for example, centre C) followed the same protocol.

Centre-wise ROC curves were then constructed (Fig. 5e,f). For the LSIL$^+$ end point, ASC-US was excluded. The negative class comprised NILM, and the positive class comprised {LSIL, ASC-H, HSIL, SCC}. For the HSIL$^+$ end point, ASC-H was excluded. The negative class comprised {NILM, ASC-US, LSIL}, and the positive class comprised {HSIL, SCC}. Predictors were whole-slide AI-derived counts from the CMD-based analysis. The sum of LSIL and HSIL counts was used for LSIL$^+$ detection, and the HSIL count alone was used for HSIL$^+$ detection. AUC values were

reported separately for centres C, T, K and J. Threshold sensitivity was assessed by sweeping the per-cell probability threshold from 0.60 to 0.99, with stratified bootstrap 95% confidence intervals (Extended Data Fig. 9e,f).

Finally, to evaluate spatial correspondence between AI detections and expert annotations, cytotechnologists at centre T manually annotated LSIL cell locations on two slides. AI-detected LSILs (per-cell probability greater than or equal to 0.80) were overlaid on the corresponding whole-slide images. The magnified regions confirmed strong spatial co-localization, with cells marked by experts also detected by the AI model (Extended Data Fig. 8).

## Statistical analysis

Group comparisons (diagnostic classes versus NILM, or HPV$^+$ versus HPV$^-$) used one-sided Mann–Whitney $U$-tests under pre-specified alternatives (abnormal > NILM; HPV$^+$ > HPV$^-$). $P$ values were adjusted using the Benjamini–Hochberg procedure within each analysis family (across comparator classes for a given metric in a given figure or table) and were reported as Benjamini–Hochberg $q$ values. Effect sizes were reported as Cliff's $\delta$ (primary), computed from the Mann–Whitney statistic $U_{ref}$ that counts reference-group wins: $\delta = 2(1 - U_{ref} n_{ref}^{-1} n_{comp}^{-1})$, where the convention $\delta > 0$ indicates that the comparator is greater than the reference. Implementation details and scripts are available (see 'Code availability' section). No statistical methods were used to predetermine sample size. No randomization was performed. Cytology assessment was blinded to HPV results and AI outputs; AI analyses were performed without access to cytology or HPV labels (used only for evaluation).

## Software implementation

Sectional 3D image construction and compression on the edge computer were implemented in C++ and CUDA on NVIDIA GPUs using in-house developed software. Sectional 3D image decompression for viewing, deep learning-based cell detection and classification, CMD-based cell population analysis and statistical analysis were implemented in Python (v.3.10 and v.3.12), with several open-source libraries, including NumPy, pandas, matplotlib, seaborn, scikit-learn, statsmodels, PyTorch, torchvision, albumentations, OpenCV, timm and ONNX Runtime. Deep learning models were developed in PyTorch/timm and exported to ONNX for GPU-accelerated inference with ONNX Runtime. Image annotations were created using the open-source software Computer Vision Annotation Tool (v.2.7.6).

## Inclusion and ethics

This study followed ethical guidelines, with informed consent obtained for all samples and protocols approved by institutional ethics committees. Data were analysed with awareness of potential biases. We are committed to promoting equity and inclusion in research while advancing scientific understanding.

## Reporting summary

Further information on research design is available in the Nature Portfolio Reporting Summary linked to this article.

## Data availability

Anonymized CSV files sufficient to reproduce the quantitative figures and tables are publicly available at Zenodo (https://doi.org/10.5281/zenodo.17808303)[61]. These CSV files contain derived measurements and per-slide metadata (including centre, sample preparation method, cytology diagnosis, age and HPV test results where available) but do not include raw images or directly identifiable information. The cytology datasets analysed in this study are securely maintained by CYBO to safeguard patient privacy and proprietary imaging data. Owing to ethical and regulatory constraints, these datasets are not publicly available. Academic investigators with no relevant conflicts of interest may request controlled access to selected de-identified cytological features for non-commercial, research-only purposes. Requests will be reviewed by CYBO in consultation with the sample-providing centres and their institutional review boards or ethics committees and, if approved, will require a data-use agreement that prohibits re-identification and any redistribution of the data. Requests should be directed to N.N. at nitta@cybo.co.jp, and eligible requests will receive a response within 1 month. Source data are provided with this paper.

## Code availability

Custom code developed for this study that is central to the main findings, including modules for edge-device image processing, AI model training, back-end infrastructure and AI inference pipeline, together with the source code for downstream data analysis and figure generation, is publicly available at Zenodo (https://doi.org/10.5281/zenodo.17808303)[61] under the OSI-approved GNU AGPLv3 license.

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

**Acknowledgements** This work was supported by funding from the Tokyo Metropolitan Small and Medium Enterprise Support Center, the Japan Agency for Medical Research and Development (AMED; grant no. JP20he102202), the Ichimura Foundation for New Technology and Koto City. We also acknowledge support from the AMDAP of Tokyo Prefecture, Tokyo Metropolitan Industrial Technology Research Institute, Nihon Seimitsu Sokki, NVIDIA Inception, Microsoft for Startups, AWS Startups, Google for Startups and Serendipity Lab. We are grateful to Incubate Fund and JAFCO for their financial support. We also thank A. Mishima, T. Sekiya, M. Oikawa, S. Aida, E. Saitoh, K. Ueda, M. Otani, N. Furuta, J. Fujiyama and S. Yamazaki for their valuable assistance.

**Author contributions** N.N. and T.S. conceptualized the whole-slide edge tomograph. N.N. and Y. Mase designed and constructed the optical and mechanical hardware. T.S. developed and built the electrical system. T.S., M.H. and J.Z. designed, implemented and evaluated the software for the edge computer, including image compression. N.N., T.S., A.J., P.K. and N.M.Z.A. developed, implemented and assessed the software for image decompression, stitching and real-time tomogram review. N.N., T.S. and R.U.I. performed data preparation and AI model training. They also designed, implemented and evaluated the software for nucleus detection, focusing, classification and population analysis. Y.S., T.I., K.I., H.A. and T.C. prepared and reviewed the cytology samples. Y.S., T.I., K.I., H.A., S.I., N.H. and Y. Murata annotated the cytology data. N.N., Y.S., T.S., H.K. and T.C. planned and conducted the pilot clinical study. N.N., Y.K., Y.H., H.O., T.C. and K.G. designed the multicentre clinical study. Y. Murata, A.S., Y.K., D.M., Y.H., H.N., A.H., M.U., H.O. and T.C. prepared the samples and curated data for the multicentre clinical study. N.N., T.S., N.H., M.K. and N.M.Z.A. performed the data acquisition and analysis for the multicentre clinical study. N.N., Y.S., T.S., T.C. and K.G. supervised the overall work. N.N. and K.G. supervised and validated the scientific aspects of the whole-slide edge tomograph. N.N., Y.S., T.S., F.C.S., R.Y.O., T.C. and K.G. contributed to writing the paper, with support from Y.L. and T.D. in preparing the figures and supplementary materials. All authors were involved in editing the paper.

**Competing interests** N.N., T.S. and K.G. are shareholders of CYBO and inventors on patents covering the data analysis and display method (JP6753622, US12136148B2 and their corresponding family patents and patent applications). N.N. and T.S. are inventors on patents and patent applications covering the 3D image compression and processing techniques (JP7126737 and its family patent applications). N.N. is an inventor on patent applications related to the AI pipeline. The other authors declare no competing interests.

**Additional information**
**Correspondence and requests for materials** should be addressed to Nao Nitta, Tomohiro Chiba or Keisuke Goda.

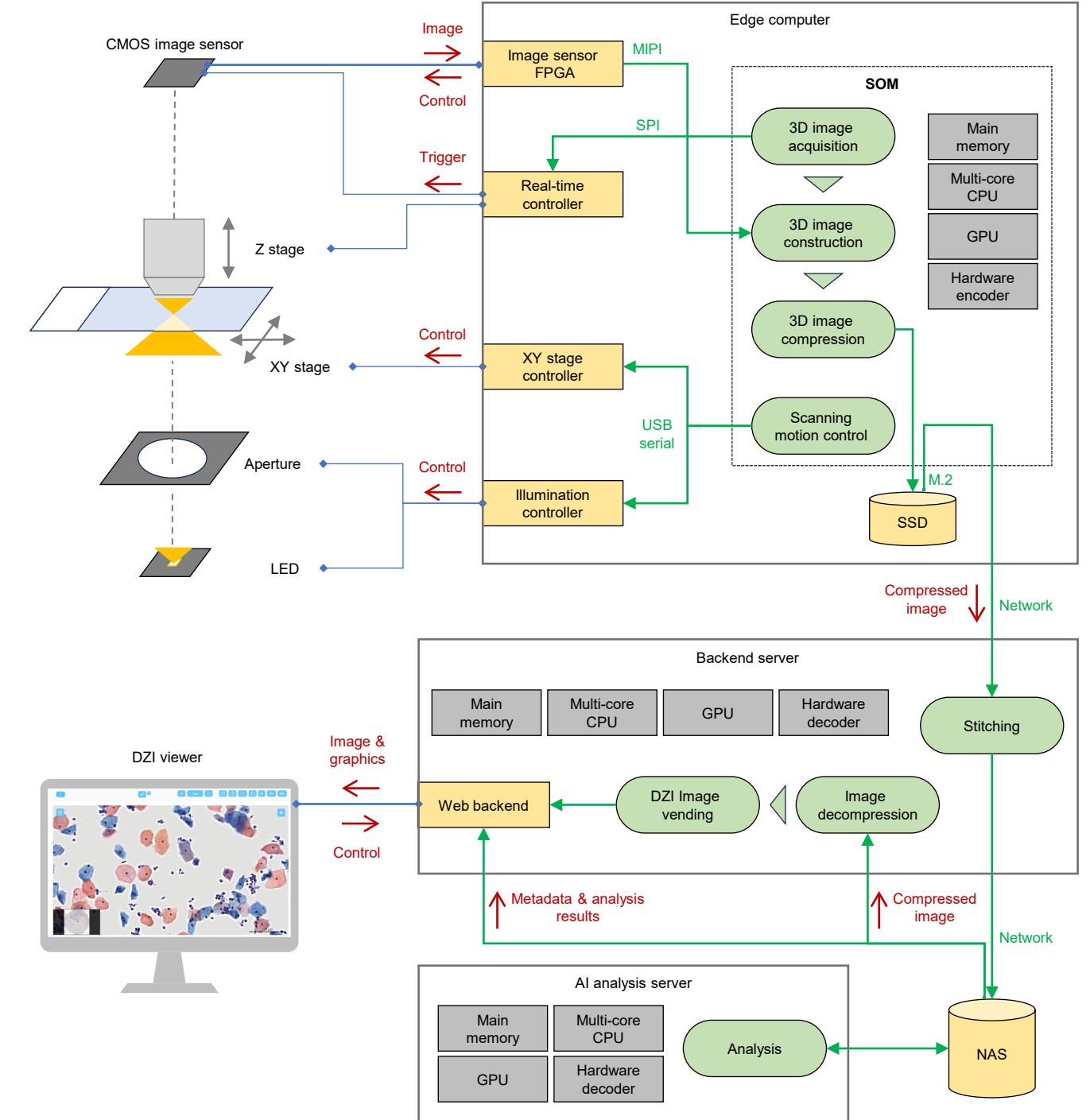

**Extended Data Fig. 1 | Architecture of the whole-slide edge tomograph.**
The system consists of an integrated hardware-software platform that enables high-speed 3D image acquisition, reconstruction, and compression directly on an SOM. The SOM coordinates with the image sensor, XY stage, and illumination controller to perform synchronized scanning and 3D imaging, with real-time processing and compression executed via its onboard GPU and hardware encoder. Scanning motion is interleaved with image acquisition to ensure full-slide coverage. Compressed image data are transmitted over the network to a backend server, where stitching is performed and results are archived in NAS. An AI analysis server retrieves the compressed data, performs decoding and inference, and stores outputs alongside image metadata. The backend also supports a DZI viewer that dynamically decompresses and streams image tiles in response to user actions such as zooming, panning, and Z-plane navigation, enabling low-latency, interactive visualization without requiring full-volume decompression.

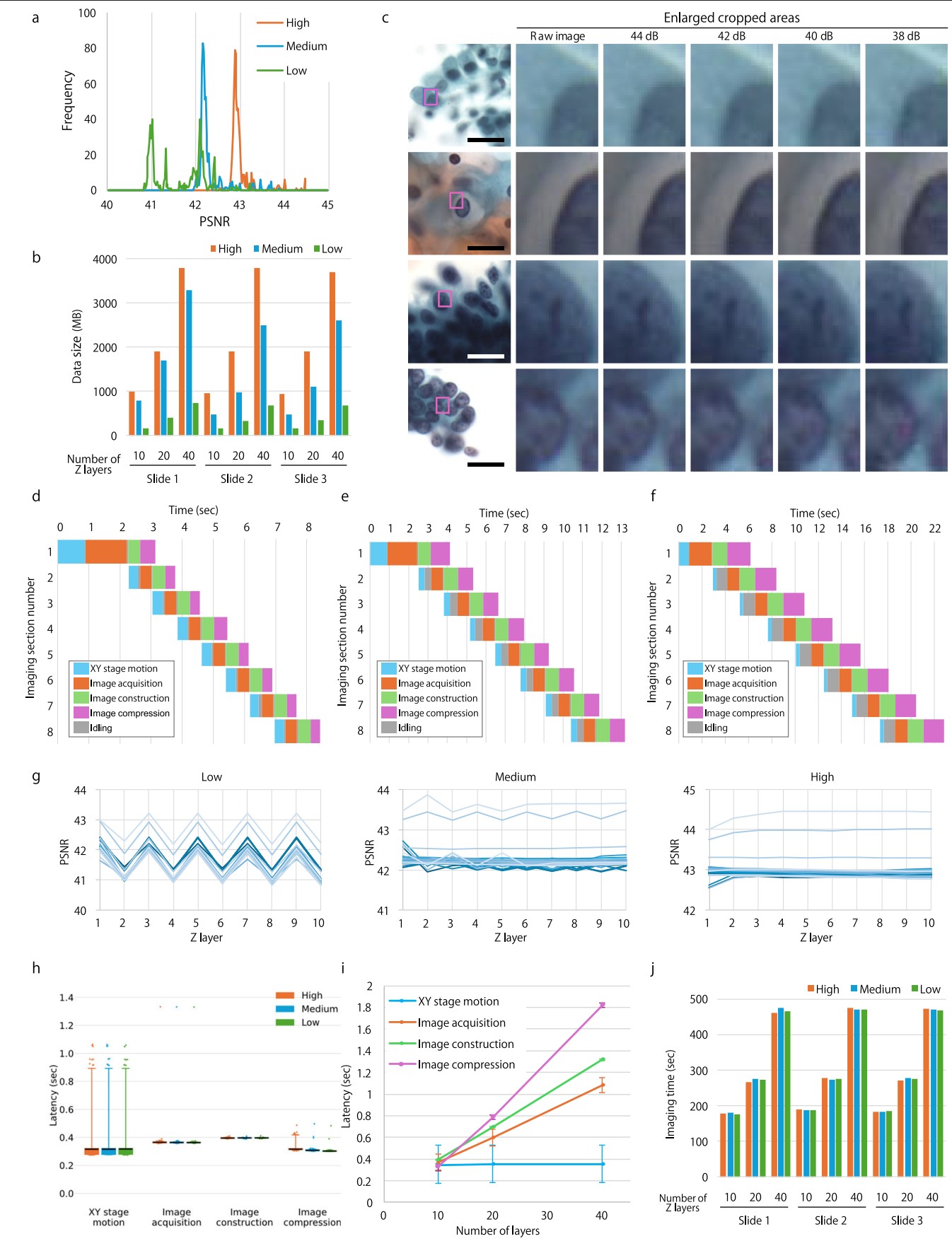

**Extended Data Fig. 2 |** See next page for caption.

**Extended Data Fig. 2 | System performance. a**, Histograms of PSNR values obtained from 300 tomographic images acquired from three liquid-based cytology slides under high, medium, and low HEVC compression settings (3 slides × 10 imaging sections × 10 Z-layers per slide). **b**, File sizes of whole-slide 3D image datasets under varying compression levels and Z-layer counts (10, 20, 40). **c**, Representative tomograms of glandular, LSIL, HSIL, and adenocarcinoma cells compressed to yield PSNR values of 38, 40, 42, and 44 dB. At 38 dB, compression artifacts are apparent. Scale bars: 200μm. **d-f**, Time logs of the imaging process from the first to the tenth imaging section under 10-layer (**d**), 20-layer (**e**), and 40-layer (**f**) Z-stack conditions, with each task colour-coded. **g**, Line plots of PSNR values across Z-layers for each of the 30 combinations of slide and imaging section (3 slides × 10 imaging sections), based on the same 300 images analyzed in Extended Data Fig. 2a. Each line represents one imaging section. Under low compression conditions, a distinct alternating pattern in PSNR values between even and odd Z-layers is observed, indicating non-uniform compression effects across depth. **h**, Latency per imaging section for each imaging task measured under high, medium, and low HEVC compression settings with 10 Z-layers. Boxes show the median and interquartile range; whiskers indicate the 5th–95th percentiles across imaging sections ($n$ = 163 sections per condition). **i**, Average latency per imaging section for 10, 20, and 40 Z-layers, segmented by task (XY stage motion, image acquisition, image construction, and compression). XY motion time remains constant, while other tasks scale linearly with Z-layer count. Error bars represent standard deviation ($n$ = 163 sections per condition). **j**, Durations for whole-slide 3D image acquisitions across varying numbers of Z-layers (10, 20, 40) and compression settings (high, medium, low). Reported values represent net acquisition time, excluding preparatory steps such as slide loading and system initialization.

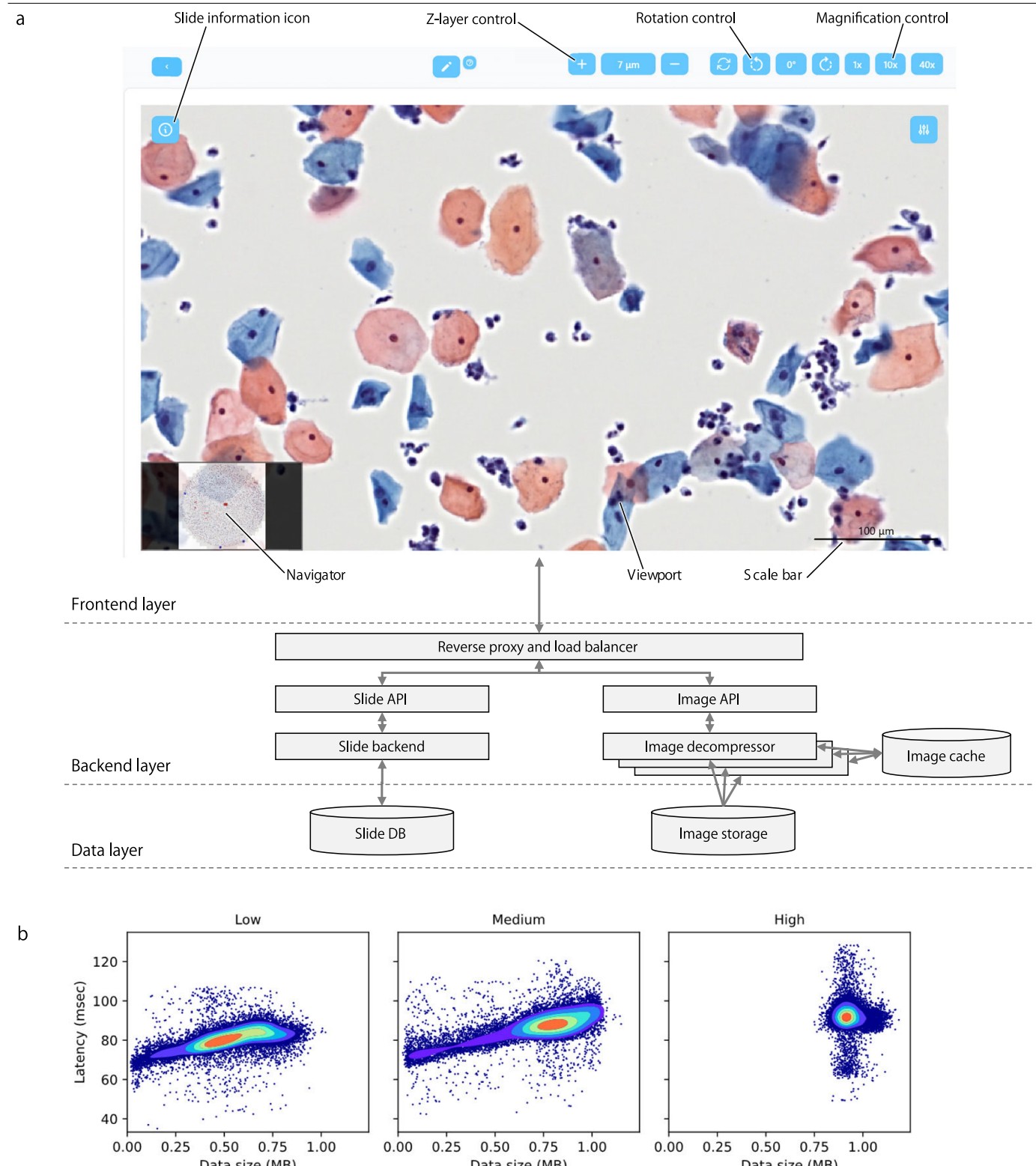

**Extended Data Fig. 3 | Architecture and performance of the DZI viewer.**
**a**, User interface and system architecture of the DZI viewer. The frontend
displays whole-slide tomographic images with associated sample metadata
and supports interactive operations such as zooming, XY panning, and Z-layer
navigation. The backend processes client-side requests, retrieves image tiles
and metadata from compressed storage, performs hardware-accelerated or
GPU-based decompression, and serves the data via the image API. The system
architecture is modularly organized into frontend, backend, and data layers.
**b**, Scatter and density contour plots illustrating frontend request latency for
DZI image tiles at the highest resolution level. The dataset includes two slides
acquired under three image quality settings (low, medium, high) and three
Z-layer configurations (10, 20, 40). Under low and medium compression, tile
sizes vary substantially and exhibit a positive correlation with latency. In
contrast, high-quality settings yield more uniform and generally larger tile
sizes. Despite the computational overhead of HEVC decompression, the
majority of requests are fulfilled within 100 msec, demonstrating real-time
responsiveness of the system.

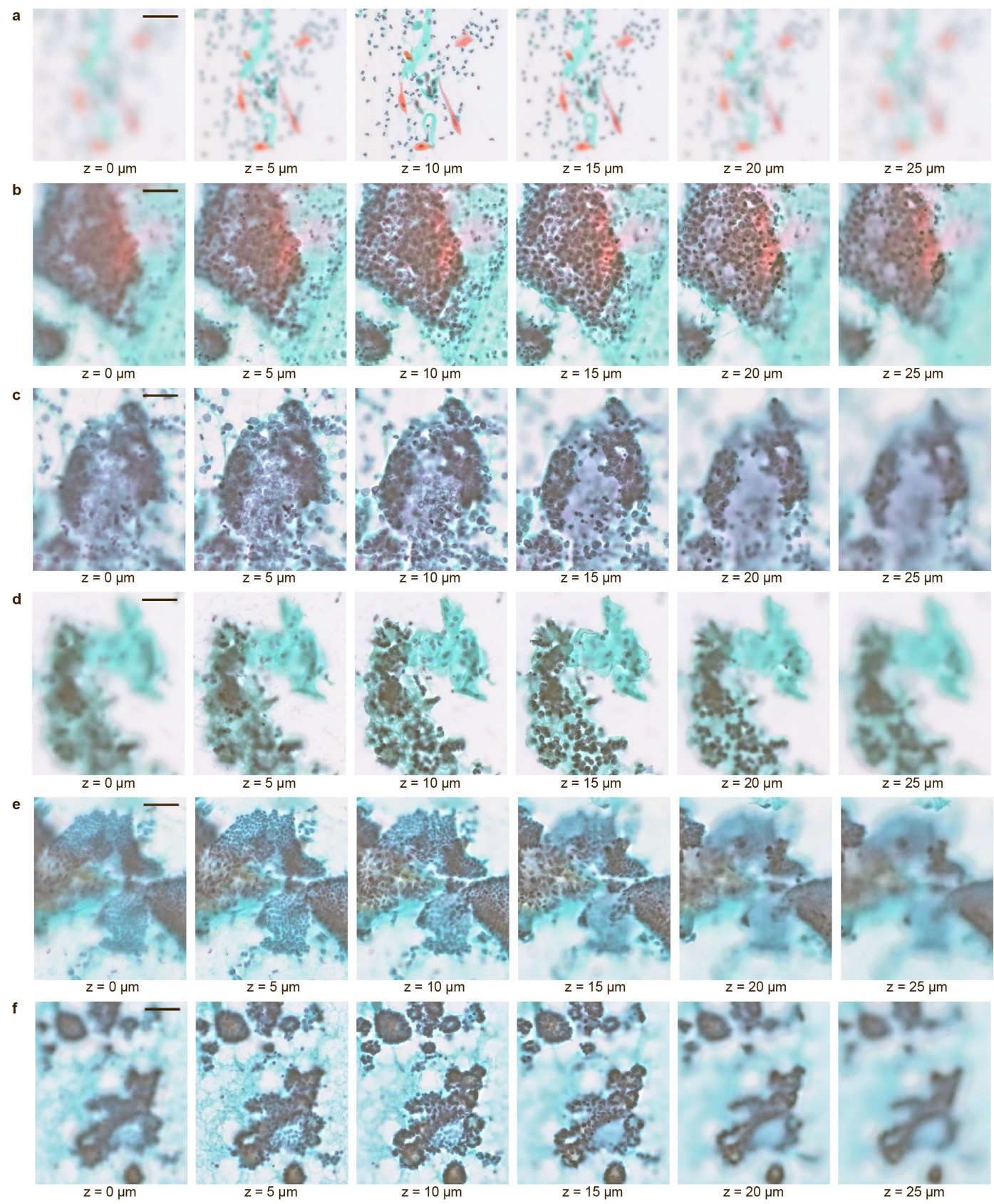

**Extended Data Fig. 4 | Depth-resolved tomograms.** Representative tomographic slices of cytology samples acquired at 1μm Z-intervals and displayed at 5μm steps. Shown are keratinizing SCC (**a**), non-keratinizing SCC (**b**), HPV-associated adenocarcinoma (**c**), HPV-independent adenocarcinoma (**d**), fibroadenoma (**e**), and follicular thyroid neoplasm (**f**). Slices highlight depth-resolved morphological features and structural deformations across the z-stack. Scale bars: 50μm.

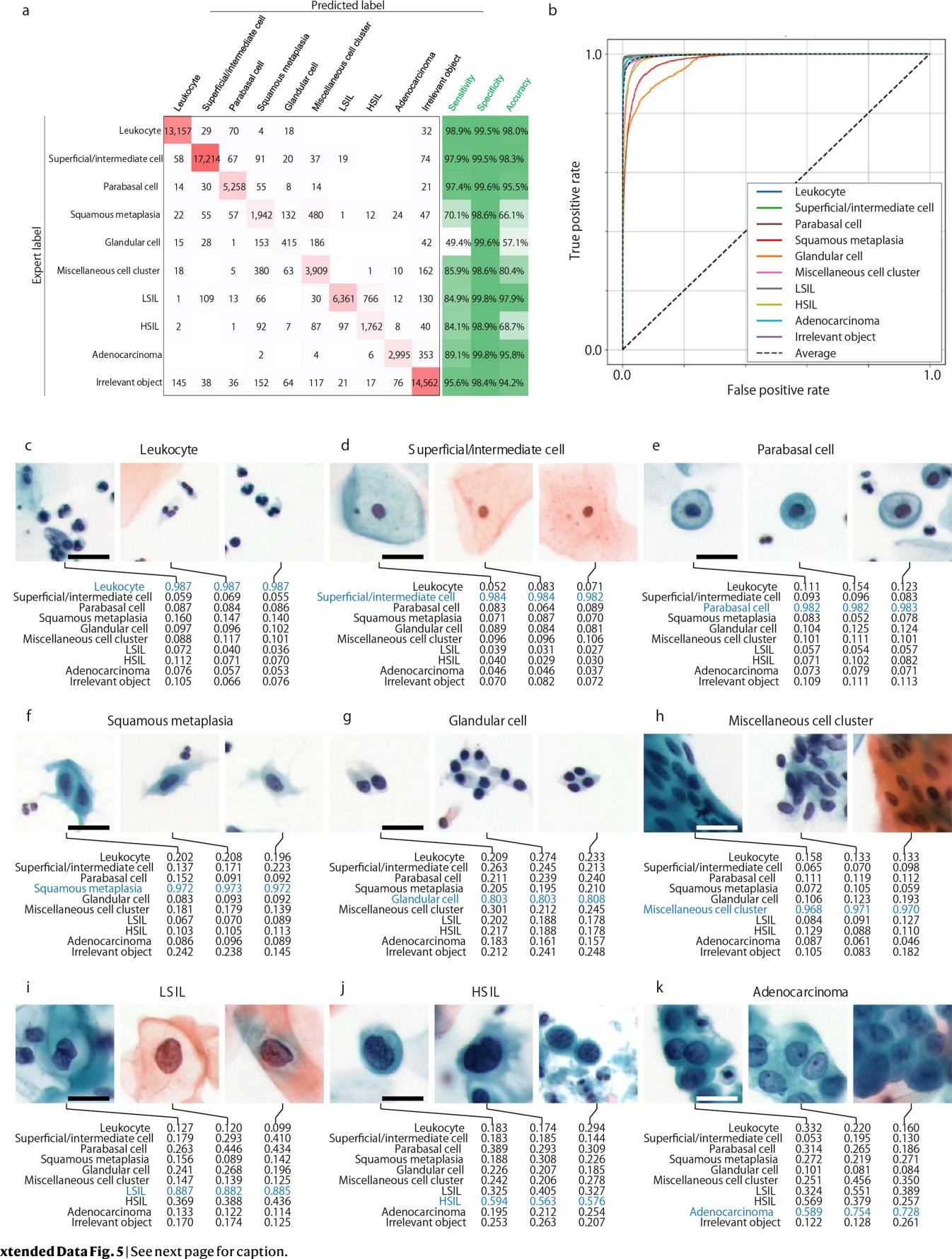

**Extended Data Fig. 5 | See next page for caption.**

**Extended Data Fig. 5 | AI-based cell classification performance. a**, Confusion matrix comparing AI inference results with expert cytologist annotations. Sensitivity, specificity, and overall accuracy metrics are provided for each cytological class. **b**, ROC curves at the single-entity (e.g., cell and cell cluster) level, illustrating classification performance across 10 cell types. AUC values were: 1.00 for leukocytes, superficial/intermediate squamous cells, parabasal cells, LSILs, adenocarcinoma cells, and irrelevant objects; 0.99 for HSILs and miscellaneous clusters; 0.98 for squamous metaplasia; and 0.97 for glandular cells. **c-k**, Representative AI-classified tomograms of individual cells from each category: leukocytes (**c**), superficial/intermediate squamous cells (**d**), parabasal cells (**e**), squamous metaplasia (**f**), glandular cells (**g**), miscellaneous cell clusters (**h**), LSILs (**i**), HSILs (**j**), and adenocarcinoma cells (**k**). Each panel includes the associated probability vector output by the vision transformer model, reflecting the AI's classification confidence across all 10 categories. Scale bars: 20μm.

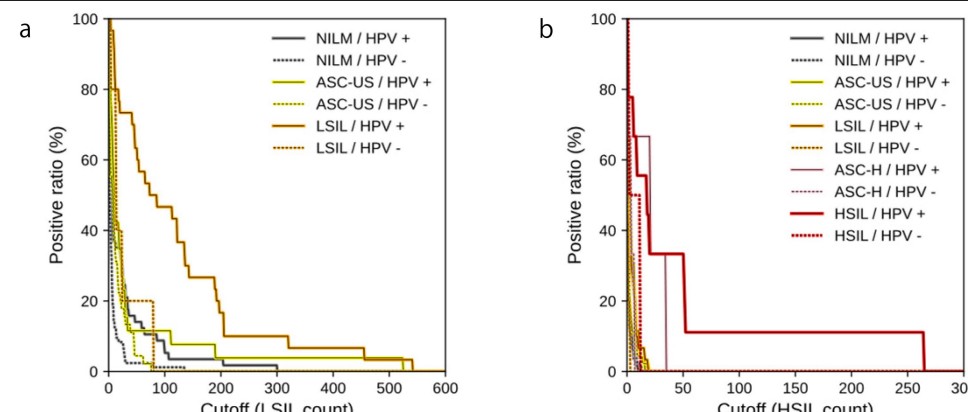

**Extended Data Fig. 6 | Proportion of slides classified as positive based on varying cell count thresholds. a**, Classification results for LSIL[+]. **b**, Classification results for HSIL[+].

a

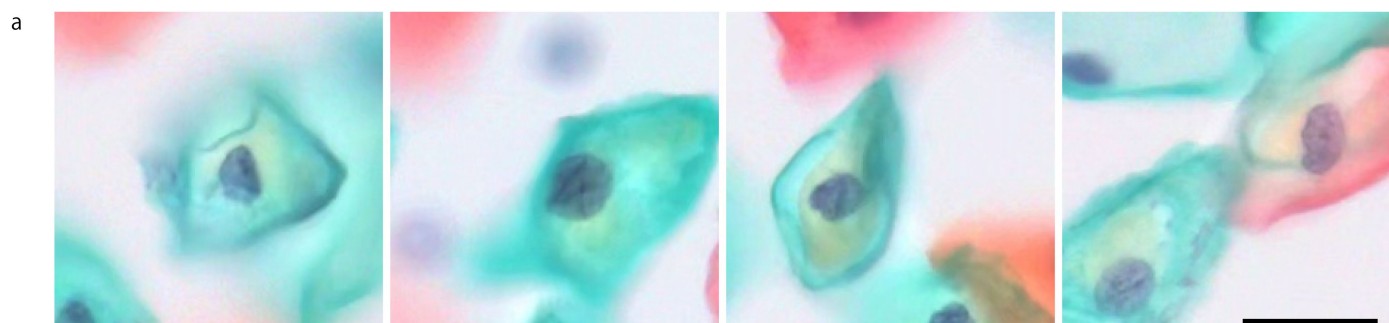

b **Validation (original dataset)** c **Validation (new dataset)**

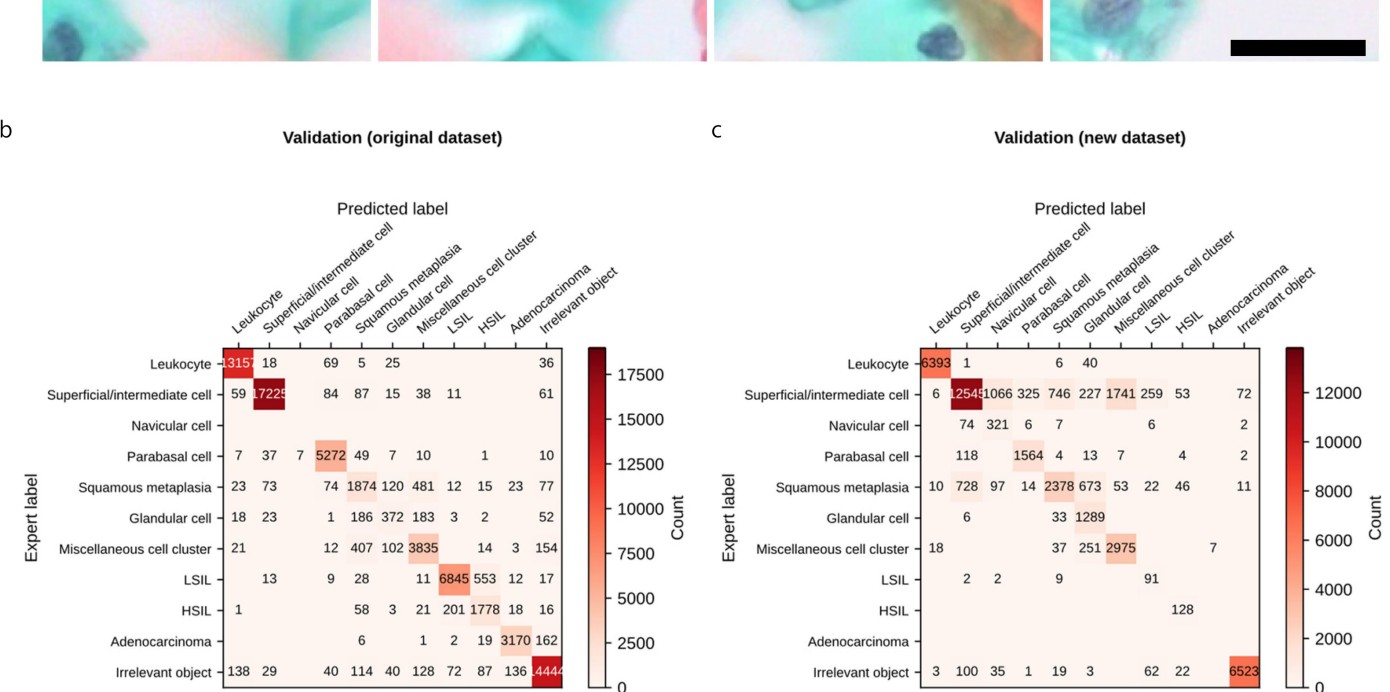

**Extended Data Fig. 7 | Training of the AI model with the navicular cell class for the multicentre study. a**, Representative examples of navicular cells from the second-round training dataset used to add the navicular class (*n* = 319 annotated navicular cell images). Scale bar: 20 μm. **b, c**, Validation of the updated AI model incorporating the navicular cell class. **b**, Confusion matrix for the original validation dataset used in the first training. **c**, Confusion matrix for the updated validation dataset prepared for the second training, which includes navicular cells as an additional category. Axes indicate expert labels (rows) versus predicted labels (columns). Cell colour intensity reflects counts (darker red = higher), with zero counts shown in white. A common colour scale is applied across panels.

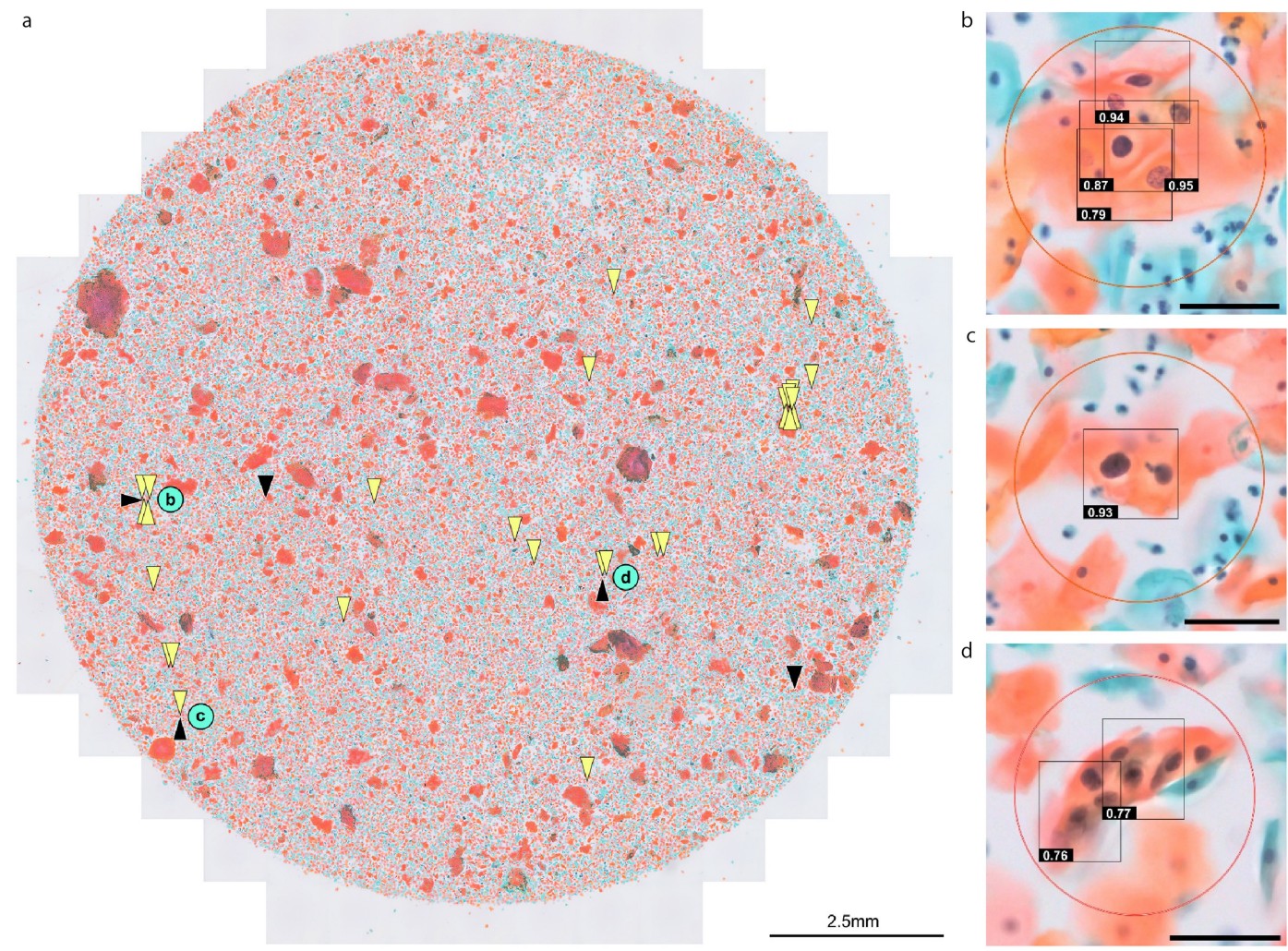

**Extended Data Fig. 8 | Whole-slide overview and zoomed comparisons of AI detections and expert annotations. a**, Whole-slide image of a cervical liquid-based cytology slide from Centre T. Yellow triangles mark cells flagged by the AI at LSIL probabil ≥ 0.75; black triangles mark locations labeled by a cytotechnologist on the same slide. Light-blue circled areas (**b-d**) indicate regions shown at higher magnification. Scale bar, 2.5 mm. **b-d**, Enlarged views of the regions indicated in **a**. Red circles denote cytotechnologist annotations; black bounding boxes indicate LSIL cells detected by the AI, with the number at the lower left giving the LSIL probability for each cell. Scale bars, 50 µm.

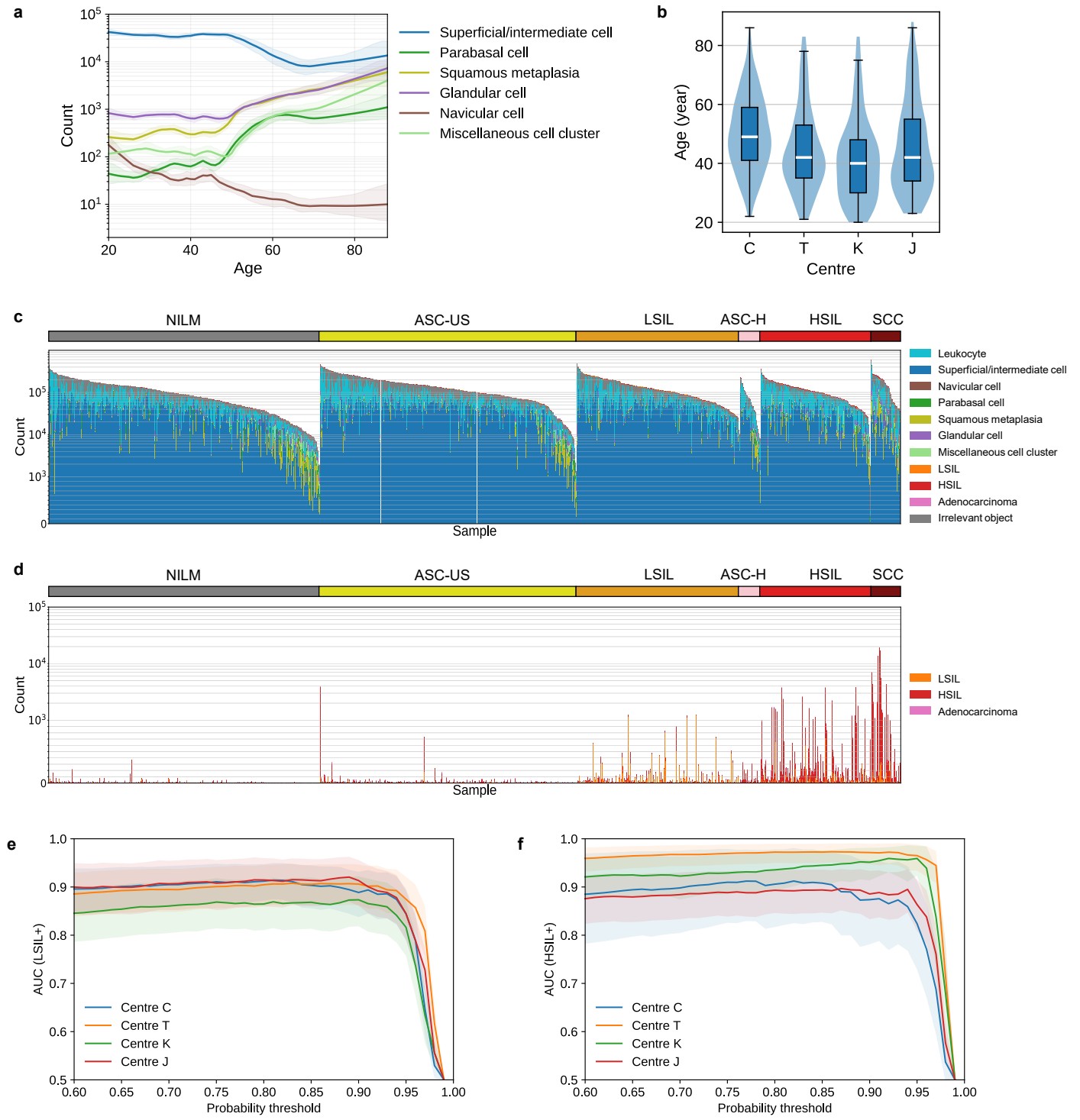

**Extended Data Fig. 9 | Data analysis of the multicentre study. a**, Whole-slide, log-scaled absolute counts of six epithelial cell types from four centres (*n* = 1,124), analyzed with the updated AI model, plotted against age. Shaded bands show 95% confidence intervals. Navicular cells (brown line) peak in the early 20 s. **b**, Age distribution by centre shown as a violin plot with overlaid boxplot (median line, IQR; whiskers = 1.5×IQR; outliers omitted). Medians (IQR): C 49 (41–59), T 42 (35–53), K 40 (30–48), J 42 (34–55). **c**, **d**, Distributions of whole-slide cell counts from 1,124 cervical liquid-based cytology samples across four centres, grouped by cytological diagnosis and sorted within each group by total cell count, counts across all annotated cell types including the additional navicular cell category (**c**); and counts restricted to abnormal (positive) classes (**d**). The y axis uses a hybrid linear–logarithmic scale (linear to $10^3$, logarithmic above) primarily to preserve visibility in **b**, where several SCC slides contain nearly $10^4$ HSIL cells. **e, f**, Threshold-AUC sensitivity for LSIL[+] (**e**) and HSIL[+] (**f**), computed per centre on the same datasets. For each endpoint, the positive-cell probability threshold was varied from 0.60 to 0.99; AUC values (solid lines, calculated from all available slides per centre) are shown with 95% bootstrap confidence intervals (shaded bands, estimated by stratified bootstrap resampling with 2,000 resamples). AUC values remained stable up to ~0.9, decreasing sharply beyond this threshold.

a
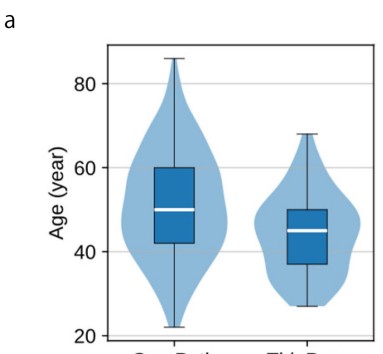

b
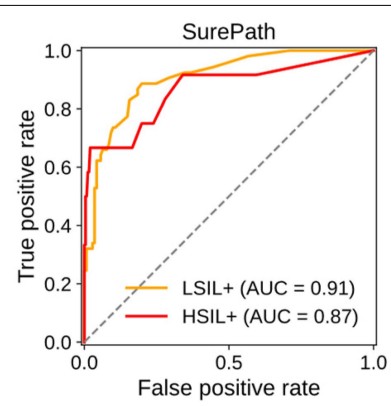

c
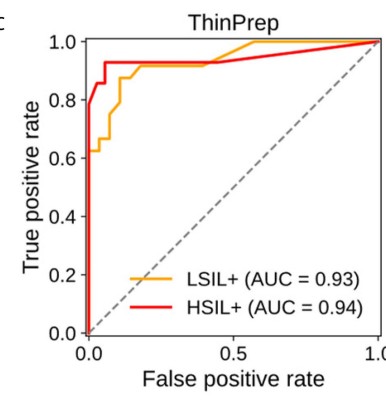

**Extended Data Fig. 10 | Comparison of sample preparation methods. a**, Age distributions for the SurePath and ThinPrep cohorts in the Cancer Institute Hospital of JFCR (median line, IQR; whiskers = 1.5×IQR; outliers omitted). Medians [IQR; *n*]: SurePath (*n* = 265) 50.0 [42.0–60.0], ThinPrep (*n* = 53) 45.0 [37.0–50.0]. **b, c**, ROC curves for detecting LSIL⁺ (LSIL, ASC-H, HSIL, SCC) and HSIL⁺ (HSIL, SCC) at the Cancer Institute Hospital of JFCR (C), stratified by liquid-based cytology sample preparation: SurePath (**b**, *n* = 265) and ThinPrep (**c**, *n* = 53). ROC curves use AI-derived cell counts as predictors; AUC values are shown in the legends for each endpoint.

Keisuke Goda
Tomohiro Chiba

# Reporting Summary

## Statistics

For all statistical analyses, confirm that the following items are present in the figure legend, table legend, main text, or Methods section.

| n/a | Confirmed | |
|---|---|---|
| ☐ | ☒ | The exact sample size (*n*) for each experimental group/condition, given as a discrete number and unit of measurement |
| ☐ | ☒ | A statement on whether measurements were taken from distinct samples or whether the same sample was measured repeatedly |
| ☐ | ☒ | The statistical test(s) used AND whether they are one- or two-sided *Only common tests should be described solely by name; describe more complex techniques in the Methods section.* |
| ☐ | ☒ | A description of all covariates tested |
| ☐ | ☒ | A description of any assumptions or corrections, such as tests of normality and adjustment for multiple comparisons |
| ☐ | ☒ | A full description of the statistical parameters including central tendency (e.g. means) or other basic estimates (e.g. regression coefficient) AND variation (e.g. standard deviation) or associated estimates of uncertainty (e.g. confidence intervals) |
| ☐ | ☒ | For null hypothesis testing, the test statistic (e.g. *F*, *t*, *r*) with confidence intervals, effect sizes, degrees of freedom and *P* value noted *Give P values as exact values whenever suitable.* |
| ☒ | ☐ | For Bayesian analysis, information on the choice of priors and Markov chain Monte Carlo settings |
| ☒ | ☐ | For hierarchical and complex designs, identification of the appropriate level for tests and full reporting of outcomes |
| ☐ | ☒ | Estimates of effect sizes (e.g. Cohen's *d*, Pearson's *r*), indicating how they were calculated |

*Our web collection on statistics for biologists contains articles on many of the points above.*

## Software and code

Policy information about availability of computer code

| Data collection | Images were acquired using in-house developed 3D imaging hardware as described in the Methods section. Annotation data were collected using the open-source software CVAT (version 2.7.6). |
|---|---|
| Data analysis | Data analysis: Data analysis was conducted using Python (versions 3.10 and 3.12) with several open-source libraries, including NumPy (1.26.4), pandas (2.2.2), matplotlib (3.10.3 / 3.9.2), seaborn (0.13.2), scikit-learn (1.6.1), statsmodels (0.14.4), PyTorch (2.1.1), torchvision (0.16.1), albumentations (2.0.8), OpenCV (4.11.0.86), timm (1.0.15), and ONNX Runtime (1.18.0). Custom Python scripts for 3D image analysis, model training and inference, and visualization were developed for this study and are publicly available at Zenodo (DOI: 10.5281/zenodo.17808303; https://doi.org/10.5281/zenodo.17808303). |

For manuscripts utilizing custom algorithms or software that are central to the research but not yet described in published literature, software must be made available to editors and reviewers. We strongly encourage code deposition in a community repository (e.g. GitHub). See the Nature Portfolio guidelines for submitting code & software for further information.

# Data

Policy information about availability of data

All manuscripts must include a data availability statement. This statement should provide the following information, where applicable:
- Accession codes, unique identifiers, or web links for publicly available datasets
- A description of any restrictions on data availability
- For clinical datasets or third party data, please ensure that the statement adheres to our policy

Anonymized CSV files sufficient to reproduce the quantitative figures and tables are publicly available (DOI: 10.5281/zenodo.17808303). These CSVs contain derived measurements and per-slide metadata (including center, sample preparation method, cytology diagnosis, age and HPV test result where available) but do not include raw images or directly identifiable information. The cytology datasets analyzed in this study are securely maintained by CYBO to safeguard patient privacy and proprietary imaging data. Due to ethical and regulatory constraints, these datasets are not publicly available. Academic investigators with no relevant conflicts of interest may request controlled access to selected de-identified cytological features for non-commercial, research-only purposes. Requests will be reviewed by CYBO in consultation with the sample-providing centers and their institutional review boards or ethics committees, and, if approved, will require a data-use agreement that prohibits re-identification and any redistribution of the data. Requests should be directed to N.N. at nitta@cybo.co.jp, and eligible requests will receive a response within one month.

# Research involving human participants, their data, or biological material

Policy information about studies with human participants or human data. See also policy information about sex, gender (identity/presentation), and sexual orientation and race, ethnicity and racism.

| | |
|---|---|
| Reporting on sex and gender | The study analyzed cervical cytology samples from female patients only. |
| Reporting on race, ethnicity, or other socially relevant groupings | No data on race, ethnicity, or other socially relevant groupings were collected or analyzed. |
| Population characteristics | At the Cancer Institute Hospital of JFCR (center C), 770 cervical cytology samples were collected from patients undergoing cervical cancer screening between 2011 and 2019. Of these, 318 slides were included in the multicenter evaluation. From the other centers, 222 (University of Tsukuba Hospital, center T), 384 (Kaetsu Comprehensive Health Development Center, center K), and 199 (Juntendo University Urayasu Hospital, center J) slides were included for evaluation. Cytology categories and HPV test results are summarized in Supplementary Table 1. Age distributions are shown in Extended Data Figure 9b. |
| Recruitment | Participants were enrolled via an opt-out process in accordance with approved institutional protocols and public notification. No direct recruitment or interventions beyond routine care were performed. |
| Ethics oversight | The study protocol for the use of archived human cervical cytology specimens was reviewed and approved by the Medical Research Ethics Review Committee at the Cancer Institute Hospital of the Japanese Foundation for Cancer Research (IRB No. 2019-GA-1190; covering centers C and K), the Clinical Research Ethics Review Committee at the University of Tsukuba Hospital (R07-175), and the Research Ethics Committee of the Faculty of Health Science at Juntendo University (2025-016). All procedures were conducted in accordance with the Declaration of Helsinki and all relevant institutional and national guidelines and regulations. Informed consent for research use of archived cytology specimens was obtained via each institution's opt-out process under its broad research-consent framework, as approved by the corresponding ethics committees. All samples were anonymized and identifiable information was removed prior to analysis. |

Note that full information on the approval of the study protocol must also be provided in the manuscript.

# Field-specific reporting

Please select the one below that is the best fit for your research. If you are not sure, read the appropriate sections before making your selection.

☒ Life sciences ☐ Behavioural & social sciences ☐ Ecological, evolutionary & environmental sciences

For a reference copy of the document with all sections, see nature.com/documents/nr-reporting-summary-flat.pdf

# Life sciences study design

All studies must disclose on these points even when the disclosure is negative.

| | |
|---|---|
| Sample size | All sample sizes for each analysis are explicitly reported in the figure legends and Methods (including per-centre counts for the multicentre evaluation). No formal a priori statistical power calculation was performed. Instead, during initial development at centre C we gradually increased the number of test cases and found that datasets of approximately 200–300 test cases in total already yielded clinically meaningful and consistent estimates of key performance metrics (e.g. sensitivity, specificity, and area under the ROC curve). On this basis, and taking into account feasibility at each site, we pragmatically targeted roughly 200 test cases per centre in the multicentre study. In the final analyses, the AI system showed statistically significant superiority over human ASC-US-based triage for identifying HPV-positive cases, which we consider evidence that the chosen sample sizes were sufficient for the aims of this study. |

| Data exclusions | Samples were excluded only when digital imaging failed (e.g., cellular content too sparse to permit reliable 3D imaging). These failures were identified before any statistical analysis; no post hoc exclusions were made. |
|---|---|
| Replication | Analyses were performed once per sample (one slide per donor; no technical repeats). For a given scanned slide, the image acquisition and AI analysis pipelines are deterministic, so repeating the experiment on the identical specimen would be expected to yield the same results and was therefore not pursued. Instead, we focused on assessing reproducibility and robustness at the level most relevant to the clinical application by (i) increasing the number of independent donor samples and validating performance on an independent multicentre cohort across four institutions, with AI performance summarised separately by centre and showing consistent trends, and (ii) examining the stability of diagnostic performance to analysis parameter choices. For example, in Extended Data Fig. 9e–f we varied the decision threshold from 0.60 to 0.99 in 0.01 increments and observed that the area under the ROC curve (AUC) remained stable over a broad threshold range (approximately 0.60–0.90). Taken together, these analyses support the reproducibility and robustness of our findings even though individual experiments were not technically replicated. |
| Randomization | This retrospective observational study involved no intervention and thus no participant randomization. At the primary center, assignment to training/validation/test splits for model development was randomized after excluding failed samples. For the multicenter evaluation, cases were target-accrued by cytological category (NILM, ASC-US, LSIL, ASC-H, HSIL, SCC) according to pre-specified targets and each center's availability, rather than by consecutive accrual or randomized allocation; all eligible cases meeting category targets were included. Performance was summarized per center to mitigate center-specific sampling imbalances. |
| Blinding | Annotators were not blinded to the diagnostic category of each slide for the purpose of cell-level annotations. Model training and inference were conducted on de-identified data, and predictions were generated without access to ground-truth labels; performance was computed after predictions were finalized. |

# Reporting for specific materials, systems and methods

We require information from authors about some types of materials, experimental systems and methods used in many studies. Here, indicate whether each material, system or method listed is relevant to your study. If you are not sure if a list item applies to your research, read the appropriate section before selecting a response.

## Materials & experimental systems

| n/a | Involved in the study |
|---|---|
| ☒ | Antibodies |
| ☒ | Eukaryotic cell lines |
| ☒ | Palaeontology and archaeology |
| ☒ | Animals and other organisms |
| ☒ | Clinical data |
| ☒ | Dual use research of concern |
| ☒ | Plants |

## Methods

| n/a | Involved in the study |
|---|---|
| ☒ | ChIP-seq |
| ☒ | Flow cytometry |
| ☒ | MRI-based neuroimaging |

## Plants

| Seed stocks | n/a |
|---|---|
| Novel plant genotypes | n/a |
| Authentication | n/a |

