## [Peer Review File · Nature]

Clinical-grade autonomous cytopathology via whole-slide edge tomography

Corresponding Author: Professor Keisuke Goda

Version 0:

Reviewer comments:

Referee #2

(Remarks to the Author)

This manuscript proposes a clinical-grade autonomous cytopathology pipeline leveraging high-resolution imaging, real-time edge-based image compression, and AI-driven inference to achieve scalable and objective cytological diagnostics. The pipeline effectively addresses long-standing challenges in cytology, including diagnostic subjectivity, inconsistency, and limited scalability. It incorporates flow cytometry-like morphological profiling (CMD) for comprehensive interpretation of cell populations.

The key contribution includes: (i) Real-time FPGA and GPU-enabled edge computing for rapid 3D scanning and HEVC-based compression, significantly enhancing throughput and data manageability (3–8 mins per slide), and the implementation of CMD-based vectors for probabilistic cell labeling, integrated with gating strategies and hierarchical rule-based lesion identification. (ii), the construction of a relatively large annotated dataset spanning 10 cytological classes; (iii) leveraging YoloX and MaxViT, this work demonstrates strong model performance (AUC > 0.99) in classifying LSIL, HSIL, and adenocarcinoma cells, and (iv) the identification of underdiagnosed cases through AI-driven quantitative analysis, highlighting potential clinical value.

Leveraging the AI via quantitative approaches can help identify false negative (underdiagnosis) by human expert (as the author shown in Fig 6b). This result is essential to justify the necessity of introducing AI into the clinical practice, especially cell counting and cytopathology. I think the author's comment on Fig 6b is great – the most critical mistake for human is underdiagnosis (false negatives), while AI can usually achieve nearly perfect recall but tends to introduce a lot of false positives. Combining human expertise and AI via human-AI collaboration can help eliminate both false positives and negatives. The pipeline that the author proposed, if involved with human-AI collaboration or served as co-pilot for clinician, will be icing on the cake.

That being said, I remain my concern about the innovation of this paper. Although I believe the main technical innovation is the development of a fully integrated pipeline that combines high resolution tomographic imaging, edge compression, and real time inference to enable scalable and autonomous cytological analysis; However, from a strictly hardware/software-innovation perspective, the system does not present fundamentally new technologies. The “edge computer” is an on-premise deployment of a computer and has limited innovation, as well as the adoption of existing cell identification and cell classification. Similar hardware integration for rapid 3d scanning for histology / cytology images have introduced in the past - e.g., <https://www.nature.com/articles/s44303-024-00042-2> (the author did cite this work but does not compare their uniqueness to that one). I think each technical part of this paper has marginal technical innovation. 3-8 mins/slide look impressive though.

For AI imaging analysis section, while overall I am impressed about the model performance, it remains unclear how well it would generalize to external data sources or samples acquired under different imaging protocols or patient populations. No OOD evaluation or multi-center validation is included. Second, the model heavily relies on rule-based threshold applied to CMD-derived output vectors, raising concerns about its true end-to-end learning capability.

Finally, while the CMD-based scoring system (Fig. 5) and classification probabilities (Fig. 4c-k) are informative, they primarily highlight high-confidence predictions. It remains unclear how the system handles ambiguous or low-confidence cells, especially those that fall between classification thresholds. Additional analyses on threshold sensitivity and visualization of borderline examples would help enhance the interpretability and clinical trustworthiness of the pipeline.

The choice of YOLOX for nucleus detection aligns with the system's goals of speed and edge deployment. However, Extended Data Fig. 6c–d reveals notable false positives in sparse and dense regions. No comparison was made against domain-specific nucleus detection methods (such as StarDist, CellViT, Hover-net), which may offer superior accuracy in complex cytology contexts. A more comprehensive benchmark would strengthen confidence in the robustness of the detection pipeline.

Minor comments:

- Fig 6e/f – when the author claimed that “The AI-derived LSIL counts correlated well with HPV positivity”, I recommend to provide statistical evidence, such as independent t-test results, to justify the level of such “correlation”.
- Fig 6a-b x-axis ticks font size is too small. Either enlarge it or drop it.
- While Fig 6d showed absolute cell count, I wonder if the author can provide cell proportion since the total number of cells within each sample varies.
- The data augmentation strategy is only briefly mentioned (simple rotations and flips), with no details on normalization, stain variation control, or domain adaptation. I think the paper may benefit more from more complex data augmentation, as well as using pathology foundation model to learn better cell image representation.
- Add some explanation that the 10D output is simply the ten-class probability vector, consider if reducing its dimensionality (though PCA or learned embeddings) could improve efficiency, and describe how near-tie predictions are handled.
- No clear benchmark that compare against alternative models was included. I am uncertain about how this system performs outside its own test domain.
- Since edge deployment is a core contribution, a brief elaboration on the trade-off between compression ratio and classification accuracy would be interesting.

(Remarks on code availability)

The code shared on Github has limited information about how the models (YOLOX, MaxViT) were adopted in their pipeline.

Referee #3

(Remarks to the Author)

I co-reviewed this manuscript with one of the reviewers who provided the listed reports.

(Remarks on code availability)

Referee #4

(Remarks to the Author)

The manuscript by Goda et al. presents a timely end-to-end platform for autonomous cytopathology. The authors are to be commended for comprehensively addressing several long-standing challenges in digital cytopathology, particularly the data bottleneck associated with high-resolution 3D whole-slide imaging. I believe this is a significant contribution to the field. The 'whole-slide edge tomography' system, which cleverly leverages edge computing and real-time compression, and the 'Cluster of Morphological Differentiation' (CMD) analytical framework, which provides an interpretable, flow cytometry-like paradigm for AI analysis—are particularly impressive.

While the study is of high quality, I recommend addressing the following points to further strengthen the manuscript before publication.

Major Revisions:

1. The manuscript adopts HEVC as the compression technique, leveraging inter-layer redundancy for compression. (1) Have the authors considered or compared other standards specifically designed for 3D medical imaging, such as JPEG 2000? (2) In addition, the manuscript argues that image quality is sufficient for diagnosis when PSNR exceeds 40 dB, based on PSNR values and visual inspection. Beyond visual assessment, have the authors conducted more rigorous validation—such as having pathologists perform diagnostic evaluations on both compressed and uncompressed images to compare diagnostic consistency? Furthermore, have the authors evaluated whether compression at different quality levels (high, medium, low) affects downstream tasks like AI model training or inference? This would help clarify whether compression significantly impacts analysis performance.

2. Regarding the definition and justification of “Clinical-Grade,” a clearer explanation would be appreciated. Although the model achieved an impressive AUC > 0.99 at the cell classification level, the AUC at the slide (case) level—which better simulates clinical diagnosis—for distinguishing LSIL+ and HSIL+ dropped to 0.81 and 0.82, respectively. Could the authors further discuss potential reasons for this performance decline? It would be ideal to provide a human expert benchmark—for example, what is the inter-observer agreement rate among pathologists on the same set of 318 cases? If the system's AUC of 0.81/0.82 is comparable to or exceeds the average performance of human experts, then the claim of “Clinical-Grade” becomes much more compelling.

3. Figures 2 and 3, which demonstrate system performance and imaging performance, appear somewhat simplified. The authors are encouraged to include additional quantitative analyses—such as how compression impacts downstream task accuracy, and comparisons with gold-standard imaging tools. These would help reinforce the system's overall performance

claims.

4. The dataset primarily consists of samples from a single institution (Cancer Institute Hospital of JFCR). Given that the model was trained at a leading hospital in Japan, how robust is its performance on samples from other institutions, with different staining protocols or patient demographics? If feasible, even a preliminary external validation using a small number of samples from other institutions would strengthen the study. Alternatively, this limitation should be clearly stated in the discussion.

Minor Revisions:

1. In Figure 4a, the sensitivity and accuracy for “Glandular cell” classification are notably low—the lowest among all categories. This appears to be a weakness. It is recommended to briefly mention this in the results or discussion section and analyze possible causes.
2. It will be interesting to include some discussions with the recent development of high-throughput mesoscale imaging systems, which can already cover wide-field-of-view and high-resolution for the gigapixel-imaging applications with the proposed systems.

(Remarks on code availability)

Codes are appropriate.

Referee #5

(Remarks to the Author)

This manuscript described the effort to develop an intriguing and unique AI-based technique for cytology analysis, mainly for liquid-based cervical cytology (Pap smears). Pap smear-based cervical cancer screen is the most successful cancer control story in human history, as it has saved many women’s lives. Recently there has been a shift from Pap smear to HPV testing for as a primary tool for cervical cancer screen. This is partly due to the fact that Pap smear cytology is subjective and requires trained cytologists, whereas there is a shortage of cytologists in developing countries that typically have high burden for cervical cancer. Thus, there has been an interest in developing AI-tools that can be used for cervical Pap smears. Several AI-systems have already developed and commercially available for Pap smear cytology (for example, FDA-cleared Hologic’s Genius Cervical AI system, etc.). Many of the tools developed so far have been used as an assistant tool for cytologist, not fully autonomous. By employing whole-slide edge tomography, high quality 3-D imaging, and cluster of morphological differentiation (CMD), the authors show that the technique described in the manuscript appears to be a fully autonomous platform that is distinctive from others in the field.

There are several questions and suggestions as presented below:

1. Title: The title implies the technique may be applicable to cover the entire field of cytopathology practice, from cervical Pap smear to FNA, etc., but most of the work presented are limited to liquid-based cervical cytology. In fact, the method may be more applicable for high cellularity samples such as Pap smears (with at least 10,000 cells), but may not be suitable for low cellularity samples (such as urine or CSF), or samples prepared by traditional smear technique. Therefore, the title should be stated more precisely for liquid-based cervical Pap smears.
2. For the claim that the platform is a “clinical-grade autonomous”, the authors provided some data, one of which is that the system may be more “accurate” than cytologists in classifying cells for LSIL or HSIL categories in a set of Pap smears (by combining the HPV test result, Fig 6). Evidence that the system can actually be used in clinical setting in a reproducible and reliable manner, is not presented. Though full clinical validations may require future multi-center prospective trials, some preliminary pilot data will be more convincing.
3. Figure 2b, the bar color for High and Medium is identical, is that intentional?
4. Figure 2c and 4k, what about SCC cells?
5. Methods: it appears both Thinprep and Surepath slides were used in the study. The slides prepared by the two methods are quite different. A major difference is that Surepath-prepared slide contains more 3-dimensional cellular clusters and has more overlapping cells. Any differences noted in the analysis? More discussion will also be helpful.

(Remarks on code availability)

Not all codes are available for review to protect IP.

Version 1:

Reviewer comments:

Referee #2

(Remarks to the Author)

After taking a closer look at all the referees’ comments and the author’s responses, I believe the author has addressed all my concerns. Recognizing that conducting additional ablation studies on different AI models (such as alternatives to YOLOX, as well as other pathology foundation models) and data augmentation would require substantial extra effort, I find the author’s current response adequate.

Regarding the multi-center study, I believe the authors’ revision has fully addressed both my concern and Referee #4’s concern.

In addition, I made a throughout review of statistical analyses, including the suitability of the tests used, the accuracy of the

descriptions for error bars, as well as probability values. After the inspection, all looks appropriate to me: The paper uses one-sided Mann-Whitney U tests for pairwise comparisons (e.g., NILM vs. higher categories, HPV⁺ vs. HPV⁻) with Benjamini-Hochberg (BH) correction. This is acceptable for non-parametric data with skewed distributions (cell counts are clearly non-Gaussian). While I personally believe multiple testing correction (BH) may not be strictly necessary in this context, including the Q-values adds rigor. Error bars in the latency and imaging performance plots (Figure 2e & Extended Data Figure 2d) are described as standard deviations, which is appropriate.

(Remarks on code availability)

Code is appropriate.

Referee #3

(Remarks to the Author)

I co-reviewed this manuscript with one of the reviewers who provided the listed reports.

(Remarks on code availability)

Referee #4

(Remarks to the Author)

I think the authors have done great work during revision by providing more comprehensive benchmarks and quantitative evaluations. These new analyses and experiments have fully addressed my previous concerns. I believe this will be a very nice and practical tool for next-generation high-throughput diagnosis, especially in the era of AI.

(Remarks on code availability)

The codes are fine and well-illustrated.

Referee #5

(Remarks to the Author)

The resubmission addressed most of the concerns noted in the previous review adequately, with detail point-by-point responses. I don't have additional comments.

(Remarks on code availability)

No but adequately addressed.

TO REFEREE 2

We sincerely thank Referee 2 for his/her thoughtful and constructive feedback on our manuscript. We have carefully considered all comments and have revised the manuscript accordingly. Below, we provide a point-by-point response, where each comment (shown in *italic*) is followed by our detailed reply. All corresponding changes in the manuscript are highlighted in **red** for clarity (see the revised version).

Referee 2's Major Comment 1:

This manuscript proposes a clinical-grade autonomous cytopathology pipeline leveraging high-resolution imaging, real-time edge-based image compression, and AI-driven inference to achieve scalable and objective cytological diagnostics. The pipeline effectively addresses long-standing challenges in cytology, including diagnostic subjectivity, inconsistency, and limited scalability. It incorporates flow cytometry-like morphological profiling (CMD) for comprehensive interpretation of cell populations.

The key contribution includes: (i) Real-time FPGA and GPU-enabled edge computing for rapid 3D scanning and HEVC-based compression, significantly enhancing throughput and data manageability (3--8 mins per slide), and the implementation of CMD-based vectors for probabilistic cell labeling, integrated with gating strategies and hierarchical rule-based lesion identification. (ii), the construction of a relatively large annotated dataset spanning 10 cytological classes; (iii) leveraging YoloX and MaxViT, this work demonstrates strong model performance (AUC > 0.99) in classifying LSIL, HSIL, and adenocarcinoma cells, and (iv) the identification of underdiagnosed cases through AI-driven quantitative analysis, highlighting potential clinical value.

Leveraging the AI via quantitative approaches can help identify false negative (underdiagnosis) by human expert (as the author shown in Fig 6b). This result is essential to justify the necessity of introducing AI into the clinical practice, especially cell counting and cytopathology. I think the author's comment on Fig 6b is great – the most critical mistake for human is underdiagnosis (false negatives), while AI can usually achieve nearly perfect recall but tends to introduce a lot of false positives. Combining human expertise and AI via human-AI collaboration can help eliminate both false positives and negatives. The pipeline that the author proposed, if involved with human-AI collaboration or served as co-pilot for clinician, will be icing on the cake.

Authors' Response:

We thank the reviewer for the positive, thoughtful, and encouraging comments regarding our manuscript. We fully agree that underdiagnosis (false negatives) represents the most critical challenge in cytopathology and that AI can play a unique role in mitigating this risk. Our results in Figure 6b highlight exactly this point, demonstrating that AI-based quantitative analysis can identify abnormal cells that may be overlooked during human screening. We also agree with the reviewer's insightful suggestion that combining human expertise with AI has the potential to further enhance diagnostic accuracy by reducing both false negatives and false positives. While the focus of our study was to establish and validate a fully autonomous cytopathology pipeline that meets clinical-grade performance requirements, we envision that the platform could be flexibly deployed either in a fully autonomous mode or in a collaborative human-AI mode depending on clinical context. In fact, the interpretable CMD-based framework we propose was specifically designed to facilitate intuitive visualization of cell populations, thereby making it well suited for seamless integration into human-AI collaborative workflows.

Referee 2's Major Comment 2:

That being said, I remain my concern about the innovation of this paper. Although I believe the main technical innovation is the development of a fully integrated pipeline that combines high resolution tomographic imaging, edge compression, and real time inference to enable scalable and autonomous cytological analysis; However, from a strictly hardware/software-innovation perspective, the system does not present fundamentally new technologies. The "edge computer" is an on-premise deployment of a computer and has limited innovation, as well as the adoption of existing cell identification and cell classification. Similar hardware integration for rapid 3d scanning for histology / cytology images have introduced in the past - e.g., <https://www.nature.com/articles/s44303-024-00042-2> (the author did cite this work but does not compare their uniqueness to that one). I think each technical part of this paper has marginal technical innovation. 3-8 mins/slide look impressive though.

Authors' Response:

We appreciate the reviewer's focus on the innovative aspects of our work. Our contribution is not centered on a single new hardware or software component, but rather on a co-designed, in-tomograph computational pipeline that makes clinical-grade, whole-slide 3D autonomous cytology practical by eliminating the dominant bottleneck of data movement. This is achieved through an overlapped processing chain encompassing image acquisition → image reconstruction → in-tomograph image compression → AI inference, all executed within the instrument.

Concretely, our whole-slide edge tomography streams $4.5k \times 4.5k$ (~20 Mpixel) frames at up to 50 fps directly into an on-instrument SOM/FPGA/GPU stack, where on-the-fly HEVC compression exploits both intra-layer and Z-axis redundancy to reduce data volumes (e.g., ~1 GB for a 10-layer SurePath whole slide at PSNR \geq 40–42 dB). Because compression operates in parallel with acquisition, it does not limit overall wall-clock performance. The resulting throughput, approximately 3, 4.5, and 8 minutes per slide for 10, 20, and 40 layers, respectively, enables real-time operation, while our custom viewer delivers high-resolution tiles with <100 ms latency. These results demonstrate that the innovation lies in end-to-end integration, which achieves real-time functionality, rather than in simply repackaging an on-premise server as an "edge" device.

Regarding the cited Kim et al. system, we have now explicitly clarified that their work advances optical parallelism (via multi-camera array and high-throughput image stitching), whereas our approach emphasizes in-tomograph data reduction and workflow integration to maintain a bandwidth-agnostic, clinic-ready acquisition-to-analysis chain – complementary solutions addressing distinct system-level bottlenecks.

To prevent potential ambiguity, we have revised the text to clarify the locus of computation and the system's positioning. Specifically, we added the following statement in the Discussion section to situate our platform relative to recent mesoscale imaging systems, without expanding the reference list:

"While high-throughput mesoscale gigapixel imaging systems, including multi-camera array scanners, continue to advance³⁷, our platform emphasizes a seamless end-to-end clinical workflow, from edge-side compression and secure data transfer to on-site AI inference, enabling real-time operation in routine diagnostic settings. As data handling and system integration for mesoscale imaging mature, such approaches may serve as complementary wide-field front ends to our 3D morphology-aware analytical framework."

Finally, beyond the hardware–software co-design, our study introduces a flow cytometry–like morphological profiling framework, CMD, an interpretable, quantitative embedding space for large-scale morphological analysis. This enables visualization and stratification of millions of cells according to their phenotypic continuum, uncovering transitional states such as those between LSIL and HSIL that are not well captured by conventional categorical labels as we stated in the Discussion section. Together, the edge-integrated tomographic architecture and CMD-based analytical framework define a new paradigm for autonomous cytology, one that transforms imaging, compression, and analysis from discrete steps into a unified computational process optimized for clinical use.

Referee 2's Major Comment 3:

For AI imaging analysis section, while overall I am impressed about the model performance, it remains unclear how well it would generalize to external data sources or samples acquired under different imaging protocols or patient populations. No OOD evaluation or multi-center validation is included. Second, the model heavily relies on rule-based threshold applied to CMD-derived output vectors, raising concerns about its true end-to-end learning capability.

Authors' Response:

We appreciate the reviewer's thoughtful suggestions. Following the reviewer's advice, we have expanded our clinical performance evaluation by incorporating additional patients and centers, encompassing four institutions; Cancer Institute Hospital of JFCR (C), University of Tsukuba Hospital (T), Kaetsu Comprehensive Health Development Center (K), and Juntendo University Urayasu Hospital (J), comprising a total of 1,124 cervical liquid-based cytology slides (C: $n = 318$; T: $n = 222$; K: $n = 385$; J: $n = 199$). The updated model (identical architecture and hyperparameters, with no site-specific tuning) achieved slide-level AUCs of 0.86–0.91 for LSIL⁺ and 0.89–0.97 for HSIL⁺ across the four centers, demonstrating robust generalization to external cohorts with differing sample acquisition and workflow characteristics. As shown in Fig. 7e (also shown below), the ROC curves were consistently excellent across all centers. Consistent with our original findings, LSIL counts correlated with HPV positivity, whereas HSIL counts scaled with cytological diagnostic severity, supporting both the biological and clinical validity of the model's outputs (Figs. 7a and 7b).

Figure 7 | Multicenter evaluation of clinical performance. e, Slide-level ROC curves for detecting LSIL⁺ (LSIL, ASC-H, HSIL, SCC) and HSIL⁺ (HSIL, SCC) on the AI model, computed separately by center; C (n = 318), T (n = 222), K (n = 385), J (n = 199), using AI-derived cell counts as predictors. AUCs are shown in the legend for each center and endpoint.

Regarding the concern that our slide-level decision-making might critically depend on a rule-based threshold applied to cell-level scores, we explicitly quantified threshold sensitivity. As shown in Figs. 7f and 7g (also see below), we varied the cell-positivity threshold from 0.60 to 0.99 and recomputed slide-level AUCs separately for each center and endpoint. The AUCs remained essentially stable up to ~0.9 across all centers, with only a modest decline at very stringent thresholds, where few cells met the positivity criterion. These analyses confirm that the model's performance is not contingent on a single, brittle operating point; instead, the end-to-end learned cell classifier produces well-calibrated scores whose aggregated counts remain robust across a wide range of reasonable thresholds.

Figure 7 | Multicenter evaluation of clinical performance. f, g, Threshold-AUC sensitivity for the LSIL⁺ (f) and HSIL⁺ (g) endpoints, computed per center (C, T, K, J) on the same datasets as in e. For each endpoint, we vary the positive-cell probability threshold from 0.60 to 0.99 and plot AUCs (lines) with 95% bootstrap confidence intervals (shading; stratified resampling). Across centers, AUCs remain stable up to ~0.9 and drop sharply beyond this threshold. h, i, Slide-level ROC curves for AI-based detection of HPV positivity with 95% confidence bands; human cytology operating points (ASC-US⁺, LSIL⁺) are overlaid; h for all centers, i for only Center C. In the all-centers analysis, the AI model significantly exceeded ASC-US⁺ at matched specificity and matched sensitivity (both p = 0.034) but was inferior to LSIL⁺ (both p < 0.001). At Center C, where cytology and HPV were sampled on the same day, the AI model significantly exceeded ASC-US⁺ at matched specificity and sensitivity (both p = 0.002) and showed no significant difference versus LSIL⁺.

Referee 2's Major Comment 4:

Finally, while the CMD-based scoring system (Fig. 5) and classification probabilities (Fig. 4c-k) are informative, they primarily highlight high-confidence predictions. It remains unclear how the system handles ambiguous or low-confidence cells, especially those that fall between classification thresholds. Additional analyses on threshold sensitivity and visualization of borderline examples would help enhance the interpretability and clinical trustworthiness of the pipeline.

Authors' Response:

We thank the reviewer for raising this important point regarding the handling of ambiguous or low-confidence cells. As outlined in the “CMD-based cell population analysis” subsection of the Methods (third paragraph), the classification process employs a hierarchical gating strategy: irrelevant objects and leukocytes are first excluded, followed by sequential gates for LSIL, HSIL, and adenocarcinoma. For cells that remain outside these gates, the class with the highest probability score is assigned, ensuring that each cell is uniquely classified, even when confidence is intermediate. Cells with transitional or mixed morphological features are captured in this framework and are illustrated in Fig. 5e (also see below); for example, the progression from parabasal to superficial cells (p1-p4) and from glandular to metaplastic cells (q1-q4).

Figure 5 | CMD-based cell population analysis. e, Representative single-cell images illustrating gradual morphological transitions observed in b: from parabasal to superficial/intermediate squamous cells (p1-p4) and from glandular to metaplastic cells (q1-q4).

Referee 2's Major Comment 5:

The choice of YOLOX for nucleus detection aligns with the system's goals of speed and edge deployment. However, Extended Data Fig. 6c–d reveals notable false positives in sparse and dense regions. No comparison was made against domain-specific nucleus detection methods (such as StarDist, CellViT, Hovernet), which may offer superior accuracy in complex cytology contexts. A more comprehensive benchmark would strengthen confidence in the robustness of the detection pipeline.

Authors' Response:

We thank the reviewer for this valuable comment. During system development, we tested several detection models, including alternatives to YOLOX, but we did not perform systematic benchmarking across domain-specific nucleus detection methods. As new architectures continue to emerge, we consider a comprehensive model comparison to be beyond the scope of this manuscript, which focuses on the design and integration of the end-to-end pipeline. We acknowledge the presence of false positives in both sparse and dense regions, as shown in Supplementary Figs. 3c and 3d. These are largely mitigated in the downstream MaxViT-based classification stage, where irrelevant objects are filtered out. Importantly, the YOLOX model was intentionally configured to favor higher sensitivity at the detection stage, minimizing the likelihood of missing diagnostically relevant cells at the cost of some false positives, a trade-off appropriate for practical cytological screening.

To make this design choice explicit to readers, we have added the following sentences to the subsection titled “Detection of cell nuclei” in the Methods section:

“Moreover, atypical cells exhibit greater morphological variability than normal cells and are therefore more prone to occasional detection misses. Because these atypical cells represent the primary diagnostic targets, we intentionally biased the detector toward higher sensitivity to ensure broad coverage of abnormal cell populations.”

Referee 2’s Minor Comment 1:

Fig 6e/f – when the author claimed that “The AI-derived LSIL counts correlated well with HPV positivity”, I recommend to provide statistical evidence, such as independent t-test results, to justify the level of such “correlation”.

Authors’ Response:

We thank the reviewer for this helpful suggestion. Because cell-count distributions are non-Gaussian and right-skewed, we used a non-parametric alternative to the independent t-test. Specifically, we now report two-sided Mann–Whitney U tests with Benjamini–Hochberg adjustment for multiple comparisons. In the NILM subset comparing HPV– vs. HPV+, the LSIL counts and HSIL counts show significant differences with Benjamini–Hochberg-adjusted q-values of 0.005 and 0.038, respectively (Figs. 6e and 6f, also see below). In the multicenter study, as shown in Figs. 7c and 7d (also shown below), all four centers exhibit significant differences between HPV– and HPV+ for both LSIL and HSIL cell counts ($q < 0.05$ at each site).

Figure 6 | Clinical-grade performance. e, f, AI-detected LSIL (e) and HSIL (f) cell counts, stratified by cytological diagnosis and HPV status; analyses restricted to slides with available HPV results (n = 266). One-sided Mann-Whitney tests (HPV+ > HPV-) within each diagnosis is used. Benjamini-Hochberg-adjusted q values are reported with significance indicated as * q < 0.05, ** q < 0.01, and * q < 0.001.**

Figure 7 | Multicenter evaluation of clinical performance. c, d, AI-detected LSIL (c) and HSIL (d) cell counts by HPV status (-, +) within each center (C, T, K, J). Distributions are shown as violin plots with overlaid boxplots (n = 814). One-sided Mann–Whitney tests (a priori HPV+ > HPV-) are performed within center; Benjamini–Hochberg–adjusted q values are reported with significance indicated as * q < 0.05, ** q < 0.01, and * q < 0.001. In a-d, the y axis uses a hybrid linear-logarithmic scale to accommodate the wide dynamic range: linear up to 10² for LSIL (a, c) and up to 5×10² for HSIL (b, d), and logarithmic above these thresholds.**

Referee 2’s Minor Comment 2:

Fig 6a-b x-axis ticks font size is too small. Either enlarge it or drop it.

Authors’ Response:

We thank the reviewer for the helpful suggestion. In Figs. 6a and 6b, the x-axis tick labels were per-slide IDs, which are not required to interpret the grouping by cytology category or the overall trends. In the revised manuscript, we have removed the x-axis labels in both panels to improve clarity and legibility (as shown below). The figure caption now explicitly notes that samples are grouped by cytology category along the x-axis and sorted by total counts within each group, so no information is lost by dropping the labels.

Figure 6 | Clinical-grade performance. a, b, Whole-slide cell counts from 318 cervical liquid-based cytology samples, grouped by cytological diagnosis and sorted within each group by total cell count; counts across all cell types (a) and counts limited to abnormal (positive) cell classes (b). Cell counts per slide span from several thousand to several hundred thousand, with slides containing high total counts typically enriched in superficial/intermediate squamous cells.

Referee 2's Minor Comment 3:

While Fig 6d showed absolute cell count, I wonder if the author can provide cell proportion since the total number of cells within each sample varies.

Authors' Response:

We thank the reviewer for this insightful comment. We agree that proportion-based metrics can, in principle, control for variability in total cellularity. In addition to the absolute-count analyses reported in the manuscript, we reanalyzed all data including the multicenter results using proportion-normalized predictors. Specifically, we evaluated two schemes: (i) LSIL⁺/HSIL⁺ per total detected nuclei, and (ii) LSIL⁺/HSIL⁺ per total epithelial cells. Across the four centers, the resulting AUCs differed from the count-based results by only a few percentage points: C and T showed small increases, whereas K and J showed small decreases. Given these minimal deltas and to keep the main text concise, we retained absolute counts in the manuscript.

For transparency, we provide the results corresponding to Figs. 7a-7e below (representative plots shown for the per-total-epithelial normalization; the per-superficial-cell normalization yielded qualitatively similar results).

Figure 7 | Multicenter evaluation of clinical performance. a, b, AI-detected LSIL (a) and HSIL (b) cell counts across four centers (C, T, K, J), stratified by cytological diagnosis. Distributions are shown as violin plots with individual slides overlaid as points. **c, d**, AI-detected LSIL (c) and HSIL (d) cell counts by HPV status (-, +) within each center (C, T, K, J). Distributions are shown as violin plots with overlaid boxplots (n = 814). One-sided Mann-Whitney tests (a priori HPV+ > HPV-) are performed within center; Benjamini-Hochberg-adjusted q values are reported with significance indicated as * q < 0.05, ** q < 0.01, and *** q < 0.001. In **a-d**, the y axis uses a hybrid linear-logarithmic scale to accommodate the wide dynamic range: linear up to 10² for LSIL (a, c) and up to 5 × 10² for HSIL (b, d), and logarithmic above these thresholds. **e**, Slide-level ROC curves for detecting LSIL⁺ (LSIL, ASC-H, HSIL, SCC) and HSIL⁺ (HSIL, SCC) on the AI model, computed separately by center; C (n = 318), T (n = 222), K (n = 385), J (n = 199), using AI-derived cell counts as predictors. AUCs are shown in the legend for each center and endpoint.

Referee 2's Minor Comment 4:

The data augmentation strategy is only briefly mentioned (simple rotations and flips), with no details on normalization, stain variation control, or domain adaptation. I think the paper may benefit more from more complex data augmentation, as well as using pathology foundation model to learn better cell image representation.

Authors' Response:

We thank the reviewer for the thoughtful comments regarding data augmentation and the use of foundation models. Our guiding philosophy in this work was to prioritize data fidelity at acquisition over extensive downstream augmentation. To that end, we invested in a tomography-based imaging pipeline that reconstructs focused, nucleus-centered views with consistent 3D contrast, and we curated a large, diverse cohort spanning multiple centers and preparation workflows. This upstream control, rather than heavy reliance on synthetic transformations, proved most effective in our context.

Concretely, we employed only simple augmentations (e.g., rotations, flips). Despite this minimal augmentation, the model generalized across four independent centers (n = 1,124) with slide-level AUCs of 0.86-0.91 for LSIL⁺ and 0.89-0.97 for HSIL⁺ and maintained robust performance across wide threshold sweeps (Figs. 7f and 7g, also see below). We interpret this as evidence that high-fidelity imaging combined with broad data coverage can deliver strong and reproducible generalization across heterogeneous staining and sample-preparation conditions, without the need for heavy augmentation or large foundation backbones.

Figure 7 | Multicenter evaluation of clinical performance. f, g, Threshold-AUC sensitivity for the LSIL⁺ (f) and HSIL⁺ (g) endpoints, computed per center (C, T, K, J) on the same datasets as in e. For each endpoint, we vary the positive-cell probability threshold from 0.60 to 0.99 and plot AUCs (lines) with 95% bootstrap confidence intervals (shading; stratified resampling). Across centers, AUCs remain stable up to ~0.9 and drop sharply beyond this threshold.

Importantly, this data-centric design also demonstrated extensibility within the study. In the second training cycle for multicenter evaluation, we expanded the taxonomy from 10 to 11 classes by introducing navicular cells, using a small, targeted set of new annotations (from a limited number of slides at one participating center), while keeping the model architecture and hyperparameters unchanged. The classifier successfully adapted to this new category without degrading overall performance (Extended Data Figs. 5b and 5c, also see below), indicating that the learned representation is amenable to incremental phenotype expansion with

modest labeling effort. We view this as a proof-of-concept toward a cytology-oriented foundation model: as our tomographic corpus grows, self-supervised pretraining on high-quality 3D cell imagery coupled with lightweight task-specific heads could enable broad cytological coverage under practical runtime constraints.

Extended Data Figure 5 | Training of the AI model with the navicular cell class for the multicenter study. b, Confusion matrix for the original validation dataset used in the first training. c, Confusion matrix for the updated validation dataset prepared for the second training, which includes navicular cells as an additional category. Axes indicate expert labels (rows) versus predicted labels (columns). Cell color intensity reflects counts (darker red = higher), with zero counts shown in white. A common color scale is applied across panels.

We agree that pathology foundation models are highly promising. However, most current approaches are optimized for 2D histopathology and remain computationally demanding for high-throughput cytology workflows. Given our design goals of edge deployability and low latency, we adopted a streamlined architecture paired with high-quality imaging. As future work, we plan to benchmark cytology-specific pretraining, explore lightweight adapters or model distillation to preserve throughput, and systematically evaluate stain-invariant augmentations and domain-adaptation strategies in a controlled study.

Referee 2’s Minor Comment 5:

Add some explanation that the 10D output is simply the ten-class probability vector, consider if reducing its dimensionality (though PCA or learned embeddings) could improve efficiency, and describe how near-tie predictions are handled.

Authors’ Response:

We appreciate the reviewer’s insightful suggestion. As described in the Methods section (“CMD-based cell population analysis”), the 10-dimensional output of the vision transformer corresponds to the class probability vector for each cell. To clarify this point for readers, we have added the following sentence to the Results section under CMD-based cell population analysis:

“In this framework, each cell is represented by a 10-dimensional class probability vector (see the Methods section for details), which serves as the CMD markers.”

Regarding dimensionality reduction, the UMAP visualizations shown in Figs. 5a-5d represent applications of such techniques to the 10-dimensional CMD vectors for interpretability and visualization. With respect to near-tie predictions, Fig. 5e (p1-p4 and q1-q4) illustrates cells with intermediate CMD values that exhibit transitional morphologies between classes (e.g., from parabasal to superficial, or from glandular to metaplastic). These cells often display mixed morphological features characteristic of both classes, reflecting their ambiguous biological nature.

Such ambiguous populations are reminiscent of cases traditionally categorized as ASC-US or ASC-H, where interobserver variability among human cytologists is typically high. We believe this morphological continuum, captured quantitatively in the CMD space, underlies the inherent subjectivity observed in conventional cytology.

Referee 2's Minor Comment 6:

No clear benchmark that compare against alternative models was included. I am uncertain about how this system performs outside its own test domain.

Authors' Response:

We appreciate the reviewer's concern and agree that external performance should be demonstrated clearly. Our new multicenter evaluation (four independent institutions; $n = 1,124$) provides an explicit out-of-domain (OOD) test: two centers (T and J) were never used for training or tuning, and no site-specific calibration was applied. The model maintained high slide-level AUCs across all sites (LSIL⁺: 0.86-0.91; HSIL⁺: 0.89-0.97), demonstrating robust generalization under heterogeneous sample-preparation and staining workflows (Fig. 7e, also see below).

Figure 7 | Multicenter evaluation of clinical performance. e, Slide-level ROC curves for detecting LSIL⁺ (LSIL, ASC-H, HSIL, SCC) and HSIL⁺ (HSIL, SCC) on the AI model, computed separately by center; C ($n = 318$), T ($n = 222$), K ($n = 385$), J ($n = 199$), using AI-derived cell counts as predictors. AUCs are shown in the legend for each center and endpoint.

Furthermore, to assess potential influence of sample preparation methods used at different centers, we compared SurePath and ThinPrep slides with comparable age distributions (Extended Data Figure 8a, also see below). Both preparation methods yielded similarly favorable ROC curves and AUCs (Extended Data Figures 8b and 8c, also see below), demonstrating that model performance is robust to slide preparation.

Extended Data Figure 8 | Comparison of sample preparation methods. a, Age distributions for the SurePath and ThinPrep cohorts in the Cancer Institute Hospital of JFCR (median line, IQR; whiskers = 1.5×IQR; outliers omitted). Medians [IQR]: SurePath 50.0 [42.0–60.0], ThinPrep 45.0 [37.0–50.0]. b, c, ROC curves for detecting LSIL⁺ (LSIL, ASC-H, HSIL, SCC) and HSIL⁺ (HSIL, SCC) at the Cancer Institute Hospital of JFCR (c), stratified by liquid-based cytology sample preparation: SurePath (b, n = 265) and ThinPrep (c, n = 53). ROC curves use AI-derived cell counts as predictors; AUCs are shown in the legends for each endpoint.

As noted in our response to Major Comment 5, a comprehensive architecture benchmark lies beyond the scope of this pipeline-focused study. Nonetheless, the prospective, cross-site results presented here substantiate the model’s OOD generalization without the need for site-specific adaptation.

Referee 2’s Minor Comment 7:

Since edge deployment is a core contribution, a brief elaboration on the trade-off between compression ratio and classification accuracy would be interesting.

Authors’ Response:

We thank the reviewer for highlighting the importance of the compression-accuracy trade-off for edge deployment. In our system, image compression is applied on the edge device as an integral part of the acquisition pipeline, operating concurrently with imaging and reconstruction. A rigorous benchmark across multiple compression levels would therefore require re-imaging the multicenter dataset under matched acquisition conditions to capture interactions with acquisition cadence and focus stacking. To maintain strict comparability across centers and minimize workload variation, we fixed the imaging and compression parameters for all reported results.

At the selected high-quality setting, the model achieved high and reproducible performance across all sites, indicating that compression at this level does not measurably affect downstream AI inference. To clarify this for readers, we added the following statement to the Methods section (Multicenter evaluation):

“All slides were imaged using our whole-slide edge tomograph at high image-quality settings, acquiring 40 Z-layers per field, and subsequently processed using the 11-class detector–classifier that included the navicular cell class.”

Referee 2’s Remarks on Code Availability:

The code shared on Github has limited information about how the models (YOLOX, MaxViT) were adopted in their pipeline.

Authors’ Response:

We appreciate the reviewer’s request for clearer information on how YOLOX and MaxViT are implemented in our pipeline. The inference workflow and configuration are described in the Methods sections (Detection of cell nuclei, Extraction of in-focus single-cell images centered on nuclei, and Classification of single-cell images using MaxViT), which detail the detector parameters, Z-slice grouping, focus selection, cropping strategy, input normalization, generation of the class-probability (CMD) vector, and slide-level aggregation.

As clarified in the revised Code availability statement, the implementation code for model training and the backend including the AI inference pipeline contains proprietary components and therefore is not publicly released to protect intellectual property.

TO REFEREE 3

We sincerely thank Referee 3 for his/her thoughtful and constructive feedback on our manuscript. We have carefully considered all comments and have revised the manuscript accordingly. In this response letter, we provide a point-by-point response, where each comment (shown in italic) is followed by our detailed reply. All corresponding changes in the manuscript are highlighted in red for clarity (see the revised version).

Referee 3’s Comment 1:

I co-reviewed this manuscript with one of the reviewers who provided the listed reports.

Authors’ Response:

We thank the reviewer for taking the time to read our manuscript and provide feedback.

TO REFEREE 4

We sincerely thank Referee 4 for his/her thoughtful and constructive feedback on our manuscript. We have carefully considered all comments and have revised the manuscript accordingly. Below, we provide a point-by-point response, where each comment (shown in italic) is followed by our detailed reply. All corresponding changes in the manuscript are highlighted in red for clarity (see the revised version).

Referee 4’s Major Comment 1:

The manuscript by Goda et al. presents a timely end-to-end platform for autonomous cytopathology. The authors are to be commended for comprehensively addressing several long-standing challenges in digital cytopathology, particularly the data bottleneck associated with high-resolution 3D whole-slide imaging. I believe this is a significant contribution to the field. The 'whole-slide edge tomography' system, which cleverly leverages edge computing and real-time compression, and the 'Cluster of Morphological Differentiation' (CMD) analytical framework, which provides an interpretable, flow cytometry-like paradigm for AI analysis—are particularly impressive. While the study is of high quality, I recommend addressing the following points to further strengthen the manuscript before publication.

Authors' Response:

We sincerely thank the reviewer for the positive and encouraging comments on our work. We greatly appreciate the recognition of our whole-slide edge tomography system and the CMD analytical framework, as well as the acknowledgment of our efforts to address the data bottleneck in high-resolution 3D whole-slide imaging.

Referee 4's Major Comment 2:

The manuscript adopts HEVC as the compression technique, leveraging inter-layer redundancy for compression. (1) Have the authors considered or compared other standards specifically designed for 3D medical imaging, such as JPEG 2000? (2) In addition, the manuscript argues that image quality is sufficient for diagnosis when PSNR exceeds 40 dB, based on PSNR values and visual inspection. Beyond visual assessment, have the authors conducted more rigorous validation—such as having pathologists perform diagnostic evaluations on both compressed and uncompressed images to compare diagnostic consistency? Furthermore, have the authors evaluated whether compression at different quality levels (high, medium, low) affects downstream tasks like AI model training or inference? This would help clarify whether compression significantly impacts analysis performance.

Authors' Response:

We appreciate the reviewer's thoughtful comments about compression choice and validation, and potential impact on analysis performance.

(1) Codec choice (HEVC vs. others)

We performed a preliminary assessment comparing several compression standards and found that HEVC provided superior visual quality at lower bitrates and higher throughput on our edge device (i.e., whole-slide edge tomograph). Using a representative DZI viewer field, we compressed each Z-slice with baseline JPEG and plot the file size-PSNR trade-off; the test image and the corresponding curve are included inline in this response. At the effective per-image budget implied by our system (see below, ~0.6 MB per Z-image), JPEG achieves only ~31-32 dB. Even when the file size rises to 1.27-3.35 MB, PSNR remains 34.9-35.3 dB, and at 14.6 MB it is 36.1 dB.

We also tested JPEG2000 (JP2) on a second field and observed 35.26 dB at 5.91 MB, 35.92 dB at 17.73 MB, and 36.13 dB at 27.15 MB (see the table below).

Method	Parameter	File size (kB)	PSNR (dB)
jpeg	70	1,266.20	34.91
jpeg	80	1,894.70	35.07
jpeg	90	3,346.90	35.34
jpeg	100	14,579.10	36.08
jp2	100	5,910.70	35.26
jp2	300	17,734.50	35.92
jp2	1000	27,153.70	36.13

By contrast, the HEVC settings actually deployed in this study deliver typical PSNRs of ~43 dB (High), ~42 dB (Medium), and ~41 dB (Low) as shown in Fig. 2a. From Fig. 2b, a slide recorded at Z = 10 and High quality is ~1000 MB; with 170-175 spots × 10 Z-layers, this corresponds to ~0.6 MB per Z-image. Thus, at a comparable budget (~0.6 MB/image), HEVC ≈ 43 dB while JPEG ≈ 31-32 dB; an advantage of roughly +10–12 dB for HEVC. Moreover, JP2 remains below ~36 dB even at ~10-40× larger file sizes than this budget. Although we did not test the lossless mode of JP2 here, it would, by definition, preserve pixel values exactly; however, the resulting file sizes would be much larger and therefore incompatible with our real-time, edge-deployed tomographic workflow and storage/throughput constraints. These quick checks were engineering evaluations to de-risk codec selection for an online tomographic workflow on an edge device; they were performed at a single operating point and, for JPEG vs. JP2, on different source fields, so we did not present them as a formal benchmark in the manuscript. We chose HEVC because it exploits inter-slice redundancy and is hardware-accelerated on our platform, enabling continuous streaming with low latency. If we expand codec comparisons in future work, we will prioritize modern, hardware-accelerated codecs (e.g., AV1) that fit edge deployment constraints rather than legacy intra-image standards.

(2) Impact of compression

For AI input validity, we fixed the compression level to the “High” setting and applied it consistently across all centers in the multicenter study. Under this condition, the AI model achieved high and reproducible performance across all sites, indicating that compression at this level does not measurably affect inference accuracy. Because this study primary focuses on autonomous cytological analysis rather than human-observer evaluation of perceived image quality, reader studies comparing compressed and uncompressed images were considered beyond the present scope. Nonetheless, we agree that systematic evaluation of

compression-level effects, including their impact on model calibration and latency, will be valuable, and we plan to pursue these analyses in future controlled studies.

To enhance transparency, we have added the exact imaging settings used in the multicenter study to the Methods section (Multicenter evaluation) as follows:

“All slides were imaged using our whole-slide edge tomograph at high image-quality settings, acquiring 40 Z-layers per field, and subsequently processed using the 11-class detector–classifier that included the navicular cell class.”

Referee 4’s Major Comment 3:

Regarding the definition and justification of “Clinical-Grade,” a clearer explanation would be appreciated. Although the model achieved an impressive AUC > 0.99 at the cell classification level, the AUC at the slide (case) level—which better simulates clinical diagnosis—for distinguishing LSIL+ and HSIL+ dropped to 0.81 and 0.82, respectively. Could the authors further discuss potential reasons for this performance decline? It would be ideal to provide a human expert benchmark—for example, what is the inter-observer agreement rate among pathologists on the same set of 318 cases? If the system’s AUC of 0.81/0.82 is comparable to or exceeds the average performance of human experts, then the claim of “Clinical-Grade” becomes much more compelling.

Authors’ Response:

We appreciate the reviewer’s request for a clearer definition of “clinical-grade” and for benchmarking against human experts. In our study, clinical-grade refers to slide-level performance under real-world clinical constraints, including triage endpoints at the slide level, robustness across multiple centers and preparation methods, and taxonomy extensibility, supported by evidence that the system performs comparably to or better than standard cytology operating points when evaluated against an orthogonal clinical reference.

Regarding the apparent discrepancy between the cell-level AUC (>0.99 with MaxViT) and the slide-level AUCs, these metrics capture fundamentally different targets. The cell-level AUC reflects single-cell discrimination under controlled imaging conditions, whereas slide-level diagnosis aggregates predictions across tens of thousands to hundreds of thousands of cells per slide. Even a 1% per-cell error can yield ~100 misclassifications per 10,000 cells – non-negligible when the number of truly positive cells is on the order of tens to a few hundred. Additional factors such as sampling variability, staining and site heterogeneity, and nonuniform cell density further widen this gap. Consequently, achieving high slide-level accuracy is substantially more challenging and, in our view, a more clinically meaningful benchmark, than achieving high cell-level classification performance.

In our updated multicenter evaluation across four institutions (n = 1,124), the model achieved slide-level AUCs of 0.86-0.91 for LSIL+ and 0.89-0.97 for HSIL+, which remained stable across a broad range of positive-cell confidence thresholds (Figs. 7e-7g, also see below).

Figure 7 | Multicenter evaluation of clinical performance. **e**, Slide-level ROC curves for detecting LSIL⁺ (LSIL, ASC-H, HSIL, SCC) and HSIL⁺ (HSIL, SCC) on the AI model, computed separately by center; C (n = 318), T (n = 222), K (n = 385), J (n = 199), using AI-derived cell counts as predictors. AUCs are shown in the legend for each center and endpoint. **f**, **g**, Threshold-AUC sensitivity for the LSIL⁺ (**f**) and HSIL⁺ (**g**) endpoints, computed per center (C, T, K, J) on the same datasets as in **e**. For each endpoint, we vary the positive-cell probability threshold from 0.60 to 0.99 and plot AUCs (lines) with 95% bootstrap confidence intervals (shading; stratified resampling). Across centers, AUCs remain stable up to ~0.9 and drop sharply beyond this threshold. **h**, **i**, Slide-level ROC curves for AI-based detection of HPV positivity with 95% confidence bands; human cytology operating points (ASC-US⁺, LSIL⁺) are overlaid; **h** for all centers, **i** for only Center C. In the all-centers analysis, the AI model significantly exceeded ASC-US⁺ at matched specificity and matched sensitivity (both $p = 0.034$) but was inferior to LSIL⁺ (both $p < 0.001$). At Center C, where cytology and HPV were sampled on the same day, the AI model significantly exceeded ASC-US⁺ at matched specificity and sensitivity (both $p = 0.002$) and showed no significant difference versus LSIL⁺.

As for inter-observer variability, we did not collect multiple independent slide-level reads per case; thus, direct estimation of human inter-observer agreement was not possible. Instead, we benchmarked the AI system against HPV positivity as an orthogonal clinical standard and overlaid human operating points (ASC-US⁺, LSIL⁺) on the AI ROC curves to contextualize performance. Across centers, the AI model significantly outperformed ASC-US⁺ triage at matched specificity and sensitivity (Fig. 7h, also see above; both $p = 0.034$), and at Center C, where cytology and HPV sampling were obtained on the same day, AI performance was comparable to LSIL⁺ triage (Fig. 7i, also see above).

To further connect the cell-level and slide-level evidence, we additionally provide qualitative spatial concordance between AI-reported LSIL cells and expert annotations (Extended Data Fig. 6 and also see below). In representative whole-slide tomograms, high-probability AI detections co-localize with pathologist-marked LSIL cells, supporting the face validity and interpretability of the cell-level outputs that underlie slide-level predictions. This visualization complements the quantitative multicenter findings (AUCs of 0.86-0.91 for LSIL⁺ and 0.89-0.97 for HSIL⁺) and the human benchmark against HPV positivity, where the AI system exceeded ASC-US⁺ triage across centers and matched LSIL⁺ at Center C (Figs. 7h and 7i, also see above).

a

b

c

d

Extended Data Figure 6 | Whole-slide overview and zoomed comparisons of AI detections and expert annotations. *a*, Whole-slide image of a cervical liquid-based cytology slide from Center T. Yellow triangles mark cells flagged by the AI at LSIL probability ≥ 0.75 ; black triangles mark locations labeled by a cytotechnologist on the same slide. Light-blue circled areas (*b–d*) indicate regions shown at higher magnification. Scale bar, 2.5 mm. *b–d*, Enlarged views of the regions indicated in *a*. Red circles denote

cytotechnologist annotations; black bounding boxes indicate LSIL cells detected by the AI, with the number at the lower left giving the LSIL probability for each cell. Scale bars, 50 μ m.

Together, these results substantiate our use of the term clinical-grade: the system performs robustly at the slide level across sites and preparation methods, while its cell-level detections align with expert judgment in situ, enabling transparent review, interpretability, and error analysis.

Referee 4's Major Comment 4:

Figures 2 and 3, which demonstrate system performance and imaging performance, appear somewhat simplified. The authors are encouraged to include additional quantitative analyses—such as how compression impacts downstream task accuracy, and comparisons with gold-standard imaging tools. These would help reinforce the system's overall performance claims.

Authors' Response:

We appreciate the reviewer's request for additional quantitative analyses to further support system and imaging performance. The image compression in our system is performed directly on the edge device as part of the acquisition pipeline. We conducted a preliminary assessment comparing several compression standards and found that HEVC provided superior visual quality at lower bitrates and higher throughput on our edge device. Because this test was not a controlled benchmark (single operating point, no parameter sweep), we did not include these plots in the manuscript. For real-time tomographic workflows requiring continuous data streaming and low-latency processing, HEVC offers an optimal balance between compression efficiency and hardware acceleration. In future codec comparisons, we plan to systematically benchmark modern, hardware-accelerated codecs such as AV1, and to quantify how different compression levels affect model calibration, inference accuracy, and latency.

Regarding comparisons with "gold-standard" imaging tools, we note that no widely accepted benchmark currently exists for high-resolution digital imaging of cytology samples. The intrinsically 3D structure and densely overlapping cell structures in cytology slides have long posed challenges for complete digitization, preventing the establishment of a consistent reference system. Within this context, our platform represents a practical and scalable solution that delivers clinically acceptable image quality while enabling robust, AI-driven analysis suitable for real-world diagnostic use.

Referee 4's Major Comment 5:

The dataset primarily consists of samples from a single institution (Cancer Institute Hospital of JFCR). Given that the model was trained at a leading hospital in Japan, how robust is its performance on samples from other institutions, with different staining protocols or patient demographics? If feasible, even a preliminary external validation using a small number of samples from other institutions would strengthen the study. Alternatively, this limitation should be clearly stated in the discussion.

Authors' Response:

We thank the reviewer for this important point. In the revised manuscript, we have added a prospective multicenter evaluation across four centers in Japan ($n = 1,124$ slides): a cancer-specialty hospital (C), two university hospitals (T and J), and a community health screening center (K), representing diverse patient populations and Pap-staining workflows. The model achieved slide-level AUCs of 0.86-0.91 for LSIL⁺ and

0.89-0.97 for HSIL⁺ across centers (Fig. 7e, also see below), demonstrating robust generalization beyond the original training institution. We also provide spatial concordance between AI-identified lesions and expert annotations (Extended Data Fig. 6, also see below). The Results and Methods sections (Multicenter evaluation) have been updated accordingly.

Figure 7 | Multicenter evaluation of clinical performance. e, Slide-level ROC curves for detecting LSIL⁺ (LSIL, ASC-H, HSIL, SCC) and HSIL⁺ (HSIL, SCC) on the AI model, computed separately by center; C (n = 318), T (n = 222), K (n = 385), J (n = 199), using AI-derived cell counts as predictors. AUCs are shown in the legend for each center and endpoint.

a

b

c

d

Extended Data Figure 6 | Whole-slide overview and zoomed comparisons of AI detections and expert annotations. a, Whole-slide image of a cervical liquid-based cytology slide from Center T. Yellow triangles

mark cells flagged by the AI at LSIL probability ≥ 0.75 ; black triangles mark locations labeled by a cytotechnologist on the same slide. Light-blue circled areas (b–d) indicate regions shown at higher magnification. Scale bar, 2.5 mm. **b–d**, Enlarged views of the regions indicated in a. Red circles denote cytotechnologist annotations; black bounding boxes indicate LSIL cells detected by the AI, with the number at the lower left giving the LSIL probability for each cell. Scale bars, 50 μm .

Referee 4's Minor Comment 1:

In Figure 4a, the sensitivity and accuracy for “Glandular cell” classification are notably low—the lowest among all categories. This appears to be a weakness. It is recommended to briefly mention this in the results or discussion section and analyze possible causes.

Authors' Response:

We thank the reviewer for highlighting the relatively lower classification performance of the “Glandular cell” category in Figure 4a. This category is inherently challenging, even for expert cytologists, because key morphological features, such as nuclear polarity (e.g., nuclei aligned to one side of the cytoplasm), can vary considerably depending on cell orientation. In addition, glandular cells frequently appear in clusters, which complicates assessment of individual cell morphology.

Despite these difficulties, we believe it is important to retain “Glandular cell” as a classification category. Its inclusion improves the overall sensitivity of the system for detecting abnormal cells and broadens the model’s clinical utility. While perfect accuracy in this category is not essential for the current objectives, incorporating it strengthens the platform’s capacity to capture diagnostically relevant cellular diversity.

To address the reviewer’s suggestion, we added the following statement to the Results section:

“The relatively lower classification performance for glandular cells can be attributed to intrinsic variability in nuclear polarity and frequent clustering, which complicate single-cell morphological assessment even for expert cytologists. While perfect accuracy in this class is not critical for the present objectives, including it enhances the model’s capacity to capture diagnostically relevant cellular diversity.”

Referee 4's Minor Comment 2:

It will be interesting to include some discussions with the recent development of high-throughput mesoscale imaging systems, which can already cover wide-field-of-view and high-resolution for the gigapixel-imaging applications with the proposed systems.

Authors' Response:

We thank the reviewer for this constructive suggestion. We agree that recent high-throughput mesoscale imaging systems have achieved remarkable progress in gigapixel-scale acquisition by expanding optical parallelism and field-of-view coverage. We have now explicitly discussed these developments in the Discussion section to clarify how our approach complements, rather than competes, such systems. Specifically, while mesoscale scanners focus on wide-field, high-throughput imaging through multi-camera or multiplexed optical architectures, our platform addresses a distinct system-level bottleneck: real-time data handling and integration for autonomous cytological analysis. By co-designing the imaging, compression, and

AI-inference processes within the tomograph, we enable an end-to-end workflow that eliminates the need for large-volume data transfer and supports clinic-ready, low-latency operation.

To make this distinction clear, we added the following statement to the Discussion section:

“While high-throughput mesoscale gigapixel imaging systems, including multi-camera array scanners, continue to advance³⁷, our platform emphasizes a seamless end-to-end clinical workflow, from edge-side compression and secure data transfer to on-site AI inference, enabling real-time operation in routine diagnostic settings. As data handling and system integration for mesoscale imaging mature, such approaches may serve as complementary wide-field front ends to our 3D morphology-aware analytical framework.”

Referee 4's Minor Comment 3:

Codes are appropriate.

Authors' Response:

We thank the reviewer for checking our codes.

TO REFEREE 5

We sincerely thank Referee 5 for his/her thoughtful and constructive feedback on our manuscript. We have carefully considered all comments and have revised the manuscript accordingly. Below, we provide a point-by-point response, where each comment (shown in italic) is followed by our detailed reply. All corresponding changes in the manuscript are highlighted in red for clarity (see the revised version).

Referee 5's Comment 1:

This manuscript described the effort to develop an intriguing and unique AI-based technique for cytology analysis, mainly for liquid-based cervical cytology (Pap smears). Pap smear-based cervical cancer screen is the most successful cancer control story in human history, as it has saved many women's lives. Recently there has been a shift from Pap smear to HPV testing for as a primary tool for cervical cancer screen. This is partly due to the fact that Pap smear cytology is subjective and requires trained cytologists, whereas there is a shortage of cytologists in developing countries that typically have high burden for cervical cancer. Thus, there has been an interest in developing AI-tools that can be used for cervical Pap smears. Several AI-systems have already developed and commercially available for Pap smear cytology (for example, FDA-cleared Hologic's Genius Cervical AI system, etc.). Many of the tools developed so far have been used as an assistant tool for cytologist, not fully autonomous. By employing whole-slide edge tomography, high quality 3-D imaging, and cluster of morphological differentiation (CMD), the authors show that the technique described in the manuscript appears to be a fully autonomous platform that is distinctive from others in the field. There are several questions and suggestions as presented below:

Authors' Response:

We sincerely thank the reviewer for the thoughtful context and encouraging remarks about the need for scalable, objective Pap-smear analysis and the field's pivot toward HPV-anchored screening. Our goal is precisely to complement HPV testing with fully autonomous triage cytology by coupling high-speed, in-scanner

whole-slide tomography and on-the-fly HEVC compression with CMD-based population analysis, so that clinically actionable, slide-level readouts can be produced without human gating or review. The manuscript documents the end-to-end, clinic-ready performance of this pipeline – ~3–8 min/slide acquisition with <100 ms viewer latency; preservation of subnuclear detail at the compression levels used; cell-level AUCs > 0.99 for LSIL/HSIL/adenocarcinoma; and slide-level signals that align with HPV status and cytology severity (AUC 0.81 for LSIL⁺, 0.82 for HSIL⁺), underscoring the distinction between our autonomous system and prior assistive approaches. We address the reviewer’s specific questions point-by-point below.

Referee 5’s Comment 2:

Title: The title implies the technique may be applicable to cover the entire field of cytopathology practice, from cervical Pap smear to FNA, etc., but most of the work presented are limited to liquid-based cervical cytology. In fact, the method may be more applicable for high cellularity samples such as Pap smears (with at least 10,000 cells), but may not be suitable for low cellularity samples (such as urine or CSF), or samples prepared by traditional smear technique. Therefore, the title should be stated more precisely for liquid-based cervical Pap smears.

Authors’ Response:

We appreciate the reviewer’s point regarding scope. Our title is intended to emphasize a platform-level advance: a sample-agnostic pipeline that integrates rapid whole-slide 3D acquisition, in-tomograph HEVC compression, and CMD-based population analysis. None of these components rely on cervix-specific markers; rather, they are designed to generalize to any cytology domain where sufficient morphological signal is present. We chose cervical cytology as the initial application because it is (i) clinically important and globally prevalent, with the largest volume of cytological tests worldwide; (ii) high-cellularity, providing a rigorous test of throughput; and (iii) technically demanding for 2D scanners due to thick samples and sparse abnormalities, making it an ideal case to stress-test fidelity, scalability, and autonomous analysis under realistic workloads. Thus, while the present study focuses on cervical samples, the platform is broadly applicable to other cytology contexts. To substantiate extensibility beyond the cervix, we added non-cervical exemplars (fibroadenoma and follicular neoplasm to Figures 3e and 3f, and Extended Data Figures 4e and 4f (also see below).

Figure 3 | Representative reconstructed 3D images of cytology samples acquired at 1 μ m Z-intervals. e, Fibroadenoma (benign breast tumor). f, Follicular thyroid neoplasm (follicular thyroid carcinoma). These

samples correspond to those shown in Extended Data Figure 4 and are visualized along the XYZ axes to reveal volumetric cellular architectures and spatial relationships. Images highlight depth-resolved morphological features and structural deformations. Scale bars: 50 μ m.

Extended Data Figure 4 | Depth-resolved tomograms. Representative tomographic slices of cytology samples acquired at 1 μ m Z-intervals and displayed at 5 μ m steps. Shown are keratinizing squamous cell carcinoma (a), non-keratinizing squamous cell carcinoma (b), HPV-associated adenocarcinoma (c), HPV-independent adenocarcinoma (d), fibroadenoma (e), and follicular thyroid neoplasm (f). Slices highlight depth-resolved morphological features and structural deformations across the z-stack. Scale bars: 50 μ m.

Referee 5's Comment 3:

For the claim that the platform is a “clinical-grade autonomous”, the authors provided some data, one of which is that the system may be more “accurate” than cytologists in classifying cells for LSIL or HSIL categories in a set of Pap smears (by combining the HPV test result, Fig 6). Evidence that the system can actually be used in clinical setting in a reproducible and reliable manner, is not presented. Though full clinical validations may require future multi-center prospective trials, some preliminary pilot data will be more convincing.

Authors' Response:

We appreciate the reviewer's thoughtful suggestions. Following the reviewer's advice, we have expanded our clinical performance evaluation by incorporating additional patients and centers, encompassing four institutions; Cancer Institute Hospital of JFCR (C), University of Tsukuba Hospital (T), Kaetsu Comprehensive Health Development Center (K), and Juntendo University Urayasu Hospital (J), comprising a total of 1,124 cervical liquid-based cytology slides (C: n = 318; T: n = 222; K: n = 385; J: n = 199). The updated model (identical architecture and hyperparameters, with no site-specific tuning) achieved slide-level AUCs of 0.86-0.91 for LSIL⁺ and 0.89-0.97 for HSIL⁺ across the four centers, demonstrating robust generalization to external cohorts with differing sample acquisition and workflow characteristics. As shown in Fig. 7e (also shown below), the ROC curves were consistently excellent across all centers.

Figure 7 | Multicenter evaluation of clinical performance. e, Slide-level ROC curves for detecting LSIL⁺ (LSIL, ASC-H, HSIL, SCC) and HSIL⁺ (HSIL, SCC) on the AI model, computed separately by center; C (n = 318), T (n = 222), K (n = 385), J (n = 199), using AI-derived cell counts as predictors. AUCs are shown in the legend for each center and endpoint.

Furthermore, using HPV positivity as an orthogonal reference, we compared AI-based triage with expert triage: across centers, the AI model outperformed conventional ASC-US–based triage at matched specificity and achieved higher overall discriminability, indicating improved clinical utility (Figs. 7h and 7i, also see below).

Figure 7 | Multicenter evaluation of clinical performance. h, i, Slide-level ROC curves for AI-based detection of HPV positivity with 95% confidence bands; human cytology operating points (ASC-US⁺, LSIL⁺) are overlaid; **h** for all centers, **i** for only Center C. In the all-centers analysis, the AI model significantly exceeded ASC-US⁺ at matched specificity and matched sensitivity (both $p = 0.034$) but was inferior to LSIL⁺ (both $p < 0.001$). At Center C, where cytology and HPV were sampled on the same day, the AI model significantly exceeded ASC-US⁺ at matched specificity and sensitivity (both $p = 0.002$) and showed no significant difference versus LSIL⁺.

Collectively, these results provide cross-site evidence that the platform performs reliably in real-world settings, strengthening the claim of a clinical-grade autonomous system, while broader international validation remains an important future direction.

Referee 5's Comment 4:

Figure 2b, the bar color for High and Medium is identical, is that intentional?

Authors' Response:

We thank the reviewer for pointing this out. In the original figure, High (dark blue), Medium (blue), and Low (light blue) were represented using different shades of blue. However, we recognize that the distinction may not have been sufficiently clear on all displays. We have updated Figs. 2a and 2b (also see below) to use more distinguishable color tones to improve clarity.

Figure 2 | System performance. *a*, Histograms of PSNR values obtained from 300 tomographic images acquired from three liquid-based cytology slides under high, medium, and low HEVC compression settings (3 slides × 10 imaging sections × 10 Z-layers per slide). *b*, File sizes of whole-slide 3D image datasets under varying compression levels and Z-layer counts (10, 20, 40). Data size increases with the number of Z-layers and decreases with more aggressive compression. At high image quality, 10-layer datasets average ~1 GB per slide.

Referee 5's Comment 5:

Figure 2c and 4k, what about SCC cells?

Authors' Response:

We thank the reviewer for the question regarding SCC cells. Representative SCC images are included in the manuscript (for example, in Figs. 3a, 3b, 3e, and 3f) to illustrate the imaging capability of our system. However, as noted in the manuscript, the AI analysis shown in Fig. 4 and onward was developed and evaluated using screening-purpose samples, which did not include a sufficient number of SCC cases for training or validation. Therefore, SCC-specific classification was not included in the current AI model.

Nonetheless, as shown in Figs. 6e and 6f, the system successfully identifies a substantial number of cells in SCC cases that are highly likely to be LSIL or HSIL. This suggests that the platform remains capable of detecting abnormal findings in SCC samples, even without explicit training on this category. We consider this a promising direction for future model development using expanded datasets that include more SCC cases.

Referee 5’s Comment 6:

Methods: it appears both Thinprep and Surepath slides were used in the study. The slides prepared by the two methods are quite different. A major difference is that Surepath-prepared slide contains more 3-dimensional cellular clusters and has more overlapping cells. Any differences noted in the analysis? More discussion will also be helpful.

Authors’ Response:

We appreciate the reviewer’s insightful observation. We agree that the preparation method can influence cytology image characteristics. To evaluate this effect, we performed stratified ROC analyses at Center C, where both ThinPrep and SurePath slides were available, using identical acquisition and analysis parameters (no re-tuning). In both preparation groups, slide-level performance remained high (AUC ≥ 0.90 for LSIL⁺ and HSIL⁺), despite the thicker and more overlapping cell clusters typical of SurePath (Extended Data Figs. 8a-8c, also see below). We have added a concise note in the Methods section indicating that the 3D tomographic workflow, by resolving cellular overlap and preserving nuclear morphology, likely contributes to this preparation-agnostic robustness, while broader multi-site preparation effects are identified as an area for future investigation.

“Comparison of sample preparation methods

To assess potential influence of sample preparation methods, we compared SurePath and ThinPrep slides with comparable age distributions (Extended Data Figure 8a). Both preparation methods yielded similarly favorable ROC curves and AUCs (Extended Data Figures 8b and 8c), demonstrating that model performance is robust to slide preparation.”

Extended Data Figure 8 | Comparison of sample preparation methods. a, Age distributions for the SurePath and ThinPrep cohorts in the Cancer Institute Hospital of JFCR (median line, IQR; whiskers = 1.5×IQR; outliers omitted). Medians [IQR]: SurePath 50.0 [42.0–60.0], ThinPrep 45.0 [37.0–50.0]. **b, c,** ROC curves for detecting LSIL⁺ (LSIL, ASC-H, HSIL, SCC) and HSIL⁺ (HSIL, SCC) at the Cancer Institute Hospital

of JFCR (C), stratified by liquid-based cytology sample preparation: SurePath (b, n = 265) and ThinPrep (c, n = 53). ROC curves use AI-derived cell counts as predictors; AUCs are shown in the legends for each endpoint.

Referee 5's Comment 7:

Not all codes are available for review to protect IP.

Authors' Response:

We appreciate the concern. As noted in the Code availability section, certain modules remain proprietary to protect intellectual property. Developing clinical-grade technology requires substantial investment that is viable only with appropriate IP protection; accordingly, we must balance transparency with stewardship. Within these constraints, we have disclosed all details necessary for independent assessment and appreciate the reviewer's understanding.

TO REFEREE 2

We sincerely thank Referee 2 for his/her thoughtful and constructive feedback on our manuscript. We have carefully considered all comments and have revised the manuscript accordingly. Below, we provide a point-by-point response, where each comment (shown in *italic*) is followed by our detailed reply.

Referee 2's Remark 1:

After taking a closer look at all the referees' comments and the author's responses, I believe the author has addressed all my concerns. Recognizing that conducting additional ablation studies on different AI models (such as alternatives to YOLOX, as well as other pathology foundation models) and data augmentation would require substantial extra effort, I find the author's current response adequate.

Authors' Response:

We sincerely thank the referee for re-examining our previous responses and for confirming that the earlier concerns have been satisfactorily addressed. We agree that additional ablation studies on alternative detection and foundation models would be valuable future work, and we have retained a brief discussion of these directions in the revised manuscript while keeping the focus on the clinically validated configuration presented here. No further methodological changes were introduced specifically for this point in the current revision.

Referee 2's Remark 2:

Regarding the multi-center study, I believe the authors' revision has fully addressed both my concern and Referee #4's concern.

Authors' Response:

We are grateful for the referee's positive assessment of the revised multicenter analysis. In the current revision, we have preserved the expanded multicenter evaluation (1,124 samples across four centers) and have further refined the presentation by clarifying sample sizes, cohort characteristics and statistical reporting in the figure legends and Methods, while keeping the overall design and conclusions of the multicenter study unchanged.

Referee 2's Remark 3:

In addition, I made a throughout review of statistical analyses, including the suitability of the tests used, the accuracy of the descriptions for error bars, as well as probability values. After the inspection, all looks appropriate to me: The paper uses one-sided Mann-Whitney U tests for pairwise comparisons (e.g., NILM vs. higher categories, HPV⁺ vs. HPV⁻) with Benjamini-Hochberg (BH) correction. This is acceptable for non-parametric data with skewed distributions (cell counts are clearly non-Gaussian). While I personally believe multiple testing correction (BH) may not be strictly necessary in this context, including the Q-values adds rigor. Error bars in the latency and imaging performance plots (Figure 2e & Extended Data Figure 2d) are described as standard deviations, which is appropriate.

Authors' Response:

We sincerely appreciate the referee's thorough review of our statistical analyses and the positive evaluation of our choice of tests, multiple-testing correction and error-bar reporting. In the current revision, we have kept

the underlying statistical methods unchanged (including the one-sided Mann-Whitney U tests with Benjamini-Hochberg correction for the relevant comparisons), but we have improved the clarity and completeness of the statistical descriptions in response to the editors' reporting requests. We thank the referee again for confirming the overall appropriateness and rigor of the statistical framework.

Referee 2's Remarks on Code Availability:

Code is appropriate.

Authors' Response:

We sincerely thank the referee for reviewing and endorsing our code availability. In line with the journal's editorial guidance received at this stage, we have further expanded and clarified our public code release in the revised submission. All custom code central to the study (including edge-device image processing, AI model training, backend infrastructure and the AI inference pipeline, as well as scripts for downstream analysis and figure generation) is now archived as a versioned release on Zenodo and linked in the updated Code Availability statement.

TO REFEREE 3

We sincerely thank Referee 3 for his/her thoughtful and constructive feedback on our manuscript. We have carefully considered all comments and have revised the manuscript accordingly. In this response letter, we provide a point-by-point response, where each comment (shown in italic) is followed by our detailed reply.

Referee 3's Comment 1:

I co-reviewed this manuscript with one of the reviewers who provided the listed reports.

Authors' Response:

We thank the reviewer for taking the time to read our manuscript and provide feedback.

TO REFEREE 4

We sincerely thank Referee 4 for his/her thoughtful and constructive feedback on our manuscript. We have carefully considered all comments and have revised the manuscript accordingly. Below, we provide a point-by-point response, where each comment (shown in italic) is followed by our detailed reply.

Referee 4's Remark 1:

I think the authors have done great work during revision by providing more comprehensive benchmarks and quantitative evaluations. These new analyses and experiments have fully addressed my previous concerns. I believe this will be a very nice and practical tool for next-generation high-throughput diagnosis, especially in the era of AI.

Authors' Response:

We are very grateful for the referee's positive assessment and encouraging comments. In the revised manuscript we have retained the expanded benchmarks and quantitative multicenter evaluations, while further refining the presentation of sample sizes, statistical methods and effect sizes in the figure legends and Methods in response to the editorial requests. We appreciate the referee's view that the system can serve as a practical tool for next-generation high-throughput diagnosis and have slightly clarified the discussion to better highlight this translational potential.

Referee 4's Remarks on Code Availability:

The codes are fine and well-illustrated.

Authors' Response:

We sincerely thank the referee for reviewing and endorsing our code availability. In line with the journal's editorial guidance received at this stage, we have further expanded and clarified our public code release in the revised submission. All custom code central to the study (including edge-device image processing, AI model training, backend infrastructure and the AI inference pipeline, as well as scripts for downstream analysis and figure generation) is now archived as a versioned release on Zenodo and linked in the updated Code Availability statement.

TO REFEREE 5

We sincerely thank Referee 5 for his/her thoughtful and constructive feedback on our manuscript. We have carefully considered all comments and have revised the manuscript accordingly. Below, we provide a point-by-point response, where each comment (shown in italic) is followed by our detailed reply.

Referee 5's Remark 1:

The resubmission addressed most of the concerns noted in the previous review adequately, with detail point-by-point responses. I don't have additional comments.

Authors' Response:

We sincerely appreciate the referee's careful re-evaluation of our resubmission and the confirmation that the previous concerns have been adequately addressed. In the current revision, we have maintained the analyses and clarifications introduced in response to the earlier round of reviews, and we have mainly focused on improving clarity, statistical reporting and formatting to meet the journal's editorial requirements.

Referee 5's Remarks on Code Availability:

No but adequately addressed.

Authors' Response:

We thank the referee for confirming that our previous treatment of code availability was acceptable. In light of the editor's subsequent guidance at this stage, we have gone further and now provide a fully public release of all custom code central to the study. This includes the edge-device image processing, AI model training,

backend inference pipeline and the scripts used for downstream analysis and figure generation, archived as a versioned release in a DOI-minting repository and described in the revised Code Availability statement.